# Spatially and temporally probing distinctive glycerophospholipid alterations in Alzheimer's disease mouse brain via high-resolution ion mobility-enabled *sn*-position resolved lipidomics

Shuling Xu [1,9], Zhijun Zhu [2,9], Daniel G. Delafield [2], Michael J. Rigby [3,4,5], Gaoyuan Lu [1], Megan Braun [3,4,5], Luigi Puglielli [3,4,6] & Lingjun Li [1,2,7,8] ✉

Dysregulated glycerophospholipid (GP) metabolism in the brain is associated with the progression of neurodegenerative diseases including Alzheimer's disease (AD). Routine liquid chromatography-mass spectrometry (LC-MS)-based large-scale lipidomic methods often fail to elucidate subtle yet important structural features such as *sn*-position, hindering the precise interrogation of GP molecules. Leveraging high-resolution demultiplexing (HRdm) ion mobility spectrometry (IMS), we develop a four-dimensional (4D) lipidomic strategy to resolve GP *sn*-position isomers. We further construct a comprehensive experimental 4D GP database of 498 GPs identified from the mouse brain and an in-depth extended 4D library of 2500 GPs predicted by machine learning, enabling automated profiling of GPs with detailed acyl chain *sn*-position assignment. Analyzing three mouse brain regions (hippocampus, cerebellum, and cortex), we successfully identify a total of 592 GPs including 130 pairs of *sn*-position isomers. Further temporal GPs analysis in the three functional brain regions illustrates their metabolic alterations in AD progression.

Glycerophospholipids (GPs) are the building blocks of cell membranes and play many critical roles in a wide variety of physiological processes, including energy storage, signal transduction, cell proliferation, and apoptosis[1,2]. Consequently, GP composition is carefully regulated to ensure proper cellular functions. A growing number of studies have demonstrated that dysregulation of lipid metabolism is associated with various pathologies, including diabetes[3], cancers[4], and neurodegenerative diseases[5]. In particular, mounting evidence has implicated that the aberrant alteration of stereospecific numbering (*sn*)-position-specific GPs in terms of abundance or the ratio of GP/lyso-GP is closely related to a variety of cancers and neurodegenerative diseases, owing to homeostatic disruption of *sn*-position selective

[1]School of Pharmacy, University of Wisconsin-Madison, Madison, WI 53705, USA. [2]Department of Chemistry, University of Wisconsin-Madison, Madison, WI 53706, USA. [3]Department of Medicine, University of Wisconsin-Madison, Madison, WI 53705, USA. [4]Waisman Center, University of Wisconsin-Madison, Madison, WI 53705, USA. [5]Neuroscience Training Program, University of Wisconsin-Madison, Madison, WI 53705, USA. [6]Geriatric Research Education Clinical Center, Veterans Affairs Medical Center, Madison, WI 53705, USA. [7]Lachman Institute for Pharmaceutical Development, School of Pharmacy, University of Wisconsin-Madison, Madison, WI 53705, USA. [8]Wisconsin Center for NanoBioSystems, School of Pharmacy, University of Wisconsin- Madison, Madison, WI 53705, USA. [9]These authors contributed equally: Shuling Xu, Zhijun Zhu. ✉e-mail: lingjun.li@wisc.edu

phospholipase and lysophospholipid acyltransferase enzymes that are involved in GP remodeling[6–8]. In the central nervous system, GPs are responsible for the proper functions of synapses, receptors, transporters, neurotransmitters, and signaling processes[9,10]. Accordingly, numerous studies have revealed that altered GP compositions and phospholipase enzyme activities in the brain region could link to neurodegeneration including Alzheimer's disease (AD)[9,11,12]. Heterogenous spatial distribution and alteration of GPs have been observed across different functional regions in accordance with aging and AD progression, which could be attributed to region-specific pathologies. Thus, performing the in-depth characterization of spatial and temporal sn-position-resolved GP profiles across complex functional brain regions would help to reveal and provide a more precise interpretation of the molecular mechanisms of AD progression underpinned by GP metabolism.

Demonstrating high sensitivity, resolution, and the capacity for structural characterization, mass spectrometry (MS) based strategies, either alone or coupled with liquid chromatography (LC-MS), have become popular choices for lipid identification and quantification[1]. However, lipidome-wide analyses with complete structural characterization still represent a long-standing analytical challenge in lipidomics[13,14]. Currently, routine lipidomics analysis workflows using collision-induced dissociation (CID)-based MS/MS can only identify GPs at the level of lipid fatty acyl compositions, but fail to reveal subtle yet important structural features such as C=C position/geometry and sn-position[7,13,15,16]. As sn-isomeric GPs are commonly present in biological mixtures in a wide dynamic range and are commonly co-eluted in LC, although we could assign the sn-position for the dominant GP component by the relative intensities of the two fatty acyl chain carboxylate anions, it would still be difficult to prove or rule out the presence of its sn-isomer. Providing comprehensive lipid structural characterization hinges on the development of alternative analytical approaches that overcome the technical obstacles of traditional methods. In recent years, significant advances have been achieved in elucidating GP sn- and C=C location isomers by chemical derivatization strategies and/or different ion activation/dissociation techniques to generate structure-specific fragment ions[13,17–22]. These include electron-induced dissociation (EID)[23], electron impact excitation of ions from organics (EIEIO)[24], ozone-induced dissociation (OzID)[25], ultraviolet photodissociation (UVPD)[26], CID and UVPD of lipid−metal ion complexes[27,28], and Paternò–Büchi (PB) reaction followed by CID[7,13,18,23]. However, it is still challenging for these strategies to precisely quantify lipid isomers, primarily due to the different dissociation efficiencies between isomers and interfering diagnostic ions from co-eluting lipid species[29]. Therefore, powerful separation techniques at intact lipid level are needed to complement ion activation and provide unequivocal structural assignment and quantification.

Ion mobility spectrometry (IMS) is a rapid gas-phase separation technique in which ions are separated by size, shape, and charge[30–32]. Collision cross section (CCS), originating from rotationally averaged measurements of the cross-sectional area of an analyte ion in the gas phase, can be directly correlated with the structure and conformation of the gas-phase ion. Coupling ion mobility (IM)-MS with LC shows promise in lipid isomer separation and annotation by providing high selectivity across four dimensions: $m/z$, retention time (RT), CCS, and MS/MS spectra[33–39]. More importantly, recent reports have demonstrated that precise and reproducible lipid CCS measurement together with machine learning algorithms could enable large-scale lipid CCS prediction. For example, Zhu and coworkers utilized support vector regression (SVR) models to predict the CCS values of a variety of lipid classes[40,41]. McLean and colleagues presented a large, unified CCS compendium to predict CCS values of compounds including 810 lipid species according to the relationship between CCS and $m/z$[42]. Recently, Zhu and co-workers reported the strategy that integrated four-dimensional (4D) ($m/z$, RT, CCS, and MS/MS) library-based matching

and rule-based refinement to reduce over-report and improve the accuracy of lipid identification[37,43]. These current 4D lipidomics studies greatly improved the depth in lipid structure characterization. However, in these studies that utilized rule-based refinement for in-depth lipid analysis, only a part of GP species was roughly identified at fatty acyl sn-position level and the existence of sn-position isomers was largely ignored. Specifically, in the rule set for lipid identification, GP species from each extracted peak was assigned to only one of the sn-isomer pairs if the intensity ratio of sn-1/sn−2 was <0.9 in negative ion mode, while the existence of corresponding sn-isomers was ignored. Additionally, the identification of GP was still at fatty acyl level when the intensity ratio of sn-1/sn−2 was ≥ 0.9. Moreover, the quantification could only be achieved for the most abundant isomer or fatty acyl sum composition. Nevertheless, these pioneering studies motivated the lipid structural annotation and quantification to the next level for providing unequivocal structural assignment of lipid isomers. However, robust implementation of such an approach for lipids with a higher level of structural elucidation requires the availability of high-resolution IM measurements. Typically, sn-position isomers exhibit around 1% differences in CCS values and therefore require IM resolving power ($R_p$) over 100 for separation[44]. This high $R_p$ was out-of-reach in previous IM-based lipidomic analyses, eliminating the capacity for any of these former methodologies to accurately measure and predict the CCS values of lipid sn-position isomers. Recently, using trapped ion mobility spectrometry (TIMS)-MS with $R_p$ up to 410 using ultra-low scanning rate, Fernandez-Lima and his team have shown the potential of high-resolution IM in discriminating phosphatidylcholine (PC) sn-isomers in human plasma[29]. High-resolution demultiplexing (HRdm) strategy was first reported in 2020, which is an extended Hadamard multiplexing and post-acquisition data processing strategy for improving the sensitivity and resolution of IM measurements without the need for instrument modifications[45]. In the past four years, a series of isomeric biomolecules in complex samples were investigated with HRdm, including glycans, peptides, metabolites, and oxidized lipids[46–48]. However, utilizing the strategy to systematically separate subtle yet crucial structural sn-position isomers of GPs in biological samples has not been widely explored in existing studies. In addition to requiring high IM $R_p$, large-scale GP sn-isomer distinguishment is also hindered by insufficient commercially available pure lipid standards with determined sn-position, thus reducing the confidence of empirical assignments. And finally, even though predictive models may be considered a worthwhile approach, very few molecular descriptors (MDs) in common simulation packages reflect differences of GP sn-position isomers[40,49]. This limitation essentially eliminates the ability to distinguish species in silico.

To overcome these challenges, in this work, we develop a high-resolution IM-MS-based 4D lipidomics strategy that leverages a machine learning-empowered library for large-scale, in-depth structural analysis of GP sn-position isomers. The use of HRdm strategy provides an increase of drift tube ion mobility spectrometry (DTIMS) $R_p$ from ~50 to 250 while still allowing millisecond IMS separation of GP sn-isomers without any instrumental modifications. We further construct a comprehensive experimental 4D GP database of 498 GPs identified from pooled mouse brain lipid extracts. These empirical measurements are redeployed to facilitate the creation of an in-depth, extended 4D library of 2500 GPs by machine learning-based prediction. With both the experimental database and the extended library, a significantly high number (>540) of GP species with sn-position information are identified and quantified from each of the three functional brain regions, revealing the spatial and temporal GP alterations in the brains of wild-type (WT) and APP/PS1 AD mouse model. By revealing significant changes in either abundance or ratios of sn-isomers in a set of GPs during aging and AD progression, we demonstrate that this developed strategy is powerful to uncover potential biomarkers for AD progression associated with dysregulated GP metabolism.

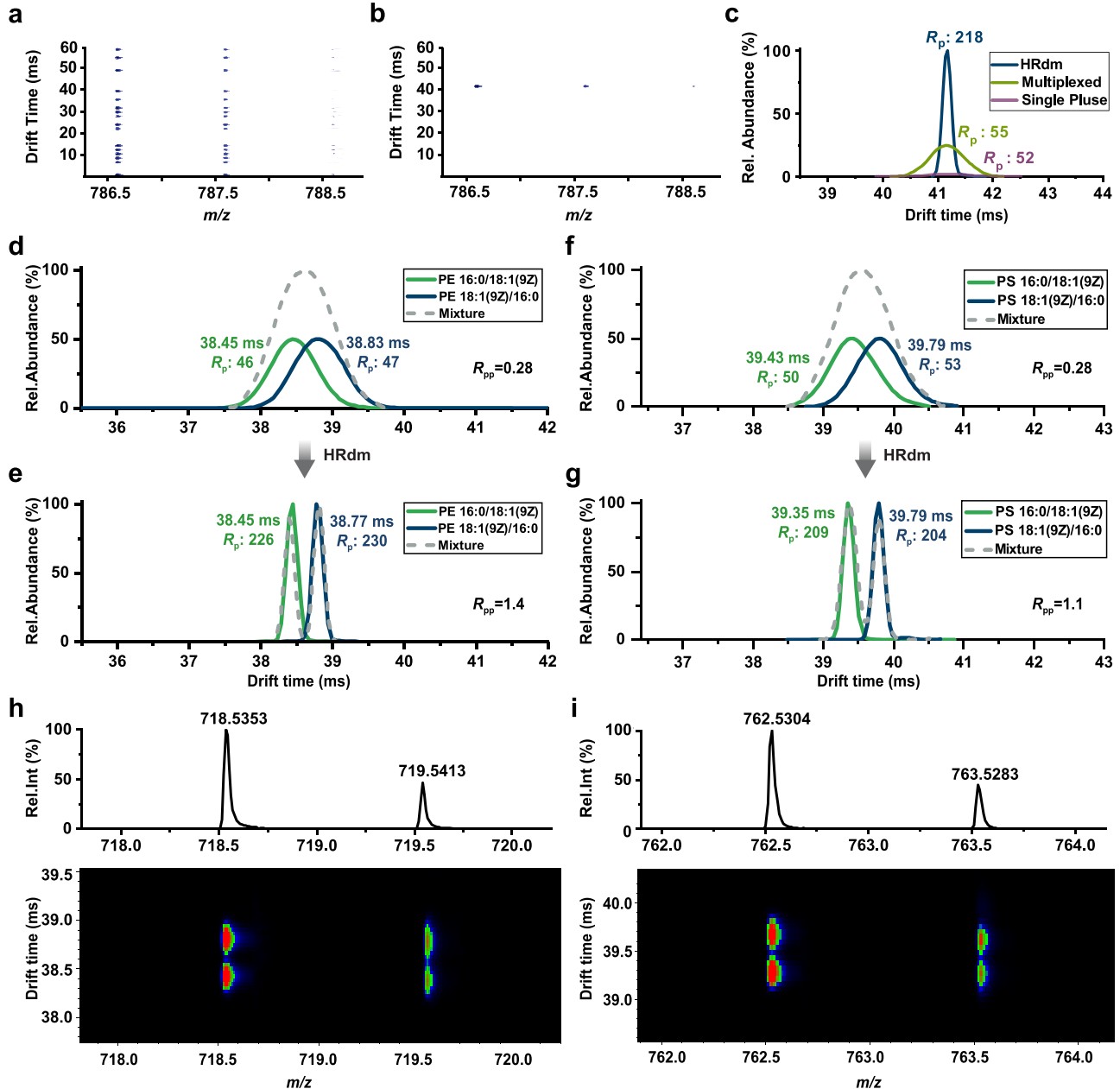

**Fig. 1 | Improved sensitivity and separation of GP *sn*-isomers with high-resolution HRdm IM-MS. a** Raw IM-MS heatmap of PC 18:1(11Z)/18:1(11Z) obtained from 5-bit multiplexing injection mode. **b** Demultiplexed IM-MS heatmap of PC 18:1(11Z)/18:1(11Z). **c** Overlaid drift spectra of the single pulse, 5-bit standard demultiplexing, and 5-bit HRdm. Overlaid drift spectra of PE 16:0/18:1(9Z), PE 18:1(9Z)/16:0, and equimolar mixture processed by standard demultiplexing (**d**)

and HRdm (**e**), and all drift spectra are normalized to maximum peak height. Overlaid drift spectra of PS 16:0/18:1(9Z), PS 18:1(9Z)/16:0, and equimolar mixture processed by standard demultiplexing (**f**) and HRdm (**g**). Mass spectra and HRdm IM-MS heatmaps of the equimolar mixture of PE 16:0/18:1(9Z) and PE 18:1(9Z)/16:0 standards (**h**) and PS 16:0/18:1(9Z) and PS 18:1(9Z)/16:0 standards (**i**). Source data are provided as a Source Data file.

## Results

### Improved sensitivity and separation of GP *sn*-isomers with HRdm IM-MS

Distinguishing the minute structural differences between GP *sn*-position isomers is a long-standing analytical challenge by IM-MS methodologies[13,18,26,27]. Recently, HRdm strategy was put forth to overcome the known limitations in IM $R_p$ and duty cycle. By using multiplexed ion injection and post-acquisition data processing, this technological development has been shown to significantly improve the sensitivity and resolution of IM-MS measurements without the need for instrument modifications[45,47]. Taking the lipid standard PC 18:1(11Z)/18:1(11Z) as an example, Fig. 1a illustrates characteristic HRdm multiplexing – the injection of 16 ion packets at

predetermined intervals across the drift time window. This multiplexing strategy results in an increased duty cycle and a decreased onset of detection saturation. The raw, multiplexed ion packets in the drift spectrum are then deconvoluted into one, resulting in a spectrum reminiscent of single pulse ion injection and bearing no difference in measured drift time (Fig. 1b). Although standard demultiplexing itself only slightly improves the IM $R_p$ (55 in standard demultiplexed mode *vs* 52 in single pulse) due to reduced space charge effects, the signal intensity increased by more than 12-fold higher when compared to single pulse mode (Fig. 1c). Further processing of the demultiplexed data by Hadamard Transform and data post-processing (see Methods), HRdm processed data revealed IM peak widths significantly narrower than those observed in either

single pulse or standard demultiplexed spectra, achieving high IM $R_p$ up to 250 across these standard trials. Because the total signal is preserved, the reduction in peak width results in an increase in peak height, providing an additional ~ 5-fold increase in sensitivity (Fig. 1c).

To ascertain whether the high IM $R_p$ from HRdm could benefit *sn*-isomer separation on a large scale, a series of isomer-pure GP standards were evaluated through both standard demultiplexing and HRdm processing with their *sn*-isomers (Fig. 1d–i). Phosphatidylethanolamine (PE) and phosphatidylserine (PS) standards were examined in their most abundant ion forms in positive mode ESI, [M + H]⁺. PE 16:0/18:1(9Z) and PE 18:1(9Z)/16:0 exhibited a small CCS or drift time difference of around 1%, precluding any mobility separation of equimolar mixtures through the modest resolution found in standard demultiplexing ($R_p$-50, Fig. 1d). Achieving $R_p$ of over 200 via HRdm, the enhanced $R_p$ readily facilitated almost baseline separation of these PE *sn*-position isomers when analyzed separately and as a mixture of two (Fig. 1e), thus increasing peak-to-peak resolution ($R_{pp}$) from 0.28 to 1.4. These results were rearticulated during our analysis of PS *sn*-isomers, PS 16:0/18:1(9Z) and PS 18:1(9Z)/16:0. Failing to differentiate equimolar mixtures of isomers through standard demultiplexing (Fig. 1f), the pair of *sn*-position isomers were successfully separated using HRdm (Fig. 1g). Generally, *sn*-position isomers, as constitutional isomers, exhibit CCS difference of approximately 1%, which needs more than 100 resolving power to achieve 10% separation and more than 200 resolving power to achieve 90% separation in low-field DTIMS instruments[50]. Owing to the high resolution afforded by HRdm, most of GP *sn*-position isomers could be separated sufficiently. The quality of a separation is quantified in terms of $R_{pp}$ with peaks deemed resolved when $R_{pp}$ exceeds 0.5. To provide a comprehensive evaluation of the separation efficiency, a series of GP *sn*-position isomers were assessed using HRdm to evaluate the effectiveness of separation by this technique (Supplementary Fig. 1). The $R_{pp}$ for GPs with different fatty acyl chain compositions from different classes were at a range from 0.78 to 1.4, indicating that each pair of isomers was successfully separated. Phosphatidylglycerol (PG), phosphatidylinositol (PI), and phosphatidic acid (PA) were predominantly present as [M + NH₄]⁺ in positive ion mode. The *sn*-isomers from these classes were also separated by HRdm in ion forms of [M + NH₄]⁺ (Supplementary Fig. 1). The IM separation of GP *sn*-isomers with sodium adduction in positive mode (Supplementary Note 1 and Supplementary Fig. 2) and their deprotonated form, [M-H]⁻, in negative mode (Supplementary Note 2 and Supplementary Fig. 3) was also investigated. Examining HRdm IM-MS heatmaps of the equimolar mixtures of the PE (Fig. 1h) and PS (Fig. 1i) *sn*-position isomers, our data reveal excellent drift time alignment across all isotope envelopes, highlighting accuracy, precision, and reproducibility of CCS measurements within HRdm. HRdm separation of GP *sn*-position isomers at various molar ratios was also demonstrated quantitatively (Supplementary Fig. 4). A mixture of isomer-pure PE 16:0/18:1(9Z) and PE 18:1(9Z)/16:0 standards at the ratios ranging from 1 to 10 were adequately separated with $R_{pp}$ of 1.01-1.46, demonstrating the consistency of separation in the complex mixture. We also validated that the relative abundance of each isomer could be truthfully reflected by the ratio of the peak areas (Supplementary Fig. 4), indicating the quantitative accuracy of HRdm. Additionally, the high accuracy in quantifying GP *sn*-isomers across a wide range of GP classes, using ion abundance in HRdm drift spectra, was validated by the well-established phospholipase A2 (PLA2) digestion method[51]. The information on GP standards was listed in Supplementary Table 1. The results indicated that the isomeric abundance obtained using IM-MS aligns consistently with that obtained via PLA2 digestion (Supplementary Note 3, Supplementary Fig. 5, and Supplementary Table 2).

In order to further demonstrate the reproducibility of HRdm technique in complex biological samples, we compared the drift spectra of isotope-encoded lipid standards spiked in pure solvent (isopropanol) and complex biological samples (mouse brain lipid extract) acquired with HRdm. As shown in Supplementary Fig. 6, all the drift spectra of deuterium-labeled lipids from the complex biological matrices could still align well with these in pure solvent, indicating that HRdm processing did not generate artificial peaks due to matrix interferences. Deuterium-labeled triacylglycerol (TG), diacylglycerol (DG), monoacylglycerol (MG), PC, PE, PS, PG, PI, PA, and sphingomyelin (SM) showed a single peak with high-resolution IM technique ($R_p$ up to 217) in both pure solvent and complex biological samples. Meanwhile, small peaks from isomers were observed in the drift spectra of LPC 18:1 (d7)/0:0 and LPE 18:1 (d7)/0:0 (Supplementary Fig. 6j-k). In order to validate that these small peaks are results from true isomers rather than artificial signals, we performed the PLA2 digestion of isopure standards to obtain the pure lyso-GP standards. As shown in Supplementary Fig. 7, all isopure lyso-GPs including LPC 18:1/0:0, LPE 18:1/0:0 appeared as a single peak and aligned well with deuterium-labeled standards. As not all these deuterium-labeled lyso-GP standards are isopure standards, we deduced that these two lyso-GPs contain small amounts of corresponding *sn*-isomers with fatty acyl at *sn*−2. This also accords with previous studies that lyso-GPs with fatty acyl at *sn*−2 have the larger CCS values[45,52]. The peaks of deuterium-labeled lyso-GP in both pure solvent and biological samples were also all aligned well. Moreover, it is important to note that our study utilized the officially released stable version 2.0 of the HRdm software, which includes several key improvements to largely reduce or eliminate artifacts compared with the initial report of HRdm in 2020[45]. Taken together, we are confident that lipid species identified by HRdm from complex biological samples could largely reflect true lipid signals rather than artificial peaks.

To provide reasonable comparison and demonstrate the advantages of HRdm strategy, we also evaluated other types of commonly used high-end IM paradigms, including TIMS and Traveling Wave IMS (TWIMS)-based cyclic IM (cIM), for large-scale GP profiling at fatty acyl *sn*-position level. The IM-MS data acquisition parameters for TIMS and cIM-MS were described in the Supplementary Information. As shown in Supplementary Fig. 8, in TIMS, *sn*-isomers could not be separated when we used the default 4D lipidomics instrument parameters with 100 ms ramping time[35]. It required a prolonged ramping time (up to 1000 ms) to achieve a slight separation. The sensitivity remarkably decreased as only ~10% ion intensities were preserved with the long ramping time. In cIM-MS (Supplementary Fig. 9), an increase in a number of cyclic passes of up to 20 passes (around 400 ms arrival time) was needed to achieve a similar degree of separation as that shown in HRdm. It is important to note that this approach is more suitable for targeted analysis as it requires a carefully calculated narrow IM selection window to prevent "wrap-around" in the cIM where high-mobility ions catch up with low-mobility ions in multipass experiments. The ion intensity was also significantly decreased to ~10% when increasing the number of cyclic passes to 20. These compromises, including prolonged drift/ramping time and/or narrowed IM selection window, would come at cost of sacrificing either sensitivity, throughput, or IM coverage for large-scale untargeted lipidome profiling[32]. In contrast, this DTIMS-based HRdm strategy demonstrated unique advantages for large-scale lipidome profiling in LC-IM-MS workflows as enhanced IM resolving power and sensitivity could still be achieved in a typical 60 ms IM scanning window for comprehensive ion collection. In addition, DTIMS provides reproducible first-principle CCS measurement, and $^{DT}CCS_{N2}$ measurement is considered the gold standard and is most widely reported in data repositories like PubChem, LIPIDM-PAS, and MSDIAL[36,40,53]. For these reasons, DTIMS-based HRdm IMS is currently the most suitable paradigm for *sn*-position-resolved lipidomic analysis and CCS database construction.

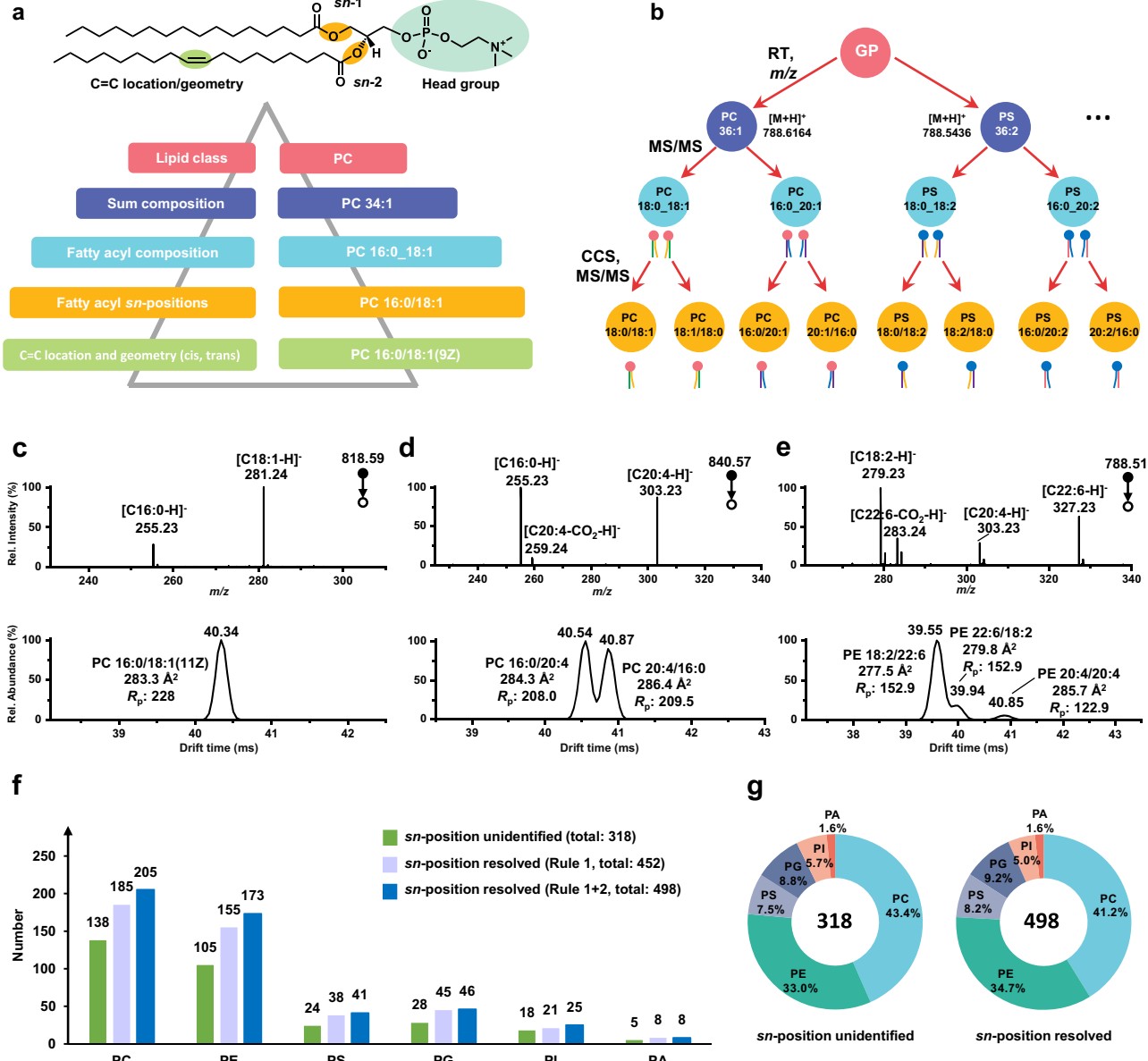

**Fig. 2 | Comprehensive GP analysis at *sn*-position resolved level. a** Hierarchy of GP identification and characterization using the identification of PC 16:0/18:1(9Z) at each level as an example. **b** Illustrating the identification of isobaric GPs detected at nominal *m/z* = 788 as an example with required molecular information for each level of identification. **c** MS2 and drift spectra of PC 16:0/18:1(11Z). **d** MS2 and drift spectra of PC 16:0_20:4. **e** MS2 and drift spectra of PE 40:8. The number (**f**) and proportion (**g**) of GP molecular species identified by LC–MS/MS (IM off) and LC–HRdm IM–MS/MS. Source data are provided as a Source Data file.

## Constructing LC-HRdm IM-MS-based experimental four-dimensional library for *sn*-position resolved GPs

Degree of GP structural identity can be categorized into five levels: lipid class, sum composition, fatty acyl composition, fatty acyl *sn*-positions, and C=C location/geometry (*cis/trans*) in the unsaturated fatty acyls[15] (Fig. 2a). Separator "_" means unspecified *sn*-position, and "/" means confirmed *sn*-position for acyl/alkyl constituents. As current routine LC-MS-based lipidomic analysis often identifies only lipid fatty acyl composition, the elucidation of fatty acyl *sn*-positon through HRdm IM pushes the lipidomic analysis to the next stage. After examining multiple GP standards, including PC 16:0/18:1(9Z) *vs* PC 16:0/18:1(11Z) and PC 18:1(9Z)/18:1(9Z) *vs* PC 18:1(11Z)/18:1(11Z) (Supplementary Fig. 10), we find such species usually exhibit a CCS or drift time difference of less than 0.2% and require IM $R_p$ ~ 1000 for baseline separation. As the relatively small CCS differences among C=C bond position isomers are negligible compared to *sn*-position isomers (~1%),

the differentiation of GP *sn*-position isomers is not compromised. In most mammals, as fatty acyl composition of GPs in *trans* C=C configuration is rare, only *cis* C=C configuration is considered in this study[1]. Knowing this, four-dimensional analysis of GPs obtained from LC-HRdm IM-MS (RT, CCS, precursor *m/z*, and MS/MS spectrum) provides an avenue towards unambiguous GP identification and structural characterization. For example, as shown in Fig. 2b, a representative GP precursor with a nominal *m/z* of 788 could be eventually classified into 8 species with *sn*-position information, twice the number of species as that may be resolved on high-resolution MS alone. However, though automatic *sn*-isomer-resolved GP identification may be facilitated through a 4D CCS library, de novo construction of such a library is too costly and not feasible given the lack of suitable standards. To remedy this shortcoming, we extracted GPs from mouse brain – a biological source known to be rich in GPs and *sn*-isomers for library construction[54].

To determine universal rules for assigning GP *sn*-position isomers based on MS/MS fragmentation patterns and CCS differences, we tested all 6 GP classes, each class containing at least 2 standards with different degrees of unsaturation together with the GPs from pooled mouse brain samples (Supplementary Table 1). The 1st rule was summarized from MS/MS fragmentation patterns; the fatty acyl chain fragment ions with higher intensities are at the *sn*−2 position in negative mode, as shown in Fig. 2c using PC 16:0/18:1(11Z) standard as an example. Many studies have also demonstrated that the peak intensity of the carboxylate anion from the *sn*−2 chain is approximately 3 times the intensity of the *sn*−1 chain of GPs due to sterically favorable release of the fatty acyl chain from the *sn*−2 position in negative ion mode[55]. Although the assignment of fatty acyl *sn*-positions could be achieved by examining fragmentation patterns, there are instances where the assignment is obfuscated through similar abundances of GP *sn*-position isomers. Furthermore, if the *sn*−2 chain is polyunsaturated, the intensity ratio could be compromised due to the partial loss of $CO_2$ during fragmentation. The loss of $CO_2$ is observed in negative-ion mode fragmentation but is not observed when using metal-adducts in positive-ion mode[27]. These commonly occurring cases may require further evidence for more confident *sn*-position assignment. Although most of GP *sn*-isomers could be assigned by matching the estimated abundance of *sn*-position isomers from the carboxylate anion intensity in MS/MS spectra and abundance of IM-resolved peaks (1st rule), there were circumstances when the two GP *sn*-isomers were of similar abundance. Based on the CCS values from GP standards and GPs successfully assigned through the fragmentation rule, we also concluded the 2nd rule to be that GPs with smaller fatty acyl chains at *sn*−1 position generally showed lower CCS values than their *sn*-position isomers, which was also reported previously[52]. For example, PE 16:0/18:1 and PS 16:0/18:1 showed smaller CCS values than their *sn*-position isomers, PE 18:1/16:0 and PS 18:1/16:0, respectively (Fig. 1e, g). As this prevailing trend has not been widely reported due to the limitations in IM $R_p$, we reasoned that this finding could be attributed to the higher gas-phase flexibility and freedom of the fatty acyl chain at the *sn*−1 located at the terminus of the glycerol backbone. We also performed in silico simulations of several pairs of representative GP *sn*-isomers in the gas phase to prove this finding[56]. The CCS values of the simulated GPs were consistent with our experimental measurement, both obeying the 2nd rule (Supplementary Fig. 11). Their coordination information (Z-matrix) is also included in Supplementary Data 1. With that, as shown in Fig. 2d, the similar intensities of C16:0 and C20:4 in the MS/MS spectrum of PC 36:4 existed together with the peak of C20:4 with $CO_2$ loss. As the fragmentation pattern was not sufficient to assign the position to the 2 peaks shown in the drift spectrum, the 2nd rule was applied to determine that the peak at 40.54 ms was PC 16:0/20:4 and the peak at 40.87 ms was PC 20:4/16:0, respectively. In terms of lyso-GPs, we found, from commercial standards, that lyso-GPs with acyl chains at *sn*−1 position generally showed smaller CCS values (Supplementary Fig. 12), which was consistent with previous report[45].

Combining both rules, identification of GPs at the *sn*-position level could be achieved within a complex mixture of isomers beyond solely *sn*-isomers. Taking PE 40:8 as an example (Fig. 2e), the fatty acyl composition could be concluded as PE 18:2_22:6 and PE 20:4_20:4 from the three carboxylate anions detected in the MS/MS spectrum. From the high intensities of C18:2 and C22:6 carboxylate anions at *m/z* 279.23 and 327.23 respectively, PE 18:2_22:6 could be concluded as the main constituents. Drift times of the two most abundant peaks, 39.55 ms and 39.94 ms, could be assigned to PE 18:2/22:6 and PE 22:6/18:2 according to the 2nd rule. The remaining peak at 40.85 ms could be subsequently annotated as PE 20:4/20:4.

In the negative MS/MS spectra of this study, we found that fatty acyl composition of most GPs in mouse brain is not too complicated and almost all isomers at fatty acyl composition level are no more than 2. This is also reported by other literature[43,53,54]. In this way, there could

be no more than 4 isomers if we take *sn*-positional isomers into account. That is the main reason that we annotated no more than 4 isomers from HRdm spectra. According to these negative MS/MS spectra and drift spectra, the annotation workflow was illustrated in Supplementary Fig. 13. Additionally, a series of GPs from mouse brain extracts were also used to comprehensively evaluate the consistency of drift spectra and negative MS/MS spectra from the complex biological matrix (Supplementary Fig. 14). Specifically, to enhance the identification accuracy and confidence, we set the minimum peak height at 3000 counts in MS1 as a criterion for identification. Together with fatty acyl composition information of lipid isomers from DDA spectra, we follow the decision tree to conduct the annotation: If HRdm spectra show the same number of peaks as the number of compositional isomers, when GP contains one fatty acyl composition, it indicates *sn*-position isomers does not exist. We can assign the *sn*-connectivity of the fatty acyl chains by rule 1 (Supplementary Fig. 14a–c). For instance, as shown in Supplementary Fig. 14a, there was only a single peak in the HRdm drift spectrum of corresponding to the *m/z* of PC 44:10. The only two fatty acyl fragment ions, corresponding to C22:6 and C22:4 at a ratio of C22:4/C22:6 ≈ 3, indicated single fatty acyl composition (PC 22:6_22:4), and the connectivity could be further assigned as PC 22:6/22:4. When GP contains two fatty acyl compositions, it would be annotated as two *sn*-resolved fatty acyl compositions or two *sn*-isomers of major composition according to rule 1 and rule 2. For instance, as shown in Supplementary Fig. 14h, from the MS/MS spectra, PE 16:0_18:1 was the predominant fatty acyl chain composition, and the very low abundance of PE 16:1_18:0 indicated it would not be the second relatively high peak in drift spectrum. The intensity ratio of fragment ions from PE 16:0_18:1 accords with their abundance in drift spectrum, the connectivity could be further assigned as PE 16:0/18:1 and PE 18:1/16:0. It was rarely observed that the number of IM peaks less than fatty acyl compositions (means one IM peak, two fatty acyl compositions), if existed, it would be annotated as the major composition with *sn*-position. Another case is that the peak number is higher than the number of lipid isomers at the fatty acyl composition level. If the maximum possible number of *sn*-isomers matches the number of peaks from HRdm spectra (Supplementary Fig. 14 d–g, i, l), we would use rule 1 (and rule 2) to assign the *sn*-resolved GPs. There are also cases where the maximum possible number of *sn*-isomers is higher than the number of peaks from HRdm spectra. For example, two compositional isomers were identified in negative MS/MS spectra, which could lead to four possible *sn*-position resolved isomers, but only three peaks were observed in HRdm spectra (Supplementary Fig. 14j–k). If this happens, we will use rule 1 to pick the top 3 abundant *sn*-position resolved GP isomers. The assignment of the three peaks in HRdm spectra is based on peak intensities and rule 2. In our data, we did not observe the case that the number of peaks in HRdm spectra is higher than the maximum possible number of *sn*-isomers from negative MS/MS spectra after we carefully eliminate artifacts if any exist. In the very rare cases when more than two compositional isomers are identified in DDA spectra, we pick the top 4 abundant *sn*-position resolved GP isomers and the top 4 abundant peaks in HRdm spectra and then assign them based on rule 1 and rule 2. For the very low abundance isomers, although they could be identified in our workflow from MS/MS spectra in negative mode, they were not annotated from the HRdm drift spectra to enhance the identification accuracy and confidence. It is worth noting that this potential limitation also exists in current mainstream label-free lipidomics methods when quantification is the major focus[35,36,43,53,54]. Through comparison with standards, the identification of GPs in mouse brains was further validated (Supplementary Fig. 15). Representative IM-MS heatmaps of GPs *sn*-isomers identified from mouse brains were shown in Supplementary Fig. 16. Hence, the combination of the carboxylate anion intensity in MS/MS spectra together with the abundance and CCS distribution in the IM spectra enables unambiguous GP isomer annotation in the drift

spectrum. Thus, the abundance ratio of isomer pairs could be subsequently concluded.

After analyzing pooled mouse brain lipid extracts by data-dependent LC-MS/MS acquisition using QTOF-only mode without IM measurement, a total of 318 molecular GP species with fatty acyl composition were identified (Supplementary Fig. 17a, Supplementary Data 2) – a number that is similar to the number of brain GPs reported using other methods[54]. When operating on multiplexed IM-QTOF mode with the same LC gradient, integration of data-independent LC-IM-MS/MS and data-dependent LC-MS/MS acquisition (Supplementary Fig. 17b), 498 distinct GP molecular species were identified from the same sample, a ~57% increase in identification numbers as a result of *sn*-position annotation with the 1st and 2nd rules while a total of 452 GP could been annotated only through 1st rule (Fig. 2f and Supplementary Data 3). One of the concerns is that infrequent artifacts might affect the identification accuracy. Notably, the infrequent artifacts could be easily filtered out according to their features and would not affect spectral interpretation[45]. Specifically, the artifacts are usually of low abundance (<10% of the primary signal within each *m/z* extraction window) and are sufficiently distant from the signal region of interest to not affect spectral interpretation. Moreover, IM drift times of genuine ion signal align across isotopic peaks due to the negligible contribution of mass to the ion mobility, whereas the isotopic peaks of artifacts are not aligned. For instance, as shown in Supplementary Fig. 18 a1, a2, the abundance of artifactual peak (at 41.69 ms) in IM-MS heatmap and extracted drift spectrum were much lower than true ions, and the isotopic peaks were not aligned. Thus, artifactual peak (at 41.69 ms) co-existing with PE 18:0/20:4 (40.38 ms) and PE 20:4/18:0 (40.72 ms) in QC sample were easily distinguished from genuine ions due to their long distance from the true ions, their low abundance, and non-aligned isotope peaks. The existence of PE 18:0/20:4 (40.38 ms) and PE 20:4/18:0 (40.72 ms) in QC sample also could be validated with the negative MS2 spectrum (Supplementary Fig. 18a3). On the one hand, the low possibility of the artificial peaks could be easily discriminated from our genuine ions and would not affect the identification accuracy. On the other hand, we also validated the drift spectra of lipid extracts from QC samples with that from most of investigated biological samples, the artificial peak, which only randomly appeared in certain spectra but not all spectra, could be distinguished from consistently appeared genuine peaks. For instance, as shown in Supplementary Fig. 18, one peak in drift spectrum of PE 18:0_20:4 happened to exist in the QC sample at a very low abundance and with an isotope of inconsistent drift time (Supplementary Fig. 18 a1&a2), but did not appear in other samples. The two major drift peaks aligned well across all drift spectra and their isotopes, indicating they were genuine peaks, PE 18:0/20:4 and PE 20:4/18:0. The very small peak (at 41.69 ms), with not aligned isotopic peaks, and far from the major peaks in the drift spectrum of QC sample, was the artificial peak. Therefore, checking drift time alignment of isotopic peaks and validation across replicates are recommended to determine the genuine peaks when using HRdm to acquire high-resolution IM data for unknown biomolecules and new database. From the 318 extracted drift spectra of 498 identified lipids in pooled mouse brain samples, artificial peaks were only observed in 15 extracted drift spectra, indicating that less than 5% of lipid ions co-existed with artifacts. Drift times of all artificial peaks from pooled mouse brain samples were labeled and supplemented in the notes of Supplementary Data 3. Herein, the low ratio of the artificial peaks could be easily discriminated from our genuine ions and would not affect the identification accuracy.

To clearly highlight the obvious improvement in identification number benefits from high-resolution IM, a comparison of identification numbers from different 4D lipidomics strategies has been conducted. we performed the analysis of the lipid extract from mouse brain using 4D library-based match and rule-based refinement without additional HRdm strategy as Zhu and co-workers reported[43]. GPs were determined as one major component of *sn*-position isomers if the intensity ratio of *sn*−1/*sn*−2 was <0.9, and the existence of their corresponding *sn*-isomers was ignored. As shown in Supplementary Fig. 19, a total of 326 GPs including 241 GPs at *sn*-resolved level and 85 GPs at fatty acyl level in mouse brain were identified. The identification result is comparable with studies that annotation of GPs at fatty acyl level using LC-MS/MS[53]. In our study, using the 4D lipidomics with HRdm, a total of 498 lipid species have been identified with additional *sn*-position isomers from *sn*-position isomer pairs.

To reduce the incidence of false positive identifications due to potential artificial peaks, the annotated GPs were manually assessed to ensure that their CCS values remained consistent across all replicate measurements. The robustness and reliability of our presented methodology were demonstrated by the precision and consistency of our measurements. Specifically, we observed an average MS error of 0.2 ppm. The average coefficients of variation (CV) of the RT was 2.1%, and the average CV of CCS was 0.1% (Supplementary Data 3). Proportions of each GP class were summarized in Fig. 2g. All six identified GP classes demonstrated an increase in identification numbers, owing to the significant portion of GPs containing *sn*-position isomers. As shown, the improved sensitivity and resolution of HRdm enabled the construction of a large and in-depth 4D GP library (Supplementary Data 4) that may be employed for automated data analysis. Compared to GP identification by only MS/MS in negative mode, although HRdm method has potential limitations in annotating very low abundance isomers when, in rare cases, multiple fatty acyl compositional isomers co-eluting, it increased annotation of high abundance fatty acyl composition isomers with additional *sn*-position information, expanding the quantifiable GP isomers. To note, current routine untargeted LC-MS/MS or LC-IM-MS/MS lipidomics methods only enable quantification at the level of GP sum composition or for the predominant composition among co-eluting GPs[40,43,53]. In contrast, benefiting from high-resolution IM, separating GP *sn*-position isomers in the drift spectra enables the unequivocal quantification of both *sn*-position isomers.

## Machine learning-enabled CCS and RT prediction for *sn*-position-resolved GP identification

As demonstrated, coupling LC with a high-resolution IM-MS/MS platform allows for comprehensive, in-depth GP profiling. However, to facilitate the automated identification of *sn*-position-resolved GPs beyond those GPs from our manually validated 4D GP library, it is necessary to obtain CCS values accurate enough to differentiate *sn*-position isomers. Broadly speaking, CCS measurement of GP isomers is found to be severely lacking as few GP-rich biological sources are known and commercial standards are often limited, indicating that greater analytical understanding of GPs is challenging yet in demand. To address these limitations, we implemented a machine learning-based workflow, already validated in previous studies[40,41], for large-scale CCS prediction of GPs. This allowed us to expand on the experimental data generated from our analyses of brain lipid extracts (Fig. 3a).

In the prediction workflow (Fig. 3a), the chemical structure of each GP was first converted to a simplified molecular-input line-entry specification (SMILES) format, from which molecular descriptors (MDs) of the molecule were calculated. MDs can be defined as mathematical representations of the molecule. By converting the chemical structures into numeric values of various MDs, they are used to quantitatively describe the physical and chemical information of molecules[57]. Prediction was performed on singly protonated PC, PE, and PS to reduce systematic errors, largely due to insufficient experimental data input from PG, PI, PA, and their preference towards ammonia adduct formation (see Methods).

The selection of appropriate MDs to effectively distinguish *sn*-position isomers in the prediction model was of vital importance and yet presented a significant challenge due to the subtle differences in GP

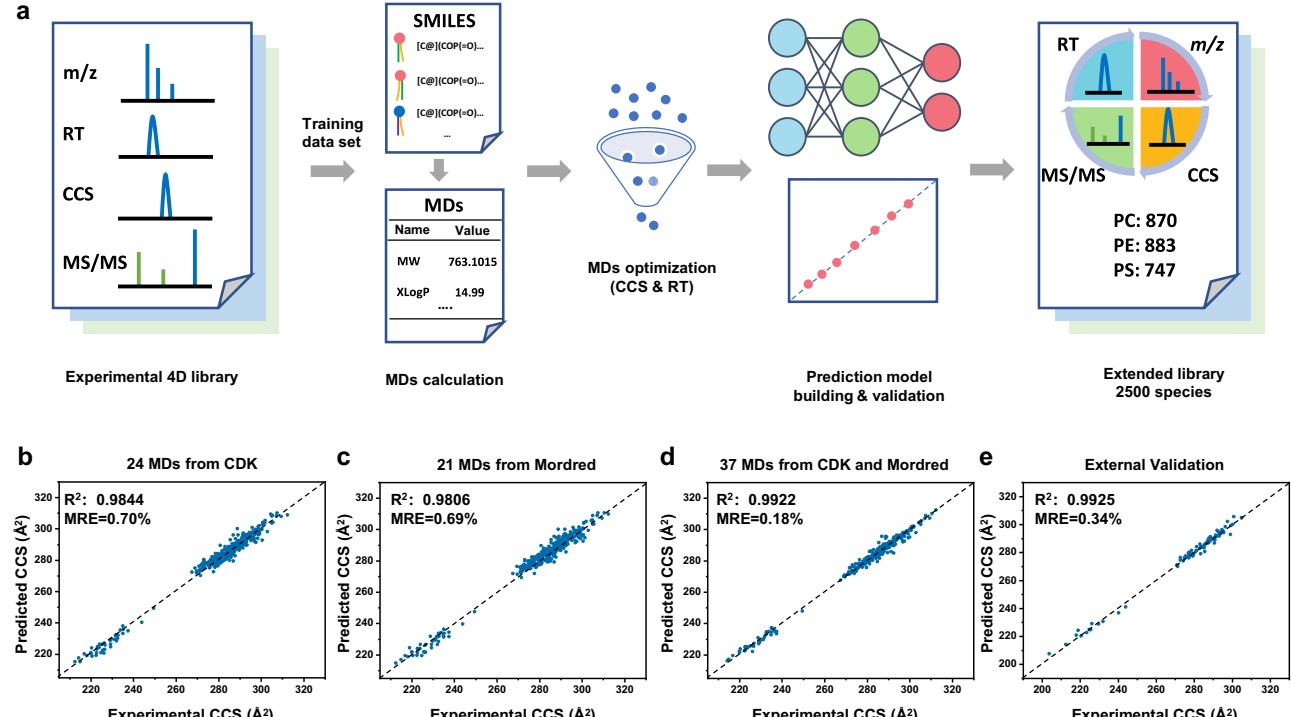

**Fig. 3 | Machine learning-enabled CCS prediction for *sn*-position-resolved GPs.**
**a** Schematic illustration of the workflow to curate the extended GP CCS library.
The internal validation of the machine learning algorithm predicted CCS values
of GPs (*n* = 335) using optimized MDs selected from CDK (**b**), Mordred (**c**), and
pool of the whole CDK and part of Mordred relevant to GP *sn*-position (**d**).
**e** The prediction performance evaluated by external validation (*n* = 84).
Source data are provided as a Source Data file.

structures and physicochemical properties. In our first trial for prediction model construction, MDs were calculated by the most widely used "rcdk" package (CDK) for lipid CCS prediction[40,41]. Using the least absolute shrinkage and selection operator (LASSO) algorithm (see Methods), 24 MDs out of the 221 MDs from CDK were selected (Supplementary Table 4). With these 24 MDs, CCS prediction was carried out using an SVR-based machine learning algorithm, giving a linear fit with $R^2 = 0.9844$ for the training set (Fig. 3b, Supplementary Data 5). While the high regression fit of our model is competitive when benchmarked against previous studies, higher prediction accuracy would still be needed for more confident GP *sn*-position assignment. As such, we also considered the 1824 two- and three-dimensional MDs calculated via Mordred for testing and optimization[57]. Again, using LASSO, 21 MDs from Mordred (Supplementary Table 5) were used within the machine learning prediction model and gave a linear fit with $R^2 = 0.9806$ for the training set (Fig. 3c and Supplementary Data 5). We reason that the unsatisfactory fits from both sets of MDs were due to the poor differentiation ability of *sn*-position isomers in the prediction models - only 15 out of 221 MDs from CDK and 500 out of 1824 MDs from Mordred software showed differences between *sn*-position isomers.

To further improve the accuracy and precision of the prediction model and reinforce the influence of *sn*-position on the entire model, we constructed the prediction model with MDs from both CDK and Mordred. As initial input, a total of 253 MDs were selected. These included 32 MDs from Modred, which demonstrated > 0.1% difference between *sn*-position isomers (Supplementary Table 6), and all 221 MDs from CDK. Thirty-seven final MDs from the 253 MDs were selected using LASSO (Supplementary Table 7) to build an SVR-based prediction model, demonstrating excellent precision with $R^2 = 0.9922$ in the internal validation (Fig. 3d and Supplementary Data 5). The external validation results also showed that the CCS prediction had a good linear fit with $R^2 = 0.9925$ and a median relative error (MRE) of 0.34% (Fig. 3e and Supplementary Data 5).

Beyond CCS prediction, a prediction model for RTs was also built and the predicted RTs were added to the extended library to improve the identification accuracy. Since *sn*-isomers almost have the same RT on a short time C18 gradient, MDs calculated by "rcdk" were used for retention time prediction. Using the LASSO algorithm, 51 MDs out of 221 MDs were selected and using an SVR-based machine learning algorithm to build the prediction model and gave a linear fit with $R^2 = 0.9622$ for the training set (Supplementary Fig. 20 and Supplementary Data 6). Overall, the results demonstrated that the prediction model has an excellent capability for precisely predicting the physicochemical properties of *sn*-position-resolved GPs. Finally, using the optimized prediction model, CCS values and RTs of 2500 GPs with *sn*-position information were predicted – 870 PCs, 883 PEs, and 747 PSs (Supplementary Data 7).

## Spatial diversity of GPs at *sn*-position resolved level in AD mouse brain

AD is a progressive neurodegenerative disorder with histopathological hallmarks of β-amyloid (Aβ) plaques and neurofibrillary tangles in the brain. Extensive studies have demonstrated that altered GP compositions and enzyme activities involved in the GP remodeling in the brain were associated with AD progression[9,11,12]. Although many studies on GPs in the AD mouse brain have been reported[11,54,58], the detailed spatial distribution of GP *sn*-position across different brain regions and their functions remains unexplored. In this study, we used APPswe/PS1dE9 transgenic mice to investigate GP alteration in AD progression. The animals express mutant forms of APP (Mo/HuAPPswe with the K595N/M596L mutation) and PS1 (deletion of exon 9). They are characterized by an early-onset of AD and age-associated increase in Aβ-levels followed by Aβ deposition, morphological alterations, and are also widely used to investigate lipid alterations in AD[5,59]. Extensive studies in patients with advanced AD have demonstrated that the hippocampus and cortex are more heavily affected by Aβ pathology

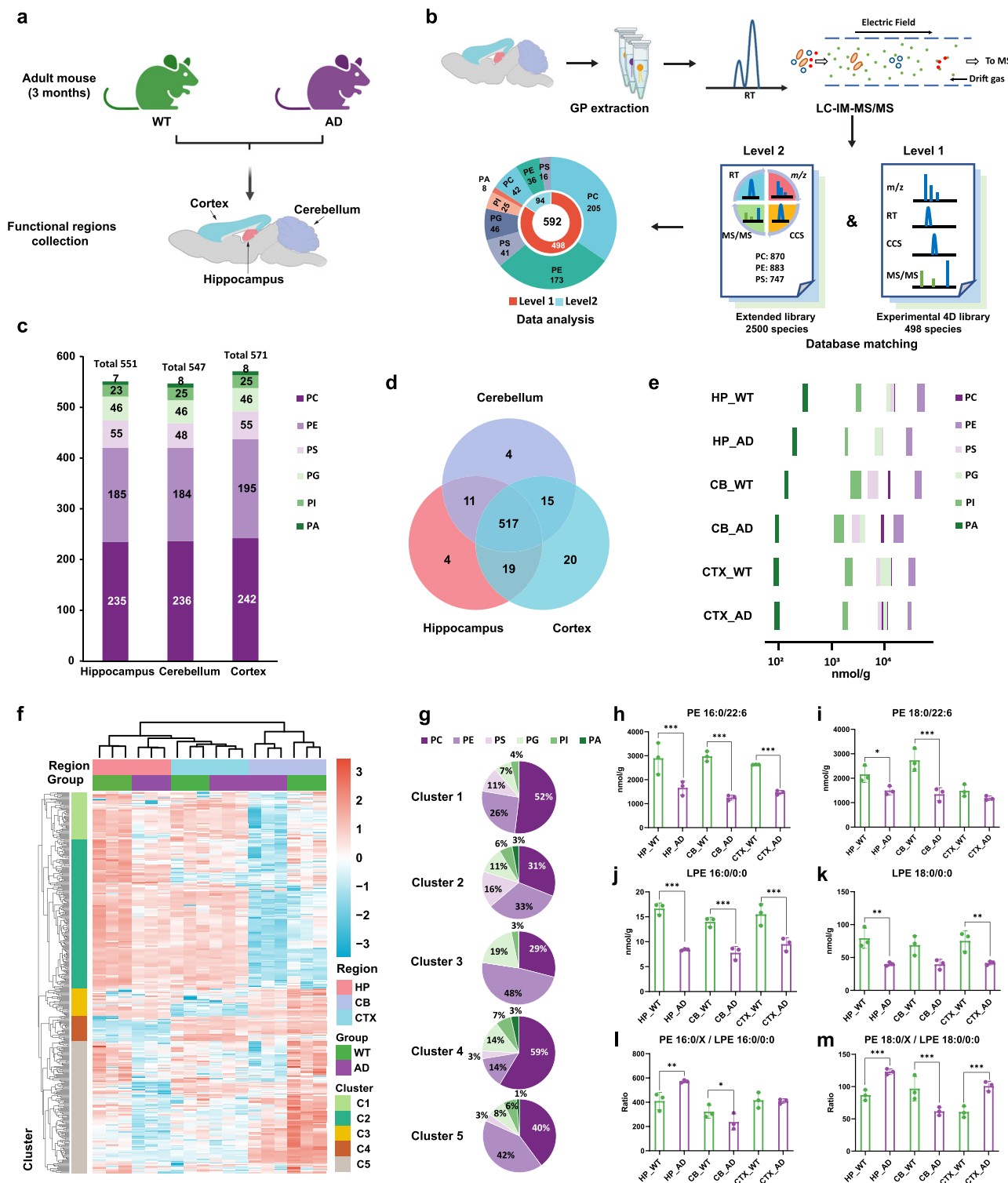

than the cerebellum[60]. Therefore, hippocampus, cortex, and cerebellum were chosen for our examination. Following extraction of GPs from both 3-month-old wild type (WT) and APPswe/PS1dE9 mice (simply referred to as AD mice thereafter) (Fig. 4a), we performed quantification analysis of GPs using the validated LC-HRdm IM-MS method in tandem with our experimental 4D library and extended CCS library (Fig. 4b; detailed schematic is shown in Supplementary Fig. 17). A total of 592 GPs were identified across the 3 regions, including 130 pairs of *sn*-position isomers (Supplementary Data 8). Specifically, 498 GPs identified by our experimental 4D library were classified as level 1

annotations in this context, and the 94 GPs identified from matching with the extended CCS library were classified as level 2 annotations in this context. The cortex was found to contain the greatest diversity of GPs with 571 GPs identified, while 551 and 547 GP species were identified in the hippocampus and cerebellum, respectively (Fig. 4c, Supplementary Data 8). Similar findings regarding the number of identified lipid species in the hippocampus and cerebellum have been reported in previous studies[54]. In terms of the number of species identified in each GP class, PC and PE were dominant across all 3 regions, consistent with findings from previous studies[11,58]. Further

**Fig. 4 | Spatial characterization of GPs in the AD and WT mouse brain.**
**a** Illustration of three functional regions (hippocampus (HP), cerebellum (CB), and cortex (CTX)) of mouse brains (age of 3 months; $n = 3$) selected for spatial characterization of GPs between WT and AD groups. Figure 4a was created with BioRender.com, released under a Creative Commons Attribution-NonCommercial-NoDerivs 4.0 International license. **b** Graphic illustration of the workflow for GP identification from mouse brain. Figure 4b was created with BioRender.com, released under a Creative Commons Attribution-NonCommercial-NoDerivs 4.0 International license **c** Numbers of GPs identified in the three brain regions. **d** Overlap of GPs identified from 3 brain regions. **e** Total abundance of 6 GP classes quantified in all brain regions of WT and AD groups. The GP abundance was normalized to wet tissue weight (mg). **f** HCA discrimination of the AD and WT group of 3 mouse brain functional regions by quantitative results of GPs at *sn*-position resolved level. **g** Percentages of GP classes in clusters 1–5. Abundance distributions of representative GPs in AD and WT mouse brain regions, PE 16:0/22:6 ($p = 0.0004$ for HP, 0.0001 for CB, and 0.0005 for CTX) (**h**), PE 18:0/22:6 ($p = 0.0140$ for HP, 0.0003 for CB) (**i**), LPE 16:0/0:0 ($p = 1.4E-5$ for HP, 7.0E-5 for CB, and 8.5E-5 for CTX) (**j**), LPE 18:0/0:0 ($p = 0.0030$ for HP, 0.0035 for CTX) (**k**). The ratio of *sn*-1 C16:0-containing PE/LPE 16:0/0:0 ($p = 0.0031$ for HP, 0.0472 for CB) (**l**) and *sn*-1 C18:0-containig PE/LPE 18:0/0:0 ($p = 0.0007$ for HP, 0.0007 for CB, and 0.0006 for CTX) (**m**) in AD and WT mouse brain regions. Data are presented as mean values +/−SD ($n = 3$, *$p < 0.05$, **$p < 0.01$, ***$p < 0.001$ (one-way ANOVA with correction for multiple comparisons using the two-stage linear step-up procedure of Benjamin, Krieger, and Yekutieli at a 0.05 FDR)). Source data are provided as a Source Data file.

cross-regional analysis revealed the diversity of GP distributions among different brain regions with a total of 517 GPs identified in all three regions, though each region did reveal unique species (Fig. 4d).

For the abundance distribution of GPs in each brain region, as shown in Fig. 4e, the total abundance of each GP class showed vast differences, ranging across 3 orders of magnitude in 3 brain regions. PE was the GP class of the highest abundance within each region with a minimum abundance of $29.30 \pm 2.86$ μmol/g in the AD mouse cortex and a maximum value of $49.12 \pm 7.57$ μmol/g in the WT mouse hippocampus. In contrast, PA was the GP class that demonstrated the lowest abundance in each region with a minimum abundance of $92.25 \pm 10.13$ nmol/g in the WT cortex and a maximum value of $324.70 \pm 36.22$ nmol/g in the WT hippocampus. It is worth noting that a decreased abundance of all GPs was observed for AD groups compared to WT groups in each of the 3 brain regions, indicating the disruption of GP homeostasis of AD which has also been reported by other studies[58,61].

To comprehensively understand the AD-induced alteration of GPs within 3 distinct brain regions, abundances of *sn*-position-resolved GPs were used to perform hierarchical clustering analysis (HCA) (Fig. 4f), which successfully discriminated AD and WT groups within all three regions. The successful clustering of the three brain regions also revealed *sn*-position-defined GPs had spatial heterogeneity, suggesting their distinct functions. Within HCA, GPs could be grouped into five clusters (Fig. 4f, g, and Supplementary Data 9), among which Cluster 2 and Cluster 5 contained the greatest number of GP species across all six GP classes. In terms of GP numbers, PE class is the most dominant in both of these two clusters (Fig. 4g). When comparing proportion of molecular abundance to number of GP species in each GP class, an orthogonal perspective emerges on the percentage in each cluster (Supplementary Fig. 21). For instance, PC is dominant in terms of GP numbers (>50%) in Cluster 1, yet it is minor (~30%) in terms of molecule percentage. GPs in Cluster 2 were expressed at the highest levels in the WT hippocampus and the lowest level in the AD cerebellum. GPs in Cluster 5 exhibited very high levels in the WT cerebellum and the lowest levels in the AD hippocampus. Quantitative analysis across three regions revealed a marked disparity in the abundance of GPs in the hippocampus and cerebellum. Moreover, the observed changes in the abundance of individual GPs between AD and WT mice across the three regions indicate that GP profiling could serve as a sensitive marker for AD.

The altered abundance of multiple prominent GPs demonstrated differences between WT and AD mice in the three brain regions, especially polyunsaturated fatty acids (PUFA)-containing GPs (Supplementary Data 10). Previous studies on fatty acid composition demonstrated a progressive decline of the omega-3 fatty acid, especially docosahexaenoic acid (DHA, C22:6), in late-stage AD brains[11,62]. PE constitutes the major storage form of DHA in the brain and can facilitate the signal transduction of bioactive mediators[9,63]. It has been reported that DHA-containing PE could protect PUFAs from oxidation in gray matter[11,62–65]. DHA-containing PE, especially in the forms of PE 16:0/22:6 and PE 18:0/22:6, were also detected in high abundance in all three investigated brain regions. The abundance of PE 16:0/22:6 and PE 18:0/22:6 was significantly lower in AD compared to WT mice across all three regions except for PE 18:0/22:6 displaying no obvious difference in the cortex (Fig. 4h, i).

Previous studies have reported a significant difference in the ratio of GP/lyso-GP between AD patients and healthy controls, with an accuracy of 82%-85% using this ratio as an additional biomarker for neuropathological diagnosis of AD[6]. However, GP molecules investigated in those prior studies were not annotated with *sn*-position specificity, the limitation directly addressed in our investigations. LPE 16:0/0:0 and LPE 18:0/0:0 were taken as examples because GPs containing C16:0 and C18:0 fatty acyl at *sn*-1 are the most abundant. We observed that LPE 16:0/0:0 and LPE 18:0/0:0 decreased in AD compared to WT mice in all three regions (Fig. 4j, k). We further evaluated the ratio of *sn*-1 C16:0-containing PE/LPE 16:0/0:0 and *sn*-1 C18:0-containing PE/LPE 18:0/0:0 in all regions (Fig. 4l, m). Our data revealed the ratio of *sn*-1 16:0-containing PE/LPE 16:0/0:0 was only significantly elevated in the hippocampus, and the ratio of *sn*-1 C18:0-containing PE/LPE 18:0/0:0 was likewise increased in hippocampus and cortex. The increased levels of PE/LPE suggested that dysregulated lipid metabolism occurred in the AD mouse brain. Furthermore, the heterogeneous alterations indicated that enzymatic activity involved GP metabolism may not be conserved across all brain regions. We also observed spatial alterations of GP *sn*-position isomers between WT and AD brain regions (Supplementary Fig. 22). As a representative example, PS 18:1/20:4 showed similar abundance in WT and AD across 3 regions. However, PS 20:4/18:1 in hippocampus, but not the cortex and cerebellum, showed a significant decrease in AD compared to WT.

## Age-associated temporal GP alterations in AD mouse brain

As AD is commonly considered as an age-related neurodegenerative disease, unraveling the potential molecular mechanisms of AD progression is of vital importance[11]. To better understand GP alteration in AD progression, we analyzed the GPs in the same three brain regions of the mice at the age of 3 and 8 months (Fig. 5a, Supplementary Fig. 23). Detailed analysis of GPs in the hippocampus is presented here as an example. Based on the abundance of *sn*-position-resolved GPs in the hippocampus, all samples were correctly classified by HCA into 4 groups, and each correctly matched on age and genotype (Fig. 5b and Supplementary Data 11). At the outset, these results indicated that the abundance of *sn*-position resolved GPs could sufficiently reflect neurodegeneration and could be used to classify normal and diseased samples. In particular, the clusters representing WT at 8 months and AD at 3 months were clustered adjacent to one another, further highlighting the age-related nature of AD. Our results demonstrate that mice of different ages and genotypes can be distinctly differentiated based on quantified GPs in the hippocampus (Fig. 5b, Supplementary Fig. 23a) and cortex (Supplementary Fig. 23b). However, the PCA of GPs in the cerebellum (Supplementary Fig. 23c) showed overlap between WT mice at 8 months and AD mice at 3 months, indicating that GP profiles in the hippocampus and cortex are more

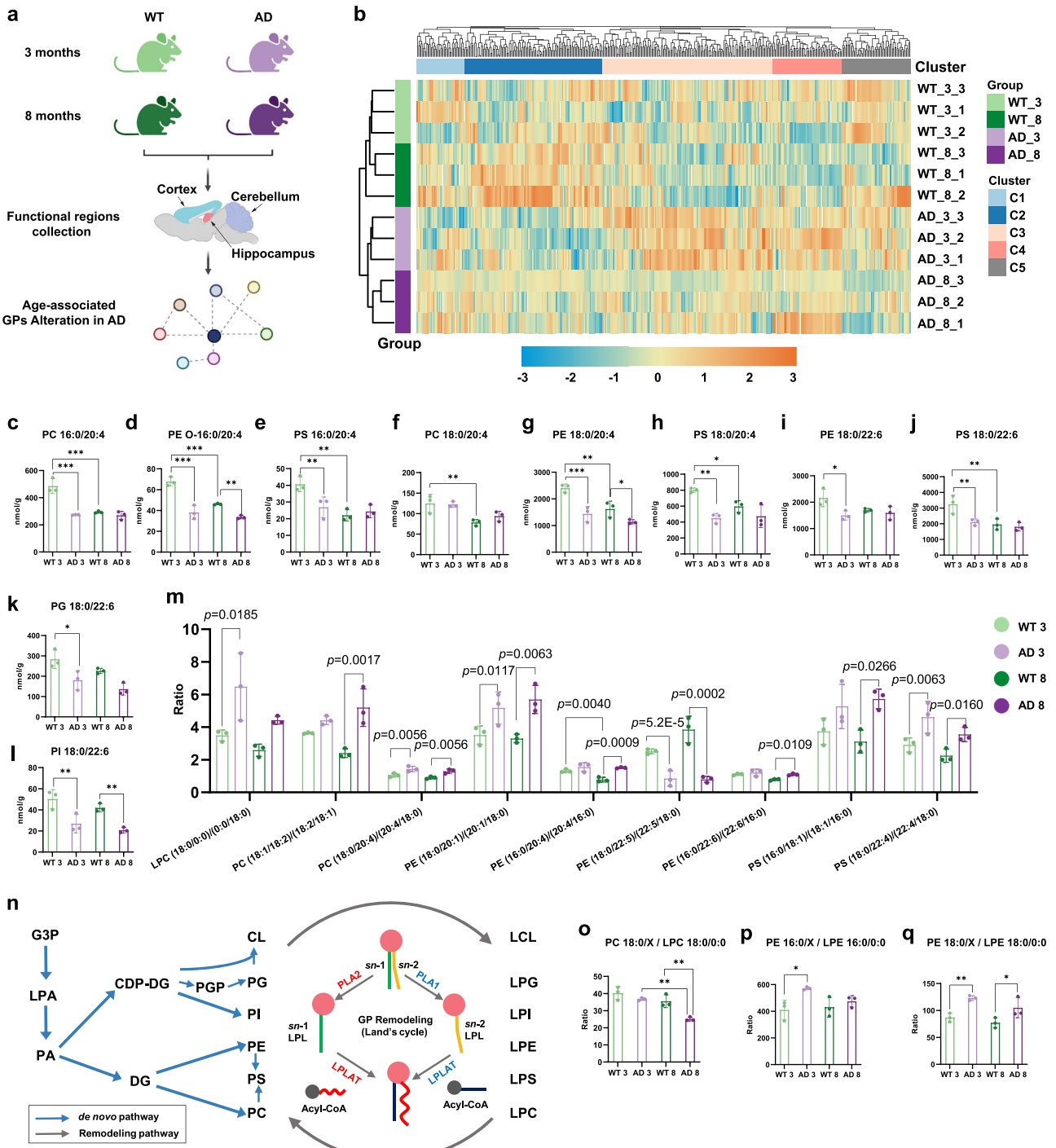

**Fig. 5 | Age-associated temporal diversity of GPs in the mouse brain.**
**a** Illustration of three functional regions of mouse brains (age of 3 months and 8 months; $n = 3$) selected for investigation of temporal diversity of GPs in aging and AD progression. Figure 5a was created with BioRender.com, released under a Creative Commons Attribution-NonCommercial-NoDerivs 4.0 International license. **b** HCA of GPs in the hippocampus. Compositional variations of GPs with C20:4 at *sn*-2 position (**c–h**), PC 16:0/20:4 ($p$ (WT 3 vs. AD 3) = 5.6E-5, $p$ (WT 3 vs. WT 8) = 5.6E-5) (**c**), PE O-16:0/20:4 ($p$ (WT 3 vs. AD 3) = 1.9E-5, $p$ (WT 3 vs. WT 8) = 1.2E-4, and $p$ (WT 8 vs. AD 8) = 0.0027) (**d**), PS 16:0/20:4 ($p$ (WT 3 vs. AD 3) = 0.0068, $p$ (WT 3 vs. WT 8) = 0.0036) (**e**), PC 18:0/20:4 ($p$ (WT 3 vs. AD 3) = 0.0095) (**f**), PE 18:0/20:4 ($p$ (WT 3 vs. AD 3) = 0.0008, $p$ (WT 3 vs. WT 8) = 0.0018, and $p$ (WT 3 vs. WT 8) = 0.0157) (**g**), PS 18:0/20:4 ($p$ (WT 3 vs. AD 3) = 0.0026, $p$ (WT 3 vs. WT 8) = 0.0211) (**h**). Compositional variations of GPs with DHA at *sn*-2 position (**i–l**) among the AD and WT mouse at 3 and 8-month age in hippocampus, PE 18:0/22:6 ($p$ (WT 3 vs. AD 3) = 0.0330) (**i**), PS 18:0/22:6 ($p$ (WT 3 vs. AD 3) = 0.0067, $p$ (WT 3 vs. WT 8) = 0.0051) (**j**), PG 18:0/22:6 ($p$ (WT 3 vs. AD 3) = 0.0133) (**k**), PI 18:0/22:6 ($p$ (WT 3 vs. AD 3) = 0.0027, $p$ (WT 3 vs. WT 8) = 0.0032) (**l**). **m** The *sn*-isomer ratios for representative GPs among the AD and WT mice at 3 and 8 months ($p$ values are shown in graphs). **n** Biosynthetic pathways of glycerophospholipids, including de novo synthesis and the fatty acyl remodeling of GPs. The ratio variations of representative GP/lyso-GP among the AD and WT mouse at 3 and 8-month age in hippocampus, PC 18:0/X / LPC 18:0/0:0 ($p$ (WT 3 vs. AD 3) = 0.0014, $p$ (WT 3 vs. WT 8) = 0.0018) (**o**), PE 16:0/X / LPE 16:0/0:0 ($p$ (WT 3 vs. AD 3) = 0.0318) (**p**), PE 18:0/X / LPE 18:0/0:0 ($p$ (WT 3 vs. AD 3) = 0.0070, $p$ (WT 3 vs. WT 8) = 0.0176) (**q**). Data are presented as mean values +/−SD ($n = 3$, *$p < 0.05$, **$p < 0.01$, ***$p < 0.001$ (one-way ANOVA with correction for multiple comparisons using the two-stage linear step-up procedure of Benjamin, Krieger, and Yekutieli at a 0.05 FDR). Source data are provided as a Source Data file.

sensitive markers for monitoring age-related AD progression than those in the cerebellum. In order to provide a comprehensive comparison of the GP changes of different genotype and age in three mouse brain regions, we performed volcano plot analysis based on GP abundance to evaluate the alterations (Supplementary Fig. 24, Supplementary Data 12). Substantial numbers of GP species exhibited a significant decrease in AD compared to WT at the age of 3 months in all 3 regions. Among the large number of significantly decreased GPs in AD, the obvious decrease of many PUFA-containing GPs in all 3 regions were noted. A more detailed analysis was also performed to elucidate the age-dependent changes in PUFA-containing GP abundance between WT and AD groups. Notably, compared to the WT 3-month group, most identified GPs with C20:4 and DHA at *sn*-2 decreased in the AD 3-month group and the WT 8-month group (Fig. 5c–l). The abundances of many GPs in AD both at 3 and 8 months were close to WT at the 8 months, indicating that our GP profiling could also reflect the fact that AD is an age-related disease. These findings still hold true in the cerebellum and cortex (Supplementary Figs. 25–28). This alteration trend was consistent with previous reports that PUFA compositions were decreased in aging and AD[62].

Subsequently, although most GPs were downregulated in AD groups, including the simultaneous decrease of many pairs of GP isomers, illustrating the differences in their ratio of change would facilitate the more detailed interpolation of the underlying mechanism. In terms of GP *sn*-position isomer ratios, definitive alteration of multiple GPs was observed among the 4 different mouse groups (Fig. 5m), marking the importance of this overlooked field in previous lipidomic studies. At the *sn*-position isomer level, the ratio of some *sn*-isomer pairs also showed significant alterations in AD groups compared to WT groups both at 3 months and 8 months. For example, the ratios of LPC 18:0/0:0 to LPC 0:0/18:0, were significantly elevated in AD mice compared to WT mice at 3 months in hippocampus. A portion of lyso-GP is the degradation product of GP catalyzed by phospholipase A (PLA). The increased ratio of LPC (18:0/0:0)/LPC (0:0/18:0) suggested that the *sn*-2 PUFA-containing GPs were more degraded to produce LPC 18:0/0:0 than their corresponding isomers. Thus, the significant deviation in ratios of *sn*-isomers was potentially correlated with the increased activities of PLA2, which was responsible for cleaving the acyl groups at the *sn*-2 position of GPs and resulting in the production of lyso-GPs and free fatty acids (Fig. 5n)[66]. To validate this finding, we further evaluated the cytosolic phospholipase A2 (cPLA2) activities in WT and AD mice across three brain regions as cPLA2 is one of the major PLA2 in the brain[66,67]. Congruent with the previous reports[67], this assay demonstrated significantly increased cPLA2 activity in AD compared to WT at 3 months in the hippocampus, which was also consistent with the results obtained through our LC-IM-MS/MS strategy (Supplementary Fig. 29). On the contrary, the ratios of some 1,2-diacyl GP isomer pairs, including PC 18:0/20:4, PE 18:0/20:1, PS 18:0/22:4 to their *sn*-isomers, were significantly elevated in 3-month-old AD mice compared to 3-month-old WT mice. The results indicated that, although increased cPLA2 activities in early AD may enhance the degradation of *sn*-2 PUFA-containing 1,2-diacyl GPs, the *sn*-1 PUFA-containing 1,2-diacyl GPs were actually more degraded than their corresponding isomers in early AD. Moreover, most of these ratios also showed significant changes at 8 months. Together, these data suggested that, in the complex lipid metabolism processes of AD and aging mouse brain, the activities of lipid metabolic-related enzymes, besides cPLA2, were likely also altered and might require a more systematic evaluation[68].

In addition, we also mapped altered ratios of GP/lyso-GP for multiple species (Fig. 5o–q). The ratio of *sn*-1 C18:0-containing PC/LPC 18:0/0:0 was significantly decreased in the AD hippocampus at 8 months (Fig. 5o). The results are consistent with a previous study showing that the decreased ratio of PC/lyso-PC in AD might be highly useful as a potential plasma biomarker for the diagnosis of early

dementia[6]. However, the ratio of PE/LPE has rarely been reported. We found the ratios of *sn*-1 C16:0-containing PE/LPE 16:0/0:0 (Fig. 5p), and *sn*-1 C18:0-containing PE/LPE 18:0/0:0 (Fig. 5q) were significantly higher in 3-month AD compared to 3-month WT in the hippocampus. The ratio of *sn*-1 C18:0-containing PE/LPE 18:0/0:0 was also significantly increased in the cortex of 3-month-old WT mice compared to that of 3-month-old AD mice, but this difference was not observed in the cerebellum. However, a significant increase in this ratio could be observed in the cerebellum of older, 8-month-old mice (Supplementary Fig. 26, 28). The cPLA2 activities in WT and AD mice across 3 brain regions were also varied (Supplementary Fig. 29). The results, together with previous studies[67], suggested the potential heterogeneous enzyme activities in different ages and functional brain regions of AD. Taken together, our results detail the spatial and temporal diversity of GPs in the mouse brain and highlight putative molecular signatures of aging and AD progression.

## Discussion

Lipids play important roles in cell signaling, cell structure, and energy storage, however detailed lipid structures remain largely underexplored in routine lipidomics owing to its complexity[13,14]. Over the past decade, significant research efforts have been devoted to developing analytical methods for the detailed structural elucidation of lipids *sn*-position, C=C location and geometry isomers[7,8,13,17–19,26]. Recently, IM-MS has emerged as a promising technology for lipidomics to facilitate the separation and identification of complex mixtures of analytes[30,39]. Studies have demonstrated the power of IM-MS for enhancing identification accuracy of lipidome in complex biological samples[33,36,40]. Moreover, studies have coupled LC-IM-MS/MS analysis and rule-based refinement for improving accuracy of lipid identification and achieving the partial annotation of GPs with the main composition of *sn*-position isomers[37,43]. Furthermore, recent research has highlighted the enormous potential of high-resolution IM-MS for indepth lipid isomer separation[29,45]. Nevertheless, to the best of our knowledge, the use of label free LC-IM-MS/MS for large-scale GP *sn*-isomers identification and quantification analysis in biological samples has not been widely explored. In this work, we develop an LC-HRdm IM-MS/MS-based 4D lipidome profiling method that allows large-scale and *sn*-position-resolved GP identification and quantification in complex biological samples. With advanced multiplexed ion injection, both increased sensitivity and enhanced IM $R_p$ up to 250 have been achieved following HRdm data processing. These improvements enable efficient separation of co-eluting GP *sn*-position isomers (~1% CCS differences) in the IM dimension at large-scale without sacrificing throughput or mobility coverage. Together with acyl chain information obtained from MS/MS analysis, our strategy provides the ability to identify GPs from the fatty acyl composition level (e.g., PC 16:0_18:1) to the *sn*-position level (e.g., PC 16:0/18:1 and/or PC 18:1/16:0). This strategy makes it possible for GP identification through intact analysis without further metal adduction, enabling highly confident quantification.

In addition to developing a strategy for in-depth GP analysis, we isolated GPs from mouse brains and compiled the empirical findings into an *sn*-position-resolved GP 4D database to benefit the broader community with an automated GP analysis pipeline that does not require prior knowledge or tedious manual curation. The 4D database contains a total of 498 GPs with acyl chain *sn*-position information which were identified from the LC-HRdm IM-QTOF platform. These results represent more than 50% increase in the number of GP identification, which is 318 without *sn*-position assignment obtained through LC-QTOF (IM off), and 326 using 4D library-based match and rule-based refinement without additional HRdm strategy. As accurate and reliable CCS values serve as the key for the identification of GP *sn*-position isomers, we used a machine learning-based prediction approach to construct an extended 4D library comprised of 2500 GPs for *sn*-position isomer identification. Our extended 4D library, with the

high prediction accuracy ($R^2 > 0.992$) of the CCS values, is more in-depth than other currently available predicted CCS libraries[36,40] which do not differentiate *sn*-position isomers.

Many studies have demonstrated significant changes of GP in AD and aging mouse brain[11,54,58]. However, these studies only revealed GP alteration at fatty acyl sum composition or fatty acyl level across different brain regions with aging, while the alteration of detailed GPs at *sn*-position resolved level remains unexplored. With both the experimental database and the extended library developed in this study, we investigated the spatial and temporal GP alterations in the brains from a mouse model of AD. Through our analyses, we found that GPs profiling at *sn*-position-resolved level allowed for correct clustering of spatial, age, and genotype-matched samples. In addition, significant changes in either abundance or ratios of *sn*-isomers in a set of GPs over aging and AD progression have been revealed. In this study, our research demonstrated a significant decrease in the total abundance of all 6 GP classes in AD mice across all three functional brain regions compared to WT at 3 months of age, indicating early temporal aberrant GP homeostasis occurred in all 3 functional regions of our AD model mice. This is in line with a body of evidence linking GP depletion with AD[9,61]. Moreover, notable reduction of GP with PUFAs (e.g., DHA), at the *sn*-2 position was observed in AD and aged mice across all 3 brain regions. Therefore, there seems to be a temporal and genotype-dependent shift in the progression of changes. Herein, our results are consistent with previous studies and suggest that a drastic decrease in DHA-containing GPs might be an indicator of AD and aging[63,65]. Obviously, it is often difficult to claim biological implications of MS-based findings and further studies are required to clarify the currently unclear molecular mechanisms of GP metabolism with aging and AD. However, it is tempting to connect our present data with independent studies that link depletion of DHA-containing GP to changes in membrane fluidity and flexibility as well as regulation of neuroinflammation[9,63–65,69].

In addition to reveal significantly altered GPs in AD and aging mice, we found notable alterations in relative abundance ratios of GP *sn*-isomeric pairs as well as GP/lyso-GPs, and their potential correlation to related enzyme activities in GP remodeling pathways. In this study, with the capability to discriminate *sn*-position, the ratio of *sn*-isomer pairs showed significant alterations between WT and AD groups (Fig. 5m). Intriguingly, although many pairs of the *sn*-position isomer simultaneously decreased in AD and aged mice, more significant changes in the ratio of many *sn*-isomer pairs were observed in AD genotype than aging. Therefore, the ratio of *sn*-isomer pairs may show distinct associations with AD pathogenesis. In addition, our study revealed a significant elevation of the ratios of LPC 18:0/0:0 to LPC 0:0/18:0 in AD mice compared to WT mice at 3 months in hippocampus. These changes might be linked to lipid biosynthetic pathways of GPs, which comprise two components: the de novo pathway and the remodeling pathway (Fig. 5n)[70]. In the de novo pathway, PA is initially synthesized from glycerol-3-phosphate (G3P) and then converted to diacylglycerol (DG) or cytidine diphosphate-DG (CDP-DG), which will be ultimately converted into cardiolipin (CL), PG, PI, and PE, PC, PS, respectively. Subsequently, in the remodeling pathway, GP acyl chains are remodeled by orchestrated reactions of PLAs, acyl-CoA synthases, acyltransferase, and lysophospholipid acyltransferases (LPLATs). The increase of the ratios of LPC 18:0/0:0 to LPC 0:0/18:0 suggested that the *sn*-2 PUFA-containing GPs might be more degraded than their corresponding isomers, potentially correlating with the increased PLA2 activities[9]. We further verified that the cPLA2 activities in AD mice at 3 months were remarkably higher compared to WT mice at 3 months in hippocampus (Supplementary Fig. 29). This finding is also aligned with existing literature reporting increased cPLA2 activity in AD brains[67]. Interestingly, we observed a significant increase of the ratio of *sn*-2 PUFA-containing 1,2-diacyl GPs to their corresponding isomers in

AD mice compared to WT mice, and most of the ratios also showed significant changes at 8 months, indicating that the *sn*-1 PUFA-containing 1,2-diacyl GPs might be more degraded than their corresponding isomers in AD as the abundance of both *sn*-isomers decreased. Previous reports also demonstrated that the activities of various enzymes were significantly changed in AD patients[68], including acyltransferase, phosphodiesterase, LPLATs, and other enzymes, which were involved in the remodeling process of acylating lyso-GP into 1,2-diacyl GP to maintain the composition of neuronal membranes[70,71]. The activities of these enzymes were likely also altered in the complex lipid metabolism processes of AD mouse brains considering both increased cPLA2 activities and the elevated ratio of *sn*-2 PUFA-containing 1,2-diacyl GPs to their corresponding isomers in AD groups. Moreover, we found that the ratios of GP/lyso-GP across different GP classes showed varied alteration trends between WT and AD mouse brains. Therefore, it is possible that some other enzymes involved in the lipid metabolic pathway of AD mouse brains were also altered[63,68,70]. The exact molecular mechanisms underlying aging and AD disease pathologies correlated with lipid metabolism warrant comprehensive research.

Overall, the present results, together with previous findings from literature, suggested that it is important to investigate the dysregulated lipid metabolism in AD pathology at GP *sn*-position resolved level. Beyond the *sn*-position resolved GP quantification results, orthogonal enzyme activity assay also disclosed altered cPLA2 activity in different brain regions in aging and AD progression, suggesting potential alteration of enzyme activities in the GP remodeling pathways. We recognize that in the absence of clear mechanistic validation, our MS data should not be overinterpreted. However, the fact that they are consistent with the above-mentioned independent studies suggests that the MS-based results provide promising candidates of GP isomers for AD pathology and a more systematic investigation of GP-related pathways in AD is warranted.

In summary, we developed a 4D lipidomics strategy that enables large-scale, label-free, robust, and rapid GP profiling at *sn*-position resolved level without additional derivatization and instrument modification. Delving deeper, we have constructed a pioneering and valuable 4D database with 498 GPs at *sn*-position resolved level, sourced from both standards and mouse brain lipid extracts. This was complemented by an expansive 4D library of 2500 GPs, constructed via a robust machine learning-based predictive approach. This database and library would be groundbreaking contributions to the field and are instrumental for researchers in biology and analytical sciences, offering valuable resources for detailed lipid structural identification. Our study not only delves into the spatial and temporal diversity of GPs in the mouse brain but also highlights potential molecular signatures of AD progression. This might offer deeper insights into disease pathology associated with dysregulated lipid metabolism and aid in uncovering potential lipid biomarkers for diseases. However, several limitations exist in our study. Firstly, for the very low abundance isomers of GPs with complicated fatty acyl compositions, although they might be identified in our workflow from MS/MS spectra in negative mode, they were not annotated from the HRdm drift spectra to enhance the identification accuracy and confidence. This limitation would be addressed by employing high-sensitivity nano-ESI-MS and next-generation IM-MS with higher resolution. The challenges of annotating GPs with multiple major fatty acyl compositional isomers could potentially be overcome by combining high-resolution IM-MS with advanced chromatographic separation, including extended chromatographic gradient and multidimensional chromatographic separation. Secondly, lipid C=C positional isomers could not be differentiated on this system since it requires IM $R_p$ ~ 1000 for baseline separation. Incorporating additional sample treatment steps such as cutting-edge derivatization/fragmentation strategies[7,18,26,28] to pinpoint C=C information will ultimately achieve complete in-depth structural

profiling of the lipidome. Thirdly, in our CCS prediction model, although small variances are still inevitably present between the experimental and predicted CCS values, we reason that involving more stereospecific 3D MDs would provide a more accurate prediction model. Additionally, we envision that a joint effort from the whole lipidomics community to expand the number of empirical measurements within the CCS database will be an invaluable addition. Another limitation of our study is that, although we performed orthogonal enzyme activity analysis to facilitate further exploration of the lipidomics data, it did not comprehensively cover all possible related enzymes. For example, we tested cPLA2 activities, but it is crucial to note that, besides cPLA2, various other isoforms of PLA2 including secretory phospholipase A2 (sPLA2), calcium-independent phospholipase A2 (iPLA2), also exist in brain regions[9]. Given the diversity of PLA2 subtypes and additional enzymes, such as LPLATs, phospholipase A1s, acyl-CoA synthases, and transacylases, involved in lipid remodeling, further investigation into the changes in the activities of multiple enzymes is necessary in the context of AD. Nevertheless, our data provides insights into possible enzyme activity alterations in aging and AD progression. It is worth mentioning that although we tried to minimize postmortem lipid changes that may occur in the brain by minimizing the euthanasia to tissue collection time, the possible oxidation of GPs is unavoidable.

Of additional note, it remains to be seen if the dynamics of bioactive lipid mediators, including eicosanoids and resolvins which are potentially evoked by the release of PUFAs from GPs, could offer deeper insights for early diagnosis and therapeutic strategies. Additionally, we plan to utilize imaging mass spectrometry in conjunction with high-resolution ion mobility, along with our established database, to map the spatial distribution of *sn*-position resolved GPs in a region-specific manner across the entire brain. Collective studies may be possible to provide a deeper perspective for early diagnosis, and preventive or therapeutic options for AD. In the future, together with biological validation, our strategy could serve as an enabling tool not only providing in-depth lipid structural characterization but also sensitively monitoring the differential expression of enzymes involved in GP remodeling to ultimately provide crucial mechanistic insights into many disease pathologies.

## Methods

### Mouse brain tissue collection

APP695/swe/PS1-dE9 (APP/PS1) double transgenic male mice were obtained from Jackson Laboratory (MMRRC Stock No. 34832-JAX). Genotyping from tail DNA was performed at weaning by Transnetyx (Cordova, TN). APP/PS1 mice were studied at the age of three months ($n = 3$) and eight months ($n = 3$). Male mice were studied with wild-type (WT) littermates used as controls. WT mice were also studied at the age of three months ($n = 3$) and eight months ($n = 3$). All male mice were housed in standard cages provided by the University Laboratory Animal Resources and grouped with littermates with 1–5 mice per cage. Sex and gender-based analysis is not considered in this study. Animals were housed in facilities with a standard 12-hour light-dark cycle, humidity of 50% at 24 °C, and were provided standard chow and water *ad libitum*. Animal experiments were performed in accordance with the National Institutes of Health Guide for the Care and Use of Laboratory Animals and were approved by the Institutional Animal Care and Use Committee of the University of Wisconsin-Madison (protocol #M005120). For tissue collection, mice were euthanized per protocol in a $CO_2$ chamber, and brains were extracted from the skull and rinsed in ice-cold 1X PBS solution. The brains were then dissected to separate out the hippocampus, cerebellum, and cortex according to the established protocol[72], the details were shown in Supplementary Fig. 30. The tissue was directly frozen in liquid nitrogen and kept at −80 °C before further experimentation.

### Sample preparation

Lipid standards were purchased from Avanti Polar Lipids (Alabama, USA). ESI low-concentration tune mix was obtained from Agilent Technologies (Santa Clara, CA). Phospholipase A2 from porcine pancreas (P6534, ≥ 600 U/mg) was purchased from Aladdin (St. Louis, MO). All other chemicals (LC-MS-grade) were purchased from Fisher Scientific (NJ, USA) and used without further purification. Lipid standard stock solutions were prepared in chloroform: methanol (v/v, 1:1).

A modified Folch method was employed for lipid extraction from brain hippocampus, cerebellum, and cortex samples[73]. Approximately 10 mg of brain tissue from wild-type (WT) and APP/PS1 (AD) mice were weighed and dissolved in 0.5 mL of cold PBS. Each sample was spiked with 3 µL of EquiSPLASH (100 µg/ml of each isotope-labeled lipid standard) before lipid extraction. Brain tissues were homogenized with a probe sonicator in an ice water bath with a pulse of 10 s on and 10 s using 60 W energy for 20 cycles and subsequently mixed with 0.5 mL cold methanol and 1 mL cold chloroform for liquid–liquid extraction. After 10 min of vortexing, the mixture was centrifuged at 4 °C for 15 min at 10, 000 × g followed by collecting the bottom layer of chloroform. The above liquid–liquid extraction procedure was repeated twice. The chloroform layers from the three extractions were then combined and evaporated to dryness in a vacuum concentrator. The extracted samples were stored at −20 °C prior to MS analysis.

### PLA2 digestion

The PLA2 digestion was performed using the reported protocol with some modifications[51]. Briefly, 10 nmol dried GP standard was resuspended in a mixture of 480 µL PBS buffer and 10 µL 100 mM $CaCl_2$. The mixture was thoroughly vortexed for 2 min. Subsequently, 10 µL of the enzyme solution was added. The resulting mixture was vortexed for 2 min, then incubated at 37 °C for 4 h. Lipids were extracted by adding 500 µL of chloroform and 500 µL of methanol. The mixture was again vortexed for 2 min and centrifuged at 10, 000 × g at 4 °C for 5 min. The bottom phase was collected and evaporated to dryness in a vacuum concentrator. Dried samples were reconstituted in methanol for LC-IM-MS/MS analysis.

### cPLA2 activity measurement

cPLA2 activity in brain tissue was determined using the cytosolic phospholipase A2 assay kit from Abcam (ab133090) according to manufacturer's instructions. Briefly, approximately 5 mg of mouse brain tissue were homogenized in 200 µL HEPES/EDTA cold buffer (50 mM HEPES, pH 7.4, 1 mM EDTA) with a probe sonicator in an ice water bath. After centrifugation at 10,000 × g for 15 min at 4 °C, 200 µl of each supernatant was concentrated using 10 K molecular weight cut-off concentrators (Thermo Fisher Scientific, San Jose, CA). sPLA2 and iPLA2 have been removed by membrane filter and inhibited by bromoenol lactone respectively according to manufacturer's instructions. Each sample (10 µl) was used to determine cPLA2 activity after 60 min of reaction. The final absorbance at 414 nm was read using a plate reader. The cPLA2 activity was normalized by protein concentration. Protein concentrations were measured using a BCA protein assay kit (Thermo Fisher Scientific, San Jose, CA).

### LC-IM-MS/MS analysis

The LC-IM-MS/MS analyses were performed using an Agilent 1290 UHPLC system coupled with Agilent 6560 IM-QTOF platform (Agilent, Santa Clara, CA). LC separations were performed on a C18 column (Phenomenex Kinetex, 150 mm × 2.1 mm, 1.7 µm particle size) with column temperature maintained at 50 °C. The LC separation was performed with mobile phases A of 5 mM ammonium acetate in methanol/acetonitrile/water (1:1:1, v/v/v) and mobile phase B of 5 mM ammonium acetate in isopropanol/acetonitrile (5:1, v/v) at a flow rate of 0.3 mL/min. The 25 min elution gradient was set as follows: 0–0.75 min: 20% B; 0.75–2.5 min: 20% B to 40% B; 2.5–4.5 min, 40% B to

60% B; 4.5–19.5 min: 60% B to 98% B; 19.5–21 min: 98% B to 20% B; 21–25 min: 20% B. The sample was maintained at 4 °C during the whole analysis.

The MS parameters for both positive and negative ESI modes were set as follows: mass range, 50–1700; sheath gas temperature, 350 °C; sheath gas flow, 12 L/min; dry gas temperature, 325 °C; drying gas flow, 8 L/min; nebulizer pressure, 35 psi; capillary voltage, 3500 V. The maximum drift time was set as 60 ms and the scan rate was set to 0.9 frames per second. Trap filling and trap release times were set as 1800 μs, and 200 μs, respectively. Pulsing sequence length was set as 5-bit. The pressure of the drift tube was set at 3.95 Torr. Capillary voltage was set at 3500 V, or −3500 V in positive and negative modes, respectively. The "Alternating frames" mode of IM-MS was used for the data-independent MS/MS acquisition. The collision energy in frame 2 was set as 25 V in positive mode or 30 V in negative mode. The entrance and exit voltages of the drift tube were set as 1500 V and 250 V, respectively. The "Auto MS/MS" of QTOF-only mode was used for the data-dependent MS/MS acquisition. All data acquisitions were carried out using Mass Hunter Workstation Data Acquisition Software (v 11.0).

### High-resolution demultiplexing (HRdm) and CCS determination
Multiplexing acquisition mode was utilized to achieve the high resolving power of DTIMS. Before HRdm and CCS analyses, multiplexed raw data files were firstly demultiplexed using the PNNL PreProcessor (v 4.0 build 2022.02.17, https://omics.pnl.gov/software/pnnl-preprocessor). Demultiplexing was performed using the data interpolation feature to facilitate the inherent reduction of data points across the drift peak observed with HRdm analyses. Parameters for data deconvolution were set as follows, interpolate drift bins: 1 drift bin becomes 3 drift bins; Demultiplexing: chromatography/infusion; (moving) average: 3; Signal Intensity lower threshold: counts 20; Spike Removal: Require 1 adjacent points per dimension. Following demultiplexing, feature finding was performed on MassHunter IM-MS Browser (v 10.0, Agilent) using the ion mobility feature extraction (IMFE) algorithm. Parameters used for IMFE were listed as follows, Processing: Chromatography; isotopic model: Common organic molecules; Limit charge state: z < =2; Ion intensity >=100. HRdm processing was then performed using High-Resolution Demultiplexer (v 2.0, Agilent), which performs a post-processing enhancement to the IM $R_p$. The detailed settings are listed as followers, HR processing level: High; m/z width multiplier: 6; IF multiplier: 1. CCS values were subsequently determined using single-field CCS method in MassHunter IM-MS Browser with the Agilent tune mix, which has been described in detail previously[45]. In this study, the CCS value obtained for each isomeric lipid ion was put together with its m/z, retention time (RT), and fragment ions and then imported as a transition list into Skyline-daily (MacCoss Lab Software, v. 22.2.1)[36], which was subsequently used for 4D lipid library construction and peak interpolation and integration for the LC-IM-MS/MS analyses.

### In sillico GP structural simulation and CCS calculation
Initially, we determine the protonation state of the gas-phase molecule and generate 500 different conformer structures using the RDKit toolkit[74]. Each conformer is then optimized with the Universal Force Field (UFF) and divided into several clusters for structurally distinct conformations. Potential structures of compounds are generated from the clusters and are optimized in Gaussian 16[75], using B3LYP Density Functional Theory (DFT) and the 3-21 G basis set. The 6-31 G + (d) basis set is then used to generate the energy-minimized structures. Theoretical CCS values for the compounds are calculated from their DFT-optimized geometries using IMoS (v. 1.13) software[76], employing the trajectory method (TM) models with 3 orientations and 300000 nitrogen gas molecules per orientation. Other settings in IMoS, including the use of nitrogen as the drift gas, are kept default.

### Development of experimental GP database
We used the LC-IM-MS/MS method to acquire the m/z, RT, CCS, and MS/MS spectra of lipid extract from pooled mouse brain samples. The GP library was developed within Skyline software according to the previous report with light modifications[36]. The workflow for LC-high resolution-IM-MS/MS-based 4D sn-resolved GP library construction was shown in Supplementary Fig. 31. The schematics of the GP identification in as shown in Supplementary Fig. 17b. Lipid subclass information was obtained from the distinct head group related fragmentation ions in positive mode. The fatty acyl composition for each lipid species was obtained from LC–MS/MS (DDA) in negative ion mode. The MS/MS spectra acquired under DDA mode in negative mode were subsequently assigned to the IM peaks processed by HRdm. The transition lists were generated manually or using LipidCreator[77] and MS DIAL (v 4.80)[53]. MS/MS spectra of mouse brain samples were validated by "Auto MS/MS" (data-dependent acquisition, DDA) at CE = 25 V in positive and CE = 30 V in negative mode. The library was populated with lipid class, name, RT, precursor formula, precursor m/z, adducts, product and neutral loss formulas, adducts, precursor drift time, precursor CCS, and high-energy drift offset values. All lipids in the library were manually validated according to the quality control requirements[78] and met the confidence criteria of presence in more than 5 pooled sample runs, ± 5 ppm mass measurement accuracy, 2 min retention window, and 0.3% CCS deviation. iRT calculators were built using 10 isotope-labeled lipid standards from EquiSPLASH spanning the LC gradient. In addition, GP standards from 6 classes were used to measure their RT and CCS to validate the accuracy of this method (Supplementary Table 1). In total, 498 GP species were incorporated into the Experimental GP library including calculated m/z, RTs, CCS values, and MS/MS spectra.

### CCS and RT prediction for extending GP library
To develop a machine learning-based prediction model of GP CCS values and retention times, we used 419 GPs from three GP classes (205 PCs, 173 PEs, and 41 PSs) from the experimental GP database and randomly divided them into a training data set (80%; n = 335) and an external validation data set (20%; n = 84). Since the SMILESs of lipids in LIPID MAPS do not contain the adducts, in this study, SMILES of adducted GPs were manually generated by combing the SMILES of head groups, adducts, fatty acyl chains in sn-1 and sn-2 in a uniformed format. The protonation sites are determined to be the quaternary amine for PC, and the primary amine for PE and PS. For each GP, 221 MDs were first calculated by R package "rcdk" (R version 4.2.1) using the SMILES structure. For CCS prediction, another 32 MDs, which demonstrated more than 0.1% differences between sn-position isomers, were selected from 1824 MDs calculated by Mordred software and added to each GP. Next, MDs were optimized using the LASSO algorithm (R package "glmnet") in the training set. 10-fold cross-validation was used to optimize the "lambda" value in LASSO, and the "lambda" with the lowest mean squared error (mse) was selected as the best model. From this optimization, 37 MDs were selected for the final CCS prediction. Employing the same methods, 51 MDs from "rcdk" were selected and used for retention time prediction. SVR algorithm (R package "e1071") was used to build the prediction model. Two parameters, cost of constraints violation (C) and gamma (γ) were first optimized from 168 parameter combinations via 10-fold cross-validation with 100 repeats to achieve the best prediction performance. The optimized parameters were optimized as follows: C: 36, γ: 0.001 for CCS prediction, and C: 8, γ: 0.001 for retention time prediction, respectively. In the end, the SVR-based CCS and retention time prediction were further validated using the external validation data set.

To supplement the experimental GP library, we developed an extended sn-position resolved GP library. Each GP class was considered in combination with the 35 most common fatty acyl chains in mammals; all unsaturated FAs were considered in their most common C=C

double bond composition since C=C double bond isomers are indistinguishable in our strategy. We calculated the MDs and predicted the CCS values of GPs with the C=C bonds in the *cis* (Z) configuration, as this configuration is the most dominant in naturally occurring lipids in mammals. The 35 most common fatty acids are listed in Supplementary Table 3. Among all 2919 GPs in the class of PC, PE, and PS in LIPID MAPS, 419 GPs were already incorporated into the experimental GP library of mouse brain extract. Consequently, the rest 2500 GPs were used for prediction. Precursor *m/z* values of the 2500 GPs were calculated using the monoisotopic formula of $[M+H]^+$. Their MS/MS fragmentation ions were primarily generated and modified by using LipidCreator[77]. Their CCS and RT values were generated by the above-established SVR-based prediction model. The transition list of this extended library was built with the predicted CCS value, RT, precursor *m/z*, and fragment ions of each GP to be compatible with automated analysis by Skyline. The extended GP library was provided in Supplementary Data 7.

## Data processing

MS and CCS values were single-field calibrated using the Agilent tune mix under the same IM-MS parameters as other data files. After HRdm deconvolution, the CCS calibration coefficients (TFix and Beta) of calibrants were calculated in IM-MS Browser and applied to data files in the same batch of experiments. The GPs in the experimental data, after CCS calibration, were annotated using Skyline with the experimental mouse brain GP library and extended GP library. We set the minimum peak height at 3000 counts in MS1 as a criterion for identification. The MS1 match tolerance was set as 20 ppm. Retention time tolerance was set as 2 min window. Drift time filtering was used with a resolving power of 160. All annotations were manually validated. The semi-quantification of a class of lipid molecular species was achieved through the normalization of the individual molecular ion peak area to its respective deuterium-labeled internal standard. Statistical analyses were conducted in R (v.4.2.1), Origin 2020, and GraphPad Prism 9.0. Abundances of GPs were compared using one-way ANOVA with correction for multiple comparisons using the two-stage linear step-up procedure of Benjamin, Krieger, and Yekutieli at a 0.05 false discovery rate (FDR).

## Reporting summary

Further information on research design is available in the Nature Portfolio Reporting Summary linked to this article.

## Data availability

The information on 498 GPs in the experimental GP library and 2500 GPs in the extended GP library are provided in Supplementary Data 4 and 7, respectively. The internal validation and external validation results of CCS and retention time are provided in Supplementary Data 5 and 6. The identified GPs in the mouse brain and their abundances are provided in Supplementary Data 8 and 10, respectively. The raw data generated in this study has been deposited in Zenodo (https://doi.org/10.5281/zenodo.12327665)[79]. Source data are provided with this paper.

## Code availability

The source code is provided in Github (https://github.com/lingjunli-research/Glycerophospholipid-CCS) and Zenodo (https://doi.org/10.5281/zenodo.12351060)[80].

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

## Acknowledgements

The authors would like to thank John Sausen, John Fjeldsted, and Sheher Mohsin from Agilent Technologies for their advice and discussions. We thank Yuanfei Wang from Peking University and Dr. Shihe Pan from Tianjin University for their helpful suggestions. We also thank Dr. Ting-Jia Gu, Dr. Danqing Wang, Peng-Kai Liu, Dr. Hua Zhang, and Dr. Min Ma from the Li Research Group for their insightful discussions. This research was supported in part by the NIH grants R01AG078794 (to L.L. and L.P.), R01 DK071801 (to L.L.), and R01 AG052324 (to L.L.). L.L. acknowledges the funding support of NIH shared instrument grants (S10OD028473, S10RR029531, and S10OD025084), an NIH grant R21AG065728, a Vilas Distinguished Achievement Professorship, and Charles Melbourne Johnson Professorship with funding provided by the Wisconsin Alumni Research Foundation and University of Wisconsin-Madison School of Pharmacy. S.X. acknowledges the funding support for a Postdoctoral Career Development Award provided by the American Society for Mass Spectrometry (2024).

## Author contributions

L.L., S.X. and Z.Z. conceived and designed the project. S.X. and Z.Z. performed most of the experiments, processed the data, and constructed the database. D.G.D. and G.L. assisted in performing experiments and data analysis. L.P., M.R. and M.B. contributed to animal experiments and mouse brain tissue collection. S.X., Z.Z. and L.L. wrote the manuscript. All authors discussed the results, provided feedback, and approved the final version.

## Competing interests

The authors declare no competing interests.
