## [Transparent Peer Review file · Nature Communications]

Spatially and temporally probing distinctive glycerophospholipid alterations in Alzheimer's disease mouse brain via high-resolution ion mobility-enabled sn-position resolved lipidomics

Corresponding Author: Professor Lingjun Li

Figures originally included in the author's rebuttal have been redacted from this file.

Version 0:

Reviewer comments:

Reviewer #1

(Remarks to the Author)

In this manuscript, authors developed a four-dimensional lipidomic strategy to resolve GP sn-position isomers on high-resolution demultiplexing (HRdm) ion mobility spectrometry. With MS2 fragmentation rule and high-resolution CCS, authors identified GP sn-position isomers in chemical standards in pooled brain samples. They trained a sn-position-level GP CCS prediction model and then built an extend GP library. Finally, they illustrated metabolic alterations in three functional brain regions of mice in AD progression. Overall, this work is potentially interesting. Here were some questions to be addresses:

Comment #1. To the best of my knowledge, HRdm was not novel for isomer separations as reported by many recent works (<https://doi.org/10.1021/acs.analchem.1c04267>, <https://doi.org/10.1021/acs.analchem.3c02213> and <https://doi.org/10.1021/jasms.3c00083>, etc.). In this manuscript, authors showed that this technology could separate GP isomers on sn-position levels in chemical standards and used it to resolved isomers in complex biological samples. GP with different fatty acyl compositions would generate various degrees of structural differences on sn-position levels. Authors did not show with how much structural differences HRdm could resolve them. On the other hand, signals of HRdm were generated through complex deconvolutions. Were the so-called isomer signals from HRdm were artificial signals but not the true isomers? The feasibility of HRdm separating GP isomers on sn-position levels needs much more comprehensive evaluation.

Comment #2. In fact, resolving lipid isomers on sn-position levels had been reported by many publications In par. MS2 fragmentation rules in the negative ionization mode had resolved this problem to a large stand (<https://doi.org/10.1038/nmeth.4470> and <https://doi.org/10.1186/s12859-017-1744-3>). Furthermore, the incorporation of IM and LC-MS improved the precursor ion fractions and generated purer MS2 spectra, boosting lipid isomers identifications on sn-position levels with MS2 fragmentation rules (<https://doi.org/10.1016/j.aca.2022.339886>). Authors did not show how many isomers could be resolved MS2 fragmentation rules and how much HRdm could improve the separation and identification based on the rules.

Comment #3. As far as I know, no MS/MS data could be obtained while using multiplexing on Agilent DTIM-MS. How were the MS/MS spectra acquired under low resolution IM mode assigned to the IM peaks acquired and processed after HRdm? Please clarify this technical point in the Methods part. Would the inter-batch drifts of drift times result in wrong MS/MS assignment? Were the MS2 spectra in Fig. 2c-e acquired with targeted or DDA method under LC-MS/MS?

Comment #4. The overall data acquisition and data processing workflow were complicated. It would be much better to illustrate them with a main figure or supplementary figure.

Comment #5. Fig.1 showed the improved separation for equimolar mixtures with HRdm. How did HRdm perform while separating isomer mixtures with different ratios? Would HRdm generate artificial peaks and result in overreports structure information?

Comment #6. Authors developed a combined strategy with MS2 fragmentation rules (the 1st rule) and CCS values from HRdm (the 2nd rule) to resolve GP isomer on sn-position level. To my knowledge, MS2 fragmentation rules (the 1st rule) were able to resolve most of the sn-position GP isomers, especially IM largely improving the MS2 spectral precursor ion fraction and reducing chimeric spectra. How much did HRdm improve the isomer resolving based on the MS2 fragmentation rules? As authors only showed the comparison of the combined strategy with LC-MS/MS strategy in Fig. 2g, please also compared performances of combined strategy with those based on MS2 fragmentation rules.

Comment #7. Line 241, Page 7: "As this prevailing trend has not been reported previously due to the limitations in IM Rp, we reasoned that this finding could be attributed to the higher gas-phase flexibility and freedom of the fatty acyl chain at the sn-1 located at the terminus of the glycerol backbone." I appreciate the rule and think it would be better if more evidence like computer-aided molecular modeling could support this rule.

Comment #8. Some technical points of CCS prediction were not clear. For the model training, MDs must be calculated from the exact structures including but not C=C location and geometry. However, the CCS values collected by authors were only assigned to sn-position level. How did authors assign an exact specific structure to a sn-position-level CCS values? And MDs and CCS values might be different if you select different C=C location and geometry isomers.

Comment #9. Line 594, page 17: "Since the SMILESs of lipids in LIPID MAPS do not contain the adducts, in this study, SMILES of adducted GPs were manually generated by combining the SMILES of head groups, adducts, fatty acyl chains in sn-1 and sn-2 in a uniformed format." is not clear, please give an example. How did the authors determine the ionization site of the adducts?

Comment #10. I am curious about how much did the sn-position-resolve CCS prediction model improved compared with those tools developed on lipid specie level like LipidCCS and CCSbase.

Comment #11. More about the extended library, the retention times were not included in the library. In fact, the retention times of lipids on a C18 column could be easily predicted with Quantitative Structure Retention Relationships (QSRR). The predicted RT would reduce false positive IDs to a large stand.

Comment #12. Fig. 4g showed the percentage of GP classes in different clusters. Would it generate more specific results with molecule percentage of different GP classes?

Comment #13. Authors made a conclusion that uniform cPLA2 enzyme activities differed in different age and functional regions. Could authors confirm the results with other orthogonal methods like comparing the enzyme expression?

Comment #14. Line 403, Page 12: "the clusters between WT at 3 months and AD at 8 months were clustered adjacent to one another" should be "WT at 8 months and AD at 3 months".

Comment #15. Supplementary Table 1, No. 20: "PA 15:0/18: (d7)" should be "PA 15:0/18:1 (d7)".

Reviewer #2

(Remarks to the Author)

In the article „Spatially and temporally probing distinctive glycerophospholipid alterations in Alzheimer's disease mouse brain via high-resolution ion mobility-enabled sn position resolved lipidomics “ by Xu et al. the authors present an ion mobility spectrometry-mass spectrometry (IMS-MS) workflow in order to study variation of stereospecific numbering (sn) isomers of lipids. These sn-isomer changes have been linked to the dysregulation of the Land's cycle enzymatic machinery but corresponding lipids are notoriously difficult to analyze. The authors address this challenge by implementing Hadamard transform-assisted demultiplexing IMS-MS of complex lipid samples. By multiple ion injections at different preset offset times to generate different arrival time offsets, the authors remeasure the same sample multiple times. The multiplexed IMS data is disentangled by Hadamard transform to create an IMS result with considerably improved signal-to-noise as well as IMS resolution. This enables to almost baseline resolve sn-isomers of various lipid classes and perform quantitation on MS1 and MS2 level. The authors use this methodology to develop a large-scale CCS database training a corresponding bioinformatic model for confirmation of sn-identity without the use of authentic standards. The optimized data analysis pipeline and bioanalytical methodology is employed to quantify the spatial (cortex, hippocampus, cerebellum) and temporal (3 and 8 months) changes of lipid sn-isomers in brains of wild type and APP/PS1 (Alzheimer's disease model) mice. For some of the lipids, age and spatial sn-isomer alterations are significant.

This is a well-written manuscript that extends a novel IMS technique to sn-isomers of lipids. The analytical application is clearly interesting for a broad scientific audience as it "only" relies on multiplexed data acquisition and data analysis. So, in principle it should be possible to adapt the methodology in labs with access to drift time IMS instruments. Additionally, the method offers new perspectives in the study of the Land's cycle that has been understudied on the lipid level due to the difficulty to separate and quantitate sn-isomers. To further improve the manuscript the authors need to address the following issues:

- The main performance test of the presented IMS-MS method is based on the comparison with established PLA2 digests. Typically, MS read-outs have uncertainties in relative lipid intensities of 3 – 5 % and PLA2 digests itself (at least when going through the literature) have intrinsic uncertainties of 3 – 5% as well. Therefore, it surprising that the comparison of a single PLA2 digest result with a single IMS-MS result of the same standard overlaps within a range of less than 1.5 %. As all

flowing conclusions are based on this single point comparison, I strongly recommend to perform replicates for the PLA2 and IMS-MS measurements (n=10-20).

- On a similar note: Please report coefficients of variation of IMS-MS measurements. This will help to demonstrate the robustness of the nice methodology presented in the manuscript.
- The authors demonstrate the capability to quantify significant changes of sn-isomers in complex samples. The manuscript would be improved when discussing potential follow-up experiments that will help to investigate the biological significance of the sn-isomer alterations.
- An extensive CCS database is created to analysis the data. Multiple groups have established CCS databases for lipids in the literature but often CCS values from different groups do not overlap within the margin of error. Why is the database created in this manuscript more relevant than other databases?
- On multiple occasions the authors mention that C20:4 can lose CO₂. Please point out that this is only true when fragmenting in negative-ion mode but not when using metal-adducts in positive-ion mode.
- Some minor wording suggestions: (line 76) Uneven should be replaced by heterogenous or non-uniform; (line 86 and rest of article) mainstream, do you mean routine or standardized?
- The authors should cite the following recent article on the intermediates during CID of lipids (10.1021/jacs.1c06944 and 10.1021/jasms.1c00277) and additionally EID (10.1016/j.ijms.2022.116998) should be discussed and some recent reviews on this matter should be cited (10.1021/acs.analchem.1c00061 and 10.1007/s00216-021-03425-1).

Reviewer #3

(Remarks to the Author)

This manuscript reports generation of a library and methodology for analysis of imaging mass spectrometry data which is of great value to the field as a resource or methods paper.

They constructed a 4-dimensional database of glycerophospholipids in mouse brain with expanded collision cross-section (CCS) reference library predicted by machine learning. This allows automated profiling of GP with specific acyl chain sn-position assignment which is of critical value. They used a novel high resolution demultiplexing (HRdm) strategy for increased diffract tub ion mobility and ion mobility separation of sn-isomers allowing identification of acyl chain specificities from pooled mouse brain extracts (bulk).

For example, valuable information is reported regarding the relative abundances of carbon double bonds in glycerol phospholipids is reported regarding a lower collisional cross section value which may be due to higher gas-phase flexibility and freedom of the fatty acyl chain at the sn-1. They also found that from commercial standards, that lyso-GPs with acyl chain at sn-1 position generally showed smaller CCS values consistent with previous reports. These are highly impactful findings and will likely affect multiple future IMS studies.

However, the biological ramifications of this study are limited based on the following issues.

The three functional regions of the AD mouse brain were not justified as to why they are under investigation.

The mouse model was not justified or compared to other potential models to be used.

The biological ramifications of the altered GP content in these regions was not discussed.

Biological prioritization of 592 lipid changes and pathways are not discussed.

Representative images are included for only 2 ions (Fig h, i) not for other regional ions of interest in hippocampus, cortex or cerebellum. Fig 1 h, i coronal sections are shown while in all other figures, sagittal section is represented in schematics (Figure 4 and supplemental figures)

No coordinated and defining atlas information for hippocampal, cerebellum or cortical identification are included.

It is unexpected that cerebellum which is generally considered "spared" in AD and hippocampus which is generally considered susceptible to AD pathology have the same number of differential lipids (ie., Fig 4.d.).

In methods, mice were euthanized in CO₂ chamber prior to brain extraction which is highly likely to affect lipid metabolism within several seconds of anoxic condition even prior to death of neurons and glia. These results cannot be considered biologically relevant changes in the mouse brain in vivo due to genetic or age related effects.

In methods it is not clear if whole brain was frozen for imaging MS analysis or only the dissected regions. No whole brain image is shown for representative ions.

Statistics are reported in figures legend as ANOVA, but no false discovery rate is described. These high-dimensional data need FDR correction. Statistical methods are not described explicitly in the methods. FDR correction should be used even if a higher cut-off criteria is used. This should be clearly described in methods in addition to the figure legends.

Supplemental figures are not labeled with figure title or figure legends.

Table S4 and S5 are not clearly labeled and may be missing from manuscript.

This would be appropriate for a methods/resources journal.

Author Rebuttal letter:

REVIEWER COMMENTS

Reviewer #1 (Remarks to the Author):

In this manuscript, authors developed a four-dimensional lipidomic strategy to resolve GP sn-position isomers on high-resolution demultiplexing (HRdm) ion mobility spectrometry. With MS2 fragmentation rule and high-resolution CCS, authors identified GP sn-position isomers in chemical standards in pooled brain samples. They trained a sn-position-level GP CCS prediction model and then built an extend GP library. Finally, they illustrated metabolic alterations in three functional brain regions of mice in AD progression. Overall, this work is potentially interesting. Here were some questions to be addresses:

Response: We are very grateful to the reviewer's positive comments on our study. We also really appreciate the reviewer's constructive comments and suggestions which will of course help strengthen our manuscript. We have included more data, performed some additional experiments, and addressed these comments as the point-to-point responses shown below.

Comment #1. To the best of my knowledge, HRdm was not novel for isomer separations as reported by many recent works (<https://doi.org/10.1021/acs.analchem.1c04267>, <https://doi.org/10.1021/acs.analchem.3c02213> and <https://doi.org/10.1021/jasms.3c00083>, etc.). In this manuscript, authors showed that this technology could separate GP isomers on sn-position levels in chemical standards and used it to resolved isomers in complex biological samples. GP with different fatty acyl compositions would generate various degrees of structural differences on sn-position levels. Authors did not show with how much structural differences HRdm could resolve them. On the other hand, signals of HRdm were generated through complex deconvolutions. Were the so-called isomer signals from HRdm were artificial signals but not the true isomers? The feasibility of HRdm separating GP isomers on sn-position levels needs much more comprehensive evaluation.

Response: We thank the reviewer for their insightful observations and constructive critique regarding the application of HRdm in our study.

HRdm was first reported in 2020, which is a combined data acquisition and data processing strategy for improving the sensitivity and resolution of ion mobility measurements without the need for instrument modifications (Anal. Chem. 2020, 92, 14, 9482-9492). In the past three years, a series of isomeric mixtures were qualitatively investigated with HRdm, including glycans (Nat Commun 2023, 14, 679), peptides (Anal. Chem. 2022, 94, 16, 6191-6199), metabolites (Anal. Chem. 2022, 94, 16, 6191-6199; Anal. Chem. 2021, 93, 51, 17094-17102), and oxidized lipids (Anal. Chem. 2023, 95, 36, 13566-13574). However, utilizing this strategy to systematically separate subtle yet crucial structural sn-position isomers of glycerophospholipids (GPs) in

1 biological samples has not been explored in existing studies. In this study, our work have several notable innovations: the first report of an LC-IM-MS strategy that enables large-scale GP profiling at sn-position resolved level in their unmodified intact conditions without additional derivatization and instrument modification; the construction of a comprehensive 4D database with 498 GPs sourced from both standards and mouse brain lipid extracts; and the development of an in-depth 4D library of 2500 GPs predicted by machine learning, enabling automated profiling of GPs with detailed acyl chain sn-position assignment. Moreover, our strategy details the spatial and temporal diversity of GPs in the mouse brain and highlights putative molecular signatures of AD progression. It will provide a suitable solution to uncover potential lipid biomarkers for disease pathology associated with dysregulated lipid metabolism.

We fully agree with the reviewer that degrees of structural differences on sn-position levels are diverse due to the complex fatty acyl chain compositions of GPs. Generally, sn-position isomers, as constitutional isomers, exhibit CCS difference of approximately 1%, which needs more than 100 resolving power to achieve 10% separation and more than 200 resolving power to achieve 90% separation in low-field DTIMS instruments (Anal. Chem. 2017, 89, 1, 952-959). Owing to the high resolution (approximately 200) afforded by HRdm, most of GP sn-position isomers could be separated sufficiently. The quality of a separation is quantified in terms of two-peak resolution (Rpp) with peaks deemed resolved when Rpp exceeds 0.5. To provide a comprehensive evaluation of the separation efficiency of GP with fatty acyl compositions by HRdm, we assessed the

separation performance of a series of GP isomer pairs, each representing different degrees of structural variation at the sn-position levels. The results are now supplied as the Supplementary Figure 1. Specifically, for GPs with different fatty acyl chain compositions from different classes, such as PC 16:0_18:1, PE 16:0_18:0, PE 16:1_18:2, PG 18:1_16:0, PC 14:0_20:4, PC 14:1_22:5, PE 18:1_22:6, PC 16:0_22:6, PI 16:0_20:4, and PG 18:1_20:4 (Supplementary Figure 1 a-j), the two-peak resolution (R_{pp}) were at a range from 0.78 to 1.15. This indicates that each pair of isomers was successfully separated.

For the analysis of GPs with C&C bond positional and geometric isomers, we conducted a detailed examination of multiple GP standards. These included PC 16:0/18:1(9Z) vs PC 16:0/18:1(11Z) and PC 18:1(9Z)/18:1(9Z) vs PC 18:1(11Z)/18:1(11Z) (Supplementary Figure 6). Typically, such species usually exhibit a CCS or drift time difference of less than 0.2% and require $IM\ R_p \sim 1000$ for baseline separation. As the relatively small CCS differences among C&C bond position isomers are negligible compared to sn-position isomers ($\sim 1\%$), the differentiation of GP sn-position isomers remains unimpeded. In terms of C&C geometry isomers (cis and trans), our study primarily focused on the cis configuration, as GPs with fatty acyl composition in trans C&C configuration are rare in most mammals. The utilization of high-resolution IM to differentiate cis and trans isomers is indeed a promising area of research, and we intend to explore this further in future studies. This approach holds significant potential for advancing our understanding of lipidomic variations and their implications.

2

Hadamard transform is a widely adopted and validated technique used across various of scientific instruments, including Fourier transform ion cyclotron resonance-MS (FTICR), TOF-MS, and capillary electrophoresis (CE), primarily for demultiplexing the multiplexed incoming spectra and thereby enhancing the signal-to-noise ratio. Like other applications of Hadamard transform, the demultiplexed data obtained from the IM-MS platform may contain artifacts, which are distinct from measurement noise. To address this problem, researchers have developed an algorithm that discovers and eliminates data artifacts (J. Am. Soc. Mass Spectrom. 2014, 25, 12, 2020â2027). The algorithm employs an analytical approach to identify and remove artifacts from the data, decreasing the likelihood of false identifications in subsequent data processing. Applying this algorithm significantly enhances IM-MS measurement sensitivity and minimizes artifacts that have previously hindered the application of the Hadamard transform to IMS. HRdm also adopted the algorithm to greatly remove artifacts. We also included an example from first HRdm paper (Anal. Chem. 2020, 92, 14, 9482â9492) that demonstrated how an artifact may look like, as shown in Figure R1. The artifacts, if exist, are usually of low abundance ($<10\%$ of the primary signal within each m/z extraction window) and are sufficiently distant from the signal region of interest to not affect spectral interpretation. IM drift times of genuine ion signal align across isotopic peaks due to the negligible contribution of mass to the ion mobility, whereas artifactual peaks might not be sufficiently distant from the signal region of interest to not affect spectral interpretation. For instance, in our study (Figure R2), artifactual peaks co-existing with PC 14:0/20:4 and PC 20:4/14:0 were easily distinguished from genuine ions due to their long distance from the true ions, their low abundance, and not-aligned isotope peaks. It is important to note that our study utilized the officially released stable version 2.0 of the HRdm software, not the beta versions referenced in some earlier literature. The HRdm 2.0 release includes several key improvements that significantly reduce or eliminate artifacts, such as the âSequence Shift and Scaling (SSS)â for correcting temporal and intensity variations in raw multiplexed data, and âPost-QCâ for ensuring that low signal level data does not produce overly resolved peaks. In Figure 1e-g, the peaks of mixture in the drift spectra also aligned well with each iso-pure isomer measured individually, showing a single peak per isomer in drift spectra without artifacts. As shown in Figure 1h-i, the two peaks of the equimolar mixture of PE and PS sn-isomers showed very close intensities, and IM drift times of their $M+1$ isotopes aligned well with the monoisotopic peaks. With evidence from above, we can safely conclude that the well separated peaks in these standards are indicative of isomers, not artifacts.

In our approach to annotating IM drift peaks from biological samples, we implemented stringent criteria to minimize the false positive identification of potential artifacts, especially for GPs of relatively low abundance. We set the minimum peak height at 3000 counts in MS1 as a criterion for identification, and only annotated the major constituent. For the abundant species with multiple fatty acyl chain possibilities, we only annotated the top four most abundant species. Upon manual inspection of the experimental data, we observed that the IM drift times of all identified GPs consistently aligned with their respective isotopic peaks. This alignment serves as an additional validation of our identification process, as genuine ion signals are expected to

3

exhibit such alignment due to the consistency of ion mobility across isotopes. Implementing these criteria greatly enhanced the reliability and improved accuracy of our identifications. By setting a high threshold for peak height and focusing on the most abundant species, we significantly reduced the likelihood of false positives and improved the overall confidence in our annotations

of IM drift peaks from biological samples.

[Redacted]

Supplementary Figure 1. Drift spectra of a series GP sn-position isomers. Drift spectrum of PC 16:0_18:1 (a), PE 16:0_18:0 (b), PE 16:1_18:2 (c), PG 18:1_16:0 (d), PC 14:0_20:4 (e), PC 14:1_22:5 (f), PE 18:1_22:6 (g), PC 16:0_22:6 (h), PI 16:0_20:4 (i), PG 18:1_22:4 (j).

[Redacted]

4

Figure R1. Example 2D IM-MS spectrum of a mixture of LNFP1 and LNFP2 obtained with 4-bit ion multiplexing and processed with HRdm (from *Anal. Chem.* 2020, 92, 14, 9482â9492)

[Redacted]

Figure R2. IM-MS heatmap of PC 14:0_20:4 in mouse brain; red star indicates the artifactual peaks.

Comment #2. In fact, resolving lipid isomers on sn-position levels had been reported by many publications In par. MS2 fragmentation rules in the negative ionization mode had resolved this problem to a large stand

(<https://doi.org/10.1038/nmeth.4470> and <https://doi.org/10.1186/s12859-017-1744-3>).

Furthermore, the incorporation of IM and LC-MS improved the precursor ion fractions and generated purer MS2 spectra, boosting lipid isomers identifications on sn-position levels with MS2 fragmentation rules (<https://doi.org/10.1016/j.aca.2022.339886>). Authors did not show how many isomers could be resolved MS2 fragmentation rules and how much HRdm

5

could improve the separation and identification based on the rules.

Response: We appreciate the reviewer's insightful comments regarding the resolution of lipid isomers at the sn-position level. While MS2 fragmentation rules in the negative ionization mode have been instrumental in annotating fatty acyl chain compositions of GPs, their capability in resolving sn-position isomers, especially in complex biological samples, has certain limitations.

For label-free lipidomic analysis using commonly used CID-based MS/MS strategies, MS2 fragmentation rule was able to assign fatty acyl composition for GPs but fail to reveal the structural feature of sn-position. Although the intensities of fatty acyl chain carboxylate anions from sn-2 were roughly 3 times higher than those from sn-1 in negative ion mode CID-MS/MS, the confident sn-position assignment by fragmentation rule is only applicable to annotate pure GP standard or well-separated GP sn-isomers in LC. It is challenging to determine if sn-isomers exist or not by only the information of the intensity ratios of fatty acyl chain carboxylate anions in negative ion mode CID-MS/MS spectra.

As sn-isomeric GPs are commonly present in biological mixtures in a wide dynamic range and are commonly co-eluted in RPLC (*J. Am. Chem. Soc.* 2017, 139, 44, 15681â15690; *Nat. Comm.* 2020, 11, 375), although we could assign the sn-position for the dominant GP component by the relative intensities of the two fatty acyl chain carboxylate anions, it would be difficult to prove or rule out the presence of its sn-isomer. Herein, the sn-position information is also not available in LipidBlast which is the most comprehensive CID-based MS/MS database for lipids (*Nat. Methods* 2013, 10, 755â758; *Nat. Methods* 2015, 12, 523â526). Moreover, sn-position features are lacking in most current leading lipidomics studies using CID-MS/MS based label-free lipidomics (*Nat Protoc* 2022, 17, 2415â2430; *Nat Commun* 2023, 14, 937; *Nat Methods* 2023, 20, 193â204; *Nat Commun* 2022, 13, 124). As shown in the valuable piece of literature reviewer provided (*Nat Methods* 2017, 14, 1171â1174. <https://doi.org/10.1038/nmeth.4470>), co-eluting sn-isomers of PE 16:0_20:4 were observed, and they were partially separated on LC in a relative long gradient time (58 min). In fact, GP sn-isomers are difficult to separate on LC using a commonly used short time gradient (15-35 min). As shown in LipidMatch paper that the reviewer provided in this comment (*BMC Bioinformatics* 2017, 18, 331. <https://doi.org/10.1186/s12859-017-1744-3>), the lipid annotation was also achieved in fatty acyl composition level but not sn-position resolved level. Overall, through sole MS2 fragmentation, the annotation could only be achieved at the fatty acyl composition level. We are sorry for not having make this point clear in the Introduction and Results section, and have now added more detailed description, hoping to offer clarification on this matter.

Previously, Zhu and co-workers incorporated IM to LC-MS to improve the accuracy and coverage for lipid identifications (Anal. Chim. Acta 2020, 1136, 115-124.

<https://doi.org/10.1016/j.aca.2020.08.048>; Anal. Chim. Acta 2022, 1210, 339886.

<https://doi.org/10.1016/j.aca.2022.339886>). In their studies, they used the threshold value, 0.96

of intensity ratio of fatty acyl chain in negative ion mode, to roughly determine the dominant fatty acyl sn-position of the GP species from each extracted peak. The clear limitation of this annotation is that, among all co-eluting sn-position isomers from complex biological samples, only the one GP species that was the most abundant was identified. Above mentioned studies greatly improved the depth in lipid structure characterization and pushing our study to the next step for providing unequivocal structural assignment of lipid isomers. These pieces of literature were cited in the revised manuscript. We also have cited more important references on the recent developments in IM-MS for lipidomics in the revised manuscript (including Curr. Opin. Chem. Biol. 2018, 42, 42-50; Curr. Opin. Chem. Biol. 2018, 42, 111-118; Nat. Biotechnol. 2020, 38, 1159-1163; Nat. Methods 2023, 20, 1836â1837).

Most GPs in complex biological samples exist as a mixture form of sn-positional isomers (J. Am. Chem. Soc. 2017, 139, 44, 15681â15690). The sn-positions of fatty acyls could not be confidently identified only using the CID fragmentation strategies, primarily due to the different dissociation efficiencies between isomers and interfering diagnostic ions from co-eluting lipid species. In our study, besides the detailed rule for annotation of the fatty acyl composition from MS/MS fragmentation patterns, the use of HRdm strategy provided high resolution in IM dimension (up to 250) while still allowing millisecond IM separation (within 60 ms) of GP sn-isomers in complex biological samples. The 1st rule was based on the carboxylate anion intensity in MS/MS spectra and abundance of IM-resolved peaks. The combination of the carboxylate anion intensity in MS/MS spectra together with the ion abundance in drift spectra (the 1st rule) and CCS features (the 2nd rule) in the IM spectra acquired by high resolution IMS enables unambiguous GP isomer annotation in the drift spectrum. We have updated Figure 2 to provide a more detailed comparison among these identification methods. As shown in Figure 2f, a total of 318 molecular GP species with fatty acyl composition were identified in mouse brain lipid extracts according to the m/z, retention time, and MS/MS obtained by LC-MS/MS using QTOF-only mode without HRdm. When operating on multiplexed IM-QTOF mode with the same LC gradient, from the ion abundance of drift spectra and MS2 fragmentation rules (the 1st rule), a total of 452 GP species could be annotated. Using the combined strategy of MS2 fragmentation together with IM ion abundance (1st rule) and CCS features (the 2nd rule), 498 distinct GP molecular species were identified from the same sample. This information is now included in the updated Figure 2f.

[Redacted]

7

Fig. 2f The number of GP molecular species identified by LCâMS/MS (IM off) (green), LCâHRdm IMâMS/MS with Rule 1 (purple), and LCâHRdm IMâMS/MS with Rule 1&2 (blue).

Comment #3. As far as I know, no MS/MS data could be obtained while using multiplexing on Agilent DTIM-MS. How were the MS/MS spectra acquired under low resolution IM mode assigned to the IM peaks acquired and processed after HRdm? Please clarify this technical point in the Methods part. Would the inter-batch drifts of drift times result in wrong MS/MS assignment? Were the MS2 spectra in Fig. 2c-e acquired with targeted or DDA method under LC-MS/MS?

Response: We thank the reviewer for highlighting this crucial aspect of our methodology. To address the issue raised, we have updated the Methods section of our manuscript to include a more detailed description of our technical approach. We also added schematics of the GP identification in Supplementary Figure 12. In our study, we utilized Data-Independent Acquisition (DIA) in conjunction with the "Alternating frames" mode when using the multiplexing on Agilent DTIM-MS. The identification was performed using mobility aligned fragment ions. In order to improve the identification accuracy, we acquired both the DIA data by LC-IM-MS/MS using multiplexing and data-dependent MS/MS acquisition (DDA) data by LC-MS/MS using QTOF-only mode in positive and negative ESI mode for validation.

Combining the GP subclass information from positive mode MS/MS and fatty acyl composition from negative mode MS/MS (Supplementary Figure 12a) is the most commonly used approach for confident GP identification (Nat. Methods 2013, 10, 755â758; Proc. Natl. Acad. Sci. U.S.A. 2016, 113, 2573). In our study, as shown in Supplementary Figure 12b, lipid subclass information was obtained from the distinct head group related fragmentation ions in positive mode. The fatty acyl composition for each lipid species was obtained from LCâMS/MS (DDA) in negative ion mode. The MS/MS spectra acquired under DDA mode in negative mode were subsequently assigned to the IM peaks processed by HRdm.

8

We totally understand the reviewer's concern on inter-batch drifts of drift times. Although drift times could have inter-batch differences due to the subtle changes in the instrument operation over time, the CCS values of GPs after calibration were very consistent (the average coefficients of variation of CCS was 0.1% of inter-batch, see Supplementary Data 3). In order to achieve the high accuracy annotation for bath process, we performed CCS calibration and ran QC samples to monitor the data quality within each batch. For each batch, CCS values were single-field calibrated using the Agilent tune mix under the same IM-MS parameters as the method for data collection of samples. Finally, in the data processing phase, CCS values, rather than drift times, were utilized for annotation to enhance the accuracy of the identifications.

MS2 spectra in Fig. 2c-e were acquired with DDA method under LC-MS/MS.

Comment #4. The overall data acquisition and data processing workflow were complicated. It would be much better to illustrate them with a main figure or supplementary figure.

Response: We thank the reviewer for the constructive suggestion, which has led us to enhance the clarity of our methodology through additional illustrative figures. We have now included figures that clearly depict both the data acquisition and data processing workflows used in our study. As shown in Supplementary Figure 12, for GP identification at fatty acyl composition level using data dependent LC-MS/MS without IM measurement, the GP subclass information was obtained from positive mode MS/MS and fatty acyl composition was obtained from negative mode MS/MS. For GP identification at sn-position resolved level operated on multiplexed IM-QTOF mode with the same LC gradient, as shown in Supplementary Figure 12b, the GP subclass information and CCS values were obtained from data independent LC-IM-MS/MS in positive mode, and fatty acyl information was obtained from negative mode data dependent LC-MS/MS. The combination of the carboxylate anion intensity from negative mode MS/MS together with the abundance (1st rule) and CCS distribution in the IM spectra (2nd rule) enabled the annotation of GP fatty acyl sn-isomers in the drift spectrum. Moreover, the GP subclass information obtained from data independent IM-MS/MS in positive mode were confirmed with the data dependent LC-MS/MS data. The comprehensive workflow for GP analysis in biological samples, including detailed data processing analysis, is now presented in Supplementary Figure 12c.

Moreover, to provide a more transparent and understandable outline of the sn-resolved GP library 4D database construction for our readers, we have also incorporated an overall workflow diagram as Supplementary Figure 23. As shown in Supplementary Figure 23, besides the lipid extraction, data acquisition steps are also included in the workflow for biological sample analysis. The workflow involves the peak extraction and annotation using MS-DIAL, IM Browser, and Qualitative Analysis. 4D data was summarized and used to generate transition list via Mass Profiler and LipidCreator.

[Redacted]

9

Supplementary Figure 12. Schematic of the LC-IM-MS/MS strategy for GP identification. a The outline of data dependent LC-MS/MS acquisition for lipid identification at fatty acyl composition level using QTOF-only mode without IM measurement. b The outline of integrated data independent LC-IM-MS/MS and data dependent LC-MS/MS acquisition for lipid identification at sn-position resolved level. c The overall workflow for GP analysis in biological samples including detailed data processing analysis.

[Redacted]

10

Supplementary Figure 23. The workflow for LC-high resolution-IM-MS/MS based 4D sn-resolved GP library construction.

Comment #5. Fig.1 showed the improved separation for equimolar mixtures with HRdm. How did HRdm perform while separating isomer mixtures with different ratios? Would HRdm generate artificial peaks and result in overreports structure information?

Response: We are grateful to the reviewer for raising this important question regarding the performance of HRdm in separating isomer mixtures at varying ratios. In response to this query,

we have conducted additional evaluations to determine the efficacy of HRdm in this context.

We have undertaken the evaluation of HRdm separation performance for isomer mixtures at different ratios. As shown in Supplementary Figure 4, the two-peak resolution (R_{pp}) of isomers at different ratios were at the range of 1.01-1.46, and the ratios of the peak area could be used to reflect their actual ratios of concentrations, demonstrating the separation performance. We also have updated our manuscript to include these findings.

Regarding the concern about the generation of artificial peaks and potential overreporting of structural information, we have addressed this issue in our study. As described in the response to Comment #1, the occurrence of artifacts was greatly decreased with the officially released HRdm 2.0 software compared with the beta version used by previous literature. When evaluating GP standards (Figure 1e,f and Supplementary Figure 4), we did not observe artifacts. When it comes to complex biological samples, the artifacts might be present in a very small number of GP precursors. The artifacts are usually of low abundance (<10% of the primary signal within each m/z extraction window) and located far enough away from the signal region of interest to

11

not interfere with spectral interpretation. IM drift times of genuine ion signal align across isotopic peaks due to the negligible contribution of mass to the ion mobility, artifactual peaks will not necessarily exhibit drift time alignment across isotopes. Through this understanding, we can effectively screen and exclude artificial peaks from our analyses.

[Redacted]

Supplementary Figure 4. Representative drift spectra of sn-position isomer mixtures at various ratios. PE 16:0/18:1 and PE 18:1/16:0 were mixed at molar ratios of 1:1 (a), 1:2 (b), 1:4 (c), 1:5 (d), and 1:10 (e).

Comment #6. Authors developed a combined strategy with MS2 fragmentation rules (the 1st rule) and CCS values from HRdm (the 2nd rule) to resolve GP isomer on sn-position level. To my knowledge, MS2 fragmentation rules (the 1st rule) were able to resolve most of the sn-position

12

GP isomers, especially IM largely improving the MS2 spectral precursor ion fraction and reducing chimeric spectra. How much did HRdm improve the isomer resolving based on the MS2 fragmentation rules? As authors only showed the comparison of the combined strategy with LC-MS/MS strategy in Fig. 2g, please also compared performances of combined strategy with those based on MS2 fragmentation rules.

Response: We thank the reviewer for the helpful suggestion and have conducted a comparative analysis to assess the contribution of the 2nd rule in the isomer resolution. We have updated Figure 2 and Supplementary Data 3 and also included the annotation rule in the Results.

We agree that MS2 fragmentation combined with ion abundance from high-resolution IM spectra (the 1st rule) were able to resolve most of the sn-position GP isomers. We have undertaken a comparison of MS2 fragmentation rules and the combined strategy. By matching the estimated abundance of sn-position isomers from relative intensities of fatty acyl chain carboxylate anions and abundance of IM-resolved peaks (the 1st rule), as shown in the updated Fig. 2f, a total of 452 GP species including 185 PCs, 155 PEs, 38 PSs, 45 PGs, 21 PIs and 8 PAs could be annotated. However, there were still circumstances when the two GP sn-isomers were of similar abundance (as shown in Fig. 2d, the mixture of PC 16:0/20:4 and PC 20:4/16:0). Besides, IM peak assignment was also challenging for lyso-GPs which only have 1 fatty acyl chain in either sn-1 or sn-2 position. In these circumstances, as we cannot simply annotate the IM peaks by rule 1, we used the combined strategy of both rule 1 and rule 2 to assign IM peaks to GP sn-isomers. Using the combined strategy, a total 498 GPs including 205 PCs, 173 PEs, 41 PSs, 46 PGs, 25 PIs and 8 PAs could be identified. The increase in identification numbers was largely attributed to the annotation of lyso-GPs with one fatty acyl chains, which identification required the MS2 for subclass and acyl chain information, and also need CCS values (the 2nd rule) for sn-position assignment. The comparison has been amended and was shown in Fig 2f, and the identification rule information was listed in Supplementary Data 3.

Comment #7. Line 241, Page 7: "As this prevailing trend has not been reported previously due to the limitations in IM R_p , we reasoned that this finding could be attributed to the higher gas-phase flexibility and freedom of the fatty acyl chain at the sn-1 located at the terminus of the glycerol backbone." I appreciate the rule and think it would be better if more evidence like computer-aided molecular modeling could support this rule.

Response: We are grateful to the reviewer's constructive suggestion regarding the need for additional evidence to support our findings on the gas-phase behavior of GP sn-isomers. In response to this, we have now conducted in silico structure simulation and theoretical CCS calculations for several representative pairs of GP sn-isomers. The predicted structures of these pairs of GP sn-isomers are detailed in Supplementary Figure 7, and their coordination information (Z-matrix) is also included in Supplementary Data 1.

13

Initially, we determined the protonation state of the gas-phase molecule and generated 500 different conformer structures using the RDKit toolkit (Greg Landrum 2013, 8, 31). Each conformer is then optimized with the Universal Force Field (UFF) and divided into several clusters for structurally distinct conformations. Potential structures of compounds are generated from the clusters and are optimized in Gaussian 16 (Gaussian 16 Rev. C.01, Wallingford, CT, 2016), using B3LYP Density Functional Theory (DFT) and the 3-21G basis set. The 6-31G+(d) basis set is then used to generate the energy-minimized structures. Theoretical CCS values for the compounds are calculated from their DFT-optimized geometries using IMoS v. 1.13 software (J. Comput. Phys. 2013, 251, 344-363; J. Phys. Chem. A 2013, 117, 19, 3887-3901), employing the trajectory method (TM) models with 3 orientations and 300000 nitrogen gas molecules per orientation. Other settings in IMoS, including the use of nitrogen as the drift gas, are kept default. The simulation results were consistent with our experimental measurement, and also suggested that GPs with smaller fatty acyl chains at sn-1 position generally showed lower CCS values than their sn-position isomers. This description of computational CCS calculation has also been added to Method section.

[Redacted]

14

Supplementary Figure 7. Energy-minimized structures and calculated theoretical CCS values of GPs. Representative simulated structures of PC 14:0/16:0 (a), PC 16:0/14:0 (b), PE 16:0/18:1(9Z) (c), PE 18:1(9Z)/16:0 (d), PS 16:0/18:1(9Z) (e), and PS 18:1(9Z)/16:0 (f).

Comment #8. Some technical points of CCS prediction were not clear. For the model training, MDs must be calculated from the exact structures including but not C=C location and geometry. However, the CCS values collected by authors were only assigned to sn-position level. How did authors assign an exact specific structure to a sn-position-level CCS values? And MDs and CCS values might be different if you select different C=C location and geometry isomers.

Response: We appreciate the reviewer's inquiry regarding the technical aspects of our CCS prediction model, particularly concerning the impact of C=C location and geometry on MDs and CCS values. In response to this query, we have revised the Methods section of our manuscript to include a more detailed explanation of our approach to calculating MDs and predicting CCS values. Specifically, our model considers each GP class in conjunction with the 35 most common FA chains found in mammals. For unsaturated FAs, we focused on their most abundant C=C double bond positions. This decision was made based on the understanding that C=C double bond positional isomers are indistinguishable using our current strategy. In our CCS prediction model, we calculated the MDs and predicted the CCS values of GPs with the C=C bonds in the cis (Z) configuration. This choice reflects the fact that most naturally occurring lipids in mammals predominantly contain C=C bonds in the cis configuration. To provide further clarity and reference for our readers, we have added a list of the 35 most common fatty acids considered in our model as Supplementary Table 3.

Comment #9. Line 594, page 17: "Since the SMILESs of lipids in LIPID MAPS do not contain the adducts, in this study, SMILES of adducted GPs were manually generated by combining the SMILES of head groups, adducts, fatty acyl chains in sn-1 and sn-2 in a uniformed format." is not clear, please give an example. How did the authors determine the ionization site of the adducts?

Response: We acknowledge the reviewer's request for clarification regarding the generation of SMILES strings for adducted GPs in our study. For the adduct sites, in the presence of an ammonium salt as a modifier in a lipid solution, singly protonated molecular species, [M+H]⁺, are readily formed in the positive ion mode for PC, PE, and PS classes. In the positive mode, the quaternary amine of protonated PC species was determined as the positive charge site, and the primary amine was determined as the positive charge site of protonated PE and PS species (Mass Spectrom. Rev., 2003, 22, 332-364). Some PG, PI, and PA species could be detected as ammonium adducts in the positive mode. Although many studies on ESI analysis of phospholipids assumed the alkaline adducted to the phosphate, their exact ammonium adducted sites are still

not clear (J. Chromatogr. B. 2009, 877 2673â2695; Mass Spectrom. Rev. 2003,22, 332â 364). In consideration of these factors, our study restricted the MD calculations, CCS, and RT predictions

15
to singly protonated PC, PE, and PS classes. This decision was made to ensure the precision of our model, given the unclear ammonium adduction sites for PG, PI, and PA, and the limited experimental data available for these classes.

Comment #10. I am curious about how much did the sn-position-resolve CCS prediction model improved compared with those tools developed on lipid specie level like LipidCCS and CCSbase.

Response: We appreciate the reviewer's curiosity regarding the performance of our sn-position-resolved CCS prediction model, especially in comparison with existing tools. Our model represents a significant advancement in terms of both depth and accuracy. While LipidCCS and CCSbase provide high accuracy for CCS predictions at the lipid sum composition and fatty acyl chain composition levels, they do not offer resolution at the sn-position level. This is a crucial distinction, as the sn-position of fatty acyl chains in GPs significantly influences their biological functions and interactions.

In our comparative analysis with LipidCCS, we found that using the MDs calculated from âCDKâ, which was also used by LipidCCS, produced a linear fit with an R2 of 0.9844 for our training set. However, this fit was not entirely satisfactory, primarily because these MDs had limited capability in differentiating sn-position isomers. Among the 221 MDs from CDK, only 15 showed differences between sn-position isomers. To address this limitation and enhance the precision of our prediction model, we expanded our approach to include MDs from both âCDKâ and âMordredâ. From an initial pool of 253 MDs, we selected all 221 MDs from CDK and an additional 32 Mordred MDs that demonstrated more than 0.1% difference between sn-position isomers. This selection is detailed in Supplementary Tables 6-7. The incorporation of these additional MDs significantly improved the precision of our sn-position-resolved CCS prediction model. In our internal validation, the model exhibited an excellent fit with an R2 of 0.9922 (as shown in Fig. 3d), and this high level of precision was maintained in the external validation with an R2 of 0.9925 (as shown in Fig. 3e). These results highlight the robustness and accuracy of our model, particularly in the context of sn-position-resolved lipid analysis. The enhanced precision and depth of our model make it a valuable tool for advanced lipidomics studies.

Comment #11. More about the extended library, the retention times were not included in the library. In fact, the retention times of lipids on a C18 column could be easily predicted with Quantitative Structure Retention Relationships (QSRR). The predicted RT would reduce false positive IDs to a large stand.

Response: We appreciate the reviewer's insightful suggestion regarding the inclusion of RT in our extended lipid library. We fully agree that RT would reduce false positive identifications and have developed a prediction model for RTs. These predicted RTs have now been incorporated into our extended 4D library, as detailed in Supplementary Data 7 and illustrated in Supplementary Figure 11. For the prediction of RTs, particularly considering that sn-isomers typically exhibit similar

16
retention behaviors during a short C18 LC gradient, we utilized molecular descriptors (MDs) calculated by ârcdkâ. Using the LASSO algorithm, we selected 51 MDs from a total of 221 for inclusion in our model (Supplementary Table 8). This model, built on an SVR-based machine learning algorithm, achieved a linear fit with an R2 of 0.9622 for the training set, as shown in Supplementary Figure 11a and b. We also checked our data and found the RTs of GPs identified from experimental samples could align with the predicted values.

The RT fluctuation for each GP subclass on a C18 column is influenced by both the length of the carbon chain and the degree of unsaturation. Generally, lipids with more double bonds or fewer carbon atoms tend to have shorter RTs (Nat Methods 2015, 12, 523â526; J. Chromatogr. A 2016, 1450, 76â85; Nat. Commun. 2021, 12, 4771). This was also a crucial consideration in our annotation process. To ensure robust validation of our identified lipids, we adhered to quality control requirements as detailed in Nat. Commun. 2021,12, 4771. For example, our experimental library demonstrated that PC classes with varying carbon atoms and double bond numbers showed excellent RT distribution, as depicted in Supplementary Figure 11c-d. Since that retention times can shift across different runs, we employed indexed retention time (iRT) calculators, constructed using data from both positive and negative modes with 10 isotope-labeled lipid standards from EquiSPLASH spanning the LC gradient in Skyline. All lipids in our brain library were assigned iRT values on a 0â100 scale using a Lowess regression. The retention times for mouse brain library lipids in experimental samples were then accurately predicted within a 2-minute window using these iRT calculators.

By combining these predicted retention times, the established retention rule for carbon atoms and double bonds, and the iRT values, we significantly improved the accuracy of our lipid

annotations. This comprehensive approach ensures high precision in our extended lipid library, effectively minimizing the occurrence of false positives in our lipidomic data.

[Redacted]

17

Supplementary Figure 11. Retention time prediction for GPs. a The internal validation of the machine learning algorithm predicted retention time of GPs (n = 335) using optimized MDs selected from CDK. b The prediction performance evaluated by external validation (n = 84). c-d, Retention times of PCs identified in mouse brain differed in either carbon numbers (c) or double bond numbers (d).

Comment #12. Fig. 4g showed the percentage of GP classes in different clusters. Would it generate more specific results with molecule percentage of different GP classes?

Response: We thank the reviewer for the insightful suggestion regarding the presentation of GP class data in our study. We have now added Supplementary Figure 13 and included more detailed discussions of findings from GP molecule percentage in Results section.

In order to show the alteration of each species between AD and WT groups across three functional regions, we conducted the hierarchical cluster analysis (HCA) to discriminate between AD and WT groups across three functional regions (Fig. 4f). Each species in this analysis was scaled (Z-score standardization) according to their abundance within a range from -3 to 3, as depicted by the color bar. In HCA heatmap, the differentiation of the 6 groups is determined by the scaled abundance rather than the actual molecular abundance. While this approach effectively demonstrates the overall abundance patterns, it inherently masks the specific molecular abundance information within each cluster.

18

We agree with the reviewer that analyzing the molecule percentage would provide a meaningful orthogonal supplement to the number percentage of GP classes across the five clusters (Fig. 4g). Although the number of GP species is consistent across all six groups in these clusters, the abundance of each GP may vary significantly within the clusters due to varying genotypes and spatial factors. To assess the molecule percentage of GP species in each cluster, we have compiled the results for all six groups, now presented in Supplementary Figure 13.

From the molecule percentage perspective, we observed that PE class is the most dominant in both cluster 2 and cluster 5, which aligns with the GP number percentage. Conversely, in cluster 1, the PC class, while dominating in terms of numbers (>50%), constitutes a minor molecule percentage (~30%). These findings have now been incorporated into the Results section.

The addition of GP molecule percentage figures and the ensuing discussion enable us to present a more detailed and nuanced understanding of the changes in GP composition associated with AD progression. This approach highlights the significantly altered species within each GP class, enriching our insights into the lipidomic alterations in AD.

[Redacted]

19

Supplementary Figure 13. Molecule percentage of GP classes in each brain region for WT and AD groups across clusters 1-5. Proportions of GP classes (%) in cluster 1 (a), cluster 2 (b), cluster 3 (c), cluster 4 (d), and cluster 5 (e) from Figure 4f are shown. The average concentration of each GP, calculated from biological replicates within the same group, was used.

Comment #13. Authors made a conclusion that uniform cPLA2 enzyme activities differed in different age and functional regions. Could authors confirm the results with other orthogonal methods like comparing the enzyme expression?

Response: We thank the reviewer for the valuable suggestion. We have now supplied the results of cytosolic phospholipase A2 (cPLA2) activity (Supplementary Figure 21) to confirm the results

20

of GP variations. We observed significant differences in cPLA2 activities in various brain regions and at different ages. For instance, in the hippocampus, the cPLA2 activities in AD mice at 3 months and WT mice at 8 months were markedly higher compared to WT mice at 3 months. This

finding aligns with existing literature reporting increased cPLA2 activity in AD brains (J. Neurochem. 2020, 154, 84-98; Acta Neuropathol. Commun. 2021, 9, 1-26). Moreover, these results are consistent with our observations of altered GP abundances and ratio of isomer pairs, such as notable increase of the ratio of LPC 18:0/0:0 to LPC 0:0/18:0 in AD mice compared to WT mice at 3 months. We also found significantly elevated cPLA2 activities in the AD group compared to WT in the cortex of 3-month-old mice. Interestingly, the changes in cPLA2 activities were less pronounced in the cerebellum, which is in line with previous studies suggesting that the hippocampus and cortex are more severely affected by A β plaques during advanced AD, while the cerebellum is relatively spared or minimally affected. These additional experimental results provide important orthogonal confirmation of our lipidomic findings. They demonstrate that variations in GP abundances are not only indicative of AD progression but are also reflective of age-related changes in enzyme activities. This comprehensive approach, combining lipidomic analysis with enzyme activity measurements, strengthens our conclusion that uniform cPLA2 enzyme activities differ significantly across different ages and functional regions of the brain.

[Redacted]

Supplementary Figure 21. cPLA2 activities in mouse brain tissue from different genotype and age. cPLA2 activities of hippocampus (a), cortex (b), and cerebellum(c).

Comment #14. Line 403, Page 12: "the clusters between WT at 3 months and AD at 8 months were clustered adjacent to one another" should be "WT at 8 months and AD at 3 months".

Response: We thank the reviewer for pointing this out. We have corrected this typo.

Comment #15. Supplementary Table 1, No. 20: "PA 15:0/18: (d7)" should be "PA 15:0/18:1 (d7)".

Response: We thank the reviewer for the careful examination. We have corrected the typo in Supplementary Table 1.

21

Reviewer #2 (Remarks to the Author):

In the article "Spatially and temporally probing distinctive glycerophospholipid alterations in Alzheimer's disease mouse brain via high-resolution ion mobility-enabled sn position resolved lipidomics" by Xu et al. the authors present an ion mobility spectrometry-mass spectrometry (IMS-MS) workflow in order to study variation of stereospecific numbering (sn) isomers of lipids. These sn-isomer changes have been linked to the dysregulation of the Land's cycle enzymatic machinery but corresponding lipids are notoriously difficult to analyze. The authors address this challenge by implementing Hadamard transform-assisted demultiplexing IMS-MS of complex lipid samples. By multiple ion injections at different preset offset times to generate different arrival time offsets, the authors remeasure the same sample multiple times. The multiplexed IMS data is disentangled by Hadamard transform to create an IMS result with considerably improved signal-to-noise as well as IMS resolution. This enables to almost baseline resolve sn-isomers of various lipid classes and perform quantitation on MS1 and MS2 level. The authors use this methodology to develop a large-scale CCS database training a corresponding bioinformatic model for confirmation of sn-identity without the use of authentic standards. The optimized data analysis pipeline and bioanalytical methodology is employed to quantify the spatial (cortex, hippocampus, cerebellum) and temporal (3 and 8 months) changes of lipid sn-isomers in brains of wild type and APP/PS1 (Alzheimer's disease model) mice. For some of the lipids, age and spatial sn-isomer alterations are significant.

This is a well-written manuscript that extends a novel IMS technique to sn-isomers of lipids. The analytical application is clearly interesting for a broad scientific audience as it "only" relies on multiplexed data acquisition and data analysis. So, in principle it should be possible to adapt the methodology in labs with access to drift time IMS instruments. Additionally, the method offers new perspectives in the study of the Land's cycle that has been understudied on the lipid level due to the difficulty to separate and quantitate sn-isomers. To further improve the manuscript the authors need to address the following issues:

Response: We sincerely appreciate the reviewer's positive evaluation and commendation of our work. We concur that our method can offer a valuable technique for in-depth lipidomics, contributing to this field in numerous ways. We are also truly grateful for the reviewer's constructive feedback and suggestions, which have undoubtedly helped strengthen our manuscript. In response, we have incorporated more data, conducted additional experiments, and addressed these comments in the point-by-point responses provided below.

- The main performance test of the presented IMS-MS method is based on the comparison with established PLA2 digests. Typically, MS read-outs have uncertainties in relative lipid intensities of 3 to 5 % and PLA2 digests itself (at least when going through the literature) have intrinsic uncertainties of 3 to 5% as well. Therefore, it is surprising that the comparison of a single PLA2 digest result with a single IMS-MS result of the same standard overlaps within a range of less than 1.5 %.

22

As all flowing conclusions are based on this single point comparison, I strongly recommend to perform replicates for the PLA2 and IMS-MS measurements (n=10-20).

Response: We are grateful to the reviewer for the detailed and constructive critique regarding the performance test of our IMS-MS method. In response to the reviewer's suggestion, we have undertaken a more thorough comparison of our IM-MS methodologies and PLA2 digestion by evaluating the sn-isomeric purity for some synthetic lipid standards. Additionally, to provide a comprehensive evaluation of the relative quantification performance of IM-MS for isomer mixtures with varying ratios, we prepared a series of mixtures of PE 16:0/18:1 (9Z) and PE 18:1(9Z)/16:0, each with different molar ratios. To provide a clearer and more detailed illustration of these results, we have included them in Supplementary Table 2. Our findings indicated that the isomeric abundance obtained using IM-MS aligns consistently with that obtained via PLA2 digestion. This comparison offers a comprehensive presentation of the quantification accuracy, thus providing a meaningful metric for evaluating the quantification performance and facilitating cross-comparisons between the two methods. We believe that these additional data and analyses significantly strengthen our manuscript by addressing the concerns raised and by offering more robust evidence to support our conclusions.

Supplementary Table 2. GP standards used for the comparison of quantitative performance of HRdm IM-MS with PLA2 digestion.

[Redacted]

- On a similar note: Please report coefficients of variation of IMS-MS measurements. This will help to demonstrate the robustness of the nice methodology presented in the manuscript.

Response: We extend our sincere gratitude to the reviewer for the valuable suggestion. In response to the recommendation, we have now incorporated additional data into our revised manuscript. Specifically, we have added the necessary mass error (in ppm), standard deviations (SDs) of both retention time and CCS values for each identified GP species. These additions can be found in Supplementary Data 2, 3, and 8. For instance, in Supplementary Data 3, which details the experimentally identified sn-position resolved GP species, we observed an average MS error of 0.2 ppm. The average CV of the retention time was 2.1%, and the average coefficients of variation (CV) of CCS was 0.1%. Moreover, we have also included the SDs of the drift time measurements in the figures or figure captions of Supplementary Figures 3, 4, 6, 7. These added details underscore the precision and consistency of our measurements, thereby demonstrating the robustness and reliability of our presented methodology.

- The authors demonstrate the capability to quantify significant changes of sn-isomers in complex samples. The manuscript would be improved when discussing potential follow-up experiments that will help to investigate the biological significance of the sn-isomer alterations.

24

Response: We thank the reviewer's suggestion to elaborate on potential follow-up experiments in our manuscript. In response, we have expanded the Discussion section to include a more detailed exploration of future experimental directions. Our data suggests that alterations in GPs occur in the early AD progression, likely involving complex interactions within lipid synthesis, transport, and degradation pathways. These changes are critical in the GP remodeling process. The lipid remodeling pathway involves removal of fatty acyls by the acyl hydrolases, phospholipase and acylglycerol lipase and reacylation, with alternative fatty acids, by acyl-CoA:lysophospholipid acyltransferases (Alz. Res. Therapy 2012, 4, 5; Cell. Metab. 2019, 30 (3), 407-408). The high levels of PUFA in the brain have led to studies centered on PLA2, enzymes responsible for cleaving the acyl groups at the sn-2 position of the phospholipids and resulting in production of PUFA and lyso-GP. Among the many subtypes of PLA2, many studies have implicated the role of cPLA2 in release of AA and increase in oxidative/nitrosative pathways in AD (Prog. Lipid Res. 2011, 50 (4), 313-30). We also evaluated the cPLA2 activity in WT and AD mouse of 3 functional regions at different ages to confirm the results of our methods

(Supplementary Figure 21). However, since there are many PLA2 subtypes and other enzymes in different brain cells involved in lipid remodeling, further investigation of changes in the activities of multiple enzymes like PLA2 and LPLAT in AD is required.

Another critical area for future research is the exploration of bioactive lipid mediators, potentially evoked by the release of PUFAs from GPs. These lipid mediators, including eicosanoids and resolvins, play significant roles in cell immune functions and inflammatory responses. Understanding their dynamics in AD could offer new insights for early diagnosis and therapeutic strategies.

Additionally, we plan to utilize imaging mass spectrometry in conjunction with high-resolution ion mobility and our established database for mapping the spatial distribution of sn-position resolved GPs across the entire brain. Collective studies may be possible to provide new perspective for early diagnosis, preventive or therapeutic options of AD.

Finally, we believe that our strategy could be of immense benefit to researchers interested in the structural identification of lipids in various tissues and disease models, offering new perspectives in lipidomics research.

- An extensive CCS database is created to analysis the data. Multiple groups have established CCS databases for lipids in the literature but often CCS values from different groups do not overlap within the margin of error. Why is the database created in this manuscript more relevant than other databases?

Response: We appreciate the reviewer's inquiry regarding the relevance of the collision cross section (CCS) database created in our manuscript, especially in the context of existing databases

with varying CCS values. CCS has emerged as an important resource for lipid characterization in biochemical research, and a significant molecular identifier, now incorporated to several chemistry databases including PubChem, for a wide range of molecules. The reported CCS values from different databases are differed largely due to different types of IM techniques and the choice of buffer gases that were adopted in their measurement. That is why the CCS values are recommended to be reported with these details.

In order to harmonize CCS values for confident identification, it is necessary to minimize the deviations of CCS between databases. Currently, most experimental CCS values were measured in nitrogen gas using commercial IMMS instruments like drift tube ion mobility spectrometry (DTIMS), trapped ion mobility spectrometry (TIMS), and traveling wave ion mobility spectrometry (TWIMS). Several research groups have used IM-MS to build libraries of lipid CCS values to facilitate the accurate identification. However, difficulties can be encountered when harmonizing measurements acquired using different instruments or disparate calibration methods. DTIMS, used in this study, is considered the "gold standard" for CCS values because DTIMS is able to determine reduced mobility, K_0 , from first principles. As a result of the trapped and ramping approach for TIMS, and the series of DC waves in TWIMS, calibration is needed to determine CCS values (Curr. Opin. Chem. Biol. 2018, 42:34-41; Nat. Methods 2023, 20, 1836-1837).

Calibration methods also play very important role for CCS value measurement. The stepped-field method in DTIMS is the only calibrant-independent approach, which allows to directly measure the CCS values of an analyte according to the Mason-Schamp equation. However, the stepped-field method is not compatible with the chromatographic separation. To overcome this challenge, a single-field method using calibrants of known CCS values (Agilent Tune mixture solution) was developed in DTIMS to measure the CCS values using one drift voltage (Anal. Chem. 2017, 89, 17, 9048-9055). An inter-laboratory evaluation reported that the calibrated single-field CCS method provided an average error of 0.27% for lipids. Currently, the single-field method is widely used for metabolomics and lipidomics. Some calibrant-dependent methods were also developed for TIMS and TWIMS to measure the experimental CCS values. To validate the TIMS CCS values, Siuzdak and co-workers compared the TIMS CCS values against DTIMS CCS values. The results illustrated an average CCS variation of $\pm 1.03\%$ for lipids, indicating good correlation and similar CCS values between the two IMS techniques for these molecule types (Nat. Methods 2023, 20, 1836-1837). However, in contrast to DTIMS, the accuracy of CCS values from TWIMS significantly depends on the structural similarity between the analyte and calibrant ions. Hines et al. has recently reported that the structurally mismatched calibrants lead to larger errors in the measurement of lipid CCS values. For example, lipid CCS values with tryptic peptide calibrants are systemically larger with an averaged relative errors of 6.4% compared to ones obtained from the DTIMS but reduce to <2% using the PC and PE standards (Anal. Chem. 2016, 88, 14, 7329-7336). Thus, the CCS values reported from different groups could have deviations and may be suitable for specific types of instruments. Currently, many research groups are devoted to improving

calibration approaches for standardization of the CCS determination of TWIMS (Anal. Chem. 2021, 93, 7, 3542â3550).

In this study, we used DTIMS which is considered the gold standard and high-reproducibility measurement for CCS values, and we employed the widely used single-field calibration method for our CCS calculation. Our CCS values are comparable with reported many database (Nat. Protoc. 2022, 17, 2415â2430; Anal. Chem. 2017, 89, 17, 9559â9566). For example, in our database, the CCS values of protonated PE 16:0/18:1 and PE 18:1/16:0 are 275.4 Å² and 277.5 Å², respectively. The values are consistent with CCS value of protonated PE 16:0_18:1 (276.6 Å², sn-position not resolved) in LipidCCS database, which falls between the two sn-position resolved isomers. Compared with other lipidCCS databases that reported CCS values for GPs at fatty acyl composition level, we first reported an in-depth lipid CCS library for GP species annotation at sn-position resolved level. Other high-resolution IM paradigms such as TIMS, TWIMS-based cyclic IM, and Structures for Lossless Ion Manipulation (SLIM) require prolonged drift/ramping time (up to 1 second) and/or narrowed IM window to achieve a similar degree of resolving power as that shown in HRdm (TrAC, Trends Anal. Chem. 2022, 157). These compromises would come at cost of sacrificing either throughput or IM coverage for large-scale untargeted lipidome profiling. In contrast, this DTIMS-based HRdm strategy demonstrates unique advantages for large-scale lipidome profiling in LC-IM-MS workflows as enhanced IM resolving power could still be achieved in a typical 60 ms IM scanning window for comprehensive ion collection. For this reason, DTIMS-based HRdm IMS is the ideal paradigm for sn-position-resolved lipidomic analysis and CCS database construction. In the future, standardizing the CCS measurement and expanding the quantity of empirical measurements by a joint effort from the lipidomics community would be an invaluable progress to support high-confident lipid characterization.

- On multiple occasions the authors mention that C20:4 can lose CO₂. Please point out that this is only true when fragmenting in negative-ion mode but not when using metal-adducts in positive-ion mode.

Response: We thank the reviewer for pointing this out. We have made the necessary revisions in the manuscript to explicitly mention that the loss of CO₂ from C20:4 is specific to fragmentation in negative-ion mode. We have clarified that this phenomenon does not occur when using metal-adducts in positive-ion mode, and have included a relevant reference to support this clarification (Anal. Chem. 2018, 90, 11486-11494).

- Some minor wording suggestions: (line 76) Uneven should be replaced by heterogenous or non-uniform; (line 86 and rest of article) mainstream, do you mean routine or standardized?

Response: We deeply appreciate the reviewer's meticulous attention and valuable suggestions. In accordance with the reviewer's recommendations, we have replaced the term "uneven" with "heterogeneous" to more accurately convey the variability in distribution. Furthermore, we have carefully reviewed the manuscript and replaced the term "mainstream" with "routine" throughout the document. We believe these alterations enhance the readability and accuracy of our manuscript, ensuring that the intended meanings are conveyed more effectively and accurately to our readers.

- The authors should cite the following recent article on the intermediates during CID of lipids (10.1021/jacs.1c06944 and 10.1021/jasms.1c00277) and additionally EID (10.1016/j.ijms.2022.116998) should be discussed and some recent reviews on this matter should be cited (10.1021/acs.analchem.1c00061 and 10.1007/s00216-021-03425-1).

Response: We are deeply thankful for the reviewer's detailed and insightful recommendations regarding the citation of recent articles. In response to the reviewer's suggestions, we have made the following amendments:

We have cited the recommended articles on the intermediates during CID of lipids, which utilize cryogenic gas-phase infrared (IR) spectroscopy and infrared multiple photon dissociation to characterize the sn-isomerism of lipids. These references have been incorporated as references 19 and 20 in our manuscript. The discussion of the EID technique has been added to the Main section of our manuscript to highlight its relevance and significance in lipid structural elucidation. Corresponding references have been included as references 23 and 24. Additionally, we have cited recent reviews on emerging MS technologies for lipid structural elucidation. These reviews

offer a broader perspective and context for our work and have been added as references 21 and 22.

Reviewer #3 (Remarks to the Author):

This manuscript reports generation of a library and methodology for analysis of imaging mass spectrometry data which is of great value to the field as a resource or methods paper.

They constructed a 4-demential database of glycerophospholipids in mouse brain with expanded collision cross-section (CCS) reference library predicted by machine learning. This allows automated profiling of GP with specific acyl chain sn-position assignment which is of critical value. They used a novel high resolution demultiplexing (HRdm) strategy for increased diffract ion mobility and ion mobility separation of sn-isomers allowing identification of acyl chain specificities from pooled mouse brain extracts (bulk).

28

For example, valuable information is reported regarding the relative abundances of carbon double bonds in glycerol phospholipids is reported regarding a lower collisional cross section value which may be due to higher gas-phase flexibility and freedom of the fatty acyl chain at the sn-1. They also found that from commercial standards, that lyso-GPs with acyl chain at sn-1 position generally showed smaller CCS values consistent with previous reports. These are highly impactful findings and will likely affect multiple future IMS studies.

However, the biological ramifications of this study are limited based on the following issues.

Response: We truly appreciate the reviewer's positive evaluation of our work and recognition of the potential impact of our methodology and findings. We agree that our strategy can offer a valuable technique for in-depth lipidomics, and the results presented in this study can provide guidance for future studies in the field and serve as a good resource for readers with different backgrounds. We also expect the breakthrough in analytical methods together with the disease-related lipid biomarker discoveries would inspire future studies that require more precise lipid structural identification. We are grateful to the reviewer for the valuable comments and thorough examination of our manuscript, which have helped improve various aspects of our work. We have carefully addressed the issues raised by the reviewer, augmented the discussion of biological ramifications, and have made the necessary corrections and clarifications as outlined below.

1. The three functional regions of the AD mouse brain were not justified as to why they are under investigation.

Response: We sincerely appreciate the reviewer's inquiry regarding the rationale behind our choice of functional brain regions. We have augmented the Results section to provide more biological perspective about the selection of functional regions for AD study.

AD is a progressive neurodegenerative disorder with histopathological hallmarks of $A\beta$ -amyloid ($A\beta$) plaques and neurofibrillary tangles in the brain (Cell 2015, 160 (6), 1061-71). Extensive characterization of AD progression in human has demonstrated that the hippocampus and cortex are heavily affected by the $A\beta$ plaque pathology while the cerebellum is spared or minimally affected (J. Alzheimers Dis. 2016, 49 (1), 13-9; Acta Neuropathologica 1991, 82 (4), 239-259; Neurology 2002, 58 (12), 1791). As such, the cerebellum is often used as a spatial negative control within studies targeting AD-neuropathology.

Furthermore, the hippocampus and cortex have been the focus of numerous studies investigating lipid metabolism associated with AD. Notably, Schultzberg and co-workers investigated GP and lipid mediators in these brain regions of AD mice for therapeutic exploration of AD (Commun. Biol. 2022, 5, 245; Acta. Neuropathol. Commun. 2021, 9, 116).

29

In our study, the profiling of GPs in these regions aligns with these previous findings. We have updated Supplementary Figure 15 to include additional principal component analysis (PCA) of the hippocampus. Our results demonstrate that mice of different ages and genotypes can be distinctly differentiated based on quantified GPs in the hippocampus (Supplementary Figure 15a) and cortex (Supplementary Figure 15b). However, the PCA of GPs in the cerebellum (Supplementary Figure 15c) showed an overlap between WT mice at 8 months and AD mice at 3 months, indicating that GP profiles in the hippocampus and cortex are more sensitive markers

for monitoring age-related AD progression than those in the cerebellum. By incorporating these details into our manuscript, we aim to provide a comprehensive understanding of the biological significance of selecting these specific brain regions in our study.

[Redacted]

Supplementary Figure 15. Principal component analysis (PCA) of AD and WT in hippocampus (a), cortex (b), and cerebellum (c).

2. The mouse model was not justified or compared to other potential models to be used.

Response: We thank the reviewer for this valuable comment regarding the justification of our choice of mouse model for AD research. Recognizing the importance of this aspect, we have now included a detailed discussion in the main text of our manuscript to address this point.

Animal models are critical for understanding disease pathogenesis. There are more than 100 different genetically engineered mouse lines reported to capture some aspect of AD. However, none of them encompasses the entire spectrum of human AD pathology (Mol. Neurodegener 2017, 12 (1), 89). In this study, we used APP^{swe}/PS1^{dE9} transgenic mice to investigate GP alteration in AD progression. The animals express mutant forms of APP (Mo/HuAPP^{swe} with the K595N/M596L mutation) and PS1 (deletion of exon 9). They are characterized by an early-onset of AD and age-associated increase in A β -levels followed by A β deposition and morphological alterations (Cell 2015, 160 (6), 1061-71); Mol. Neurodegener 2017, 12 (1), 89; EMBO J. 2017, 36 (17), 2473-2487; Neuroimage Clin., 2017, 15, 581-586; Neurobiol. Dis., 2006, 24 (3), 516-24). The APP^{swe}/PS1^{dE9} mouse model is among one of the most commonly used overall for AD study and,

30

with more than 2,500 papers (PubMed search engine), probably the most used when studying AD plaque pathology.

Additionally, numerous studies have employed the APP/PS1 mouse model to investigate lipid alterations in AD, further substantiating its relevance and utility in lipidomic research (Nat. Cell. Biol. 2016, 18, 1065-1077; Proc. Natl. Acad. Sci. U.S.A. 2021, 118 (33), e2102191118; Redox. Biol. 2021, 41, 101947; J. Chromatogr. A 2022, 1676, 463196; Neurobiol. Dis. 2009, 33 (3), 482-98; J. Nutr. Biochem. 2014, 25 (2), 157-69; Brain Behav. Immun. 2020, 83, 87-111). By choosing the APP/PS1 model, we align our study with a well-established framework within the AD research community, thereby ensuring the relevance and comparability of our findings.

3. The biological ramifications of the altered GP content in these regions was not discussed.

Response: We greatly appreciate the reviewer's insightful comments regarding the need to discuss the biological ramifications of altered GP content in mouse brain regions. In response, we have substantially augmented the discussion in our manuscript to address the spatial diversity of GP and age-associated temporal GP alterations in the AD mouse brain. We have summarized the biological ramifications into the following key points:

a) Alteration in total GP abundance: Our analysis at the age of 3 months revealed a decrease in the total abundance of each GP class in the AD group compared to the corresponding WT groups across all three brain regions (Fig. 4e-f). The reduction of GP, a recognized indicator of aberrant GP homeostasis of AD, is in agreement with previous reports that membrane damage is a core feature of AD. Significant GPs loss in brain as a result of abnormal membrane repair in AD brains ultimately leads to synaptic loss and the aggregation of A β peptide (Acta. Neuropathol. Commun. 2021, 9 (1), 116; Neurochem. Res. 2001, 26 (7), 771-782).

b) Downregulation of PUFA-containing GPs: We observed a significant decrease in PUFA-containing GP species in AD and elder mice. Docosahexaenoic acid (DHA, 22:6n-3), which is the main PUFA in brain tissues essential for normal brain development and function, has been reported to attenuate A β burden, decrease tau phosphorylation and reduce neuroinflammation, the three main pathological hallmarks of AD (Proc. Natl. Acad. Sci. U.S.A. 2018, 115 (49), 12343-12345; Curr. Clin. Pharmacol. 2015, 10 (3), 222-41). GPs play a crucial role in DHA transport to the brain, and PE constitute the major storage form of DHA in the brain and can facilitate the signal transduction of bioactive mediators and protect PUFAs, especially DHA, from oxidation in gray matter (Int. J. Mol. Sci. 2022, 23 (7)). In three brain regions of AD groups, a drastic decrease in DHA-containing PE species, e.g. PE 16:0/22:6 and PE 18:0/22:6 (Fig. 4h, i) was observed. This depletion potentially impacts synaptic transmission and brain function, as DHA-containing GPs

are crucial for maintaining membrane fluidity, neuronal survival, synaptic neurotransmission, and

31

regulation of neuroinflammation, suggesting that a reduction in DHA-containing PE might be an indicator of AD and aging (Prog. Lipid Res. 2011, 50 (4), 313-30).

c) Alteration in GP sn-position isomer ratios: GP sn-position information has long been overlooked in routine lipidomic analysis using CID-MS/MS. In this study, at the sn-position isomer resolved level, the ratio of sn-isomer pairs showed significant alterations between WT and AD groups (Fig. 5m). For example, the ratios of LPC 18:0/0:0 to its sn-isomers, were significantly elevated in AD mice compared to WT mice at both 3 months and 8 months in hippocampus. A portion of lyso-GP is the product of degradation of GP catalyzed by PLA. The increase of relative LPC 18:0/0:0 suggested that the sn-2 PUFA-containing GPs were more degraded than their corresponding isomers. The increase of relative LPC 18:0/0:0 suggested that the sn-2 PUFA-containing GPs were more degraded than their corresponding isomers. Thus, the significant deviation in ratios of sn-isomers was potentially correlated with the increased phospholipase A2 (PLA2) activities and also indicated a dysregulation of GP metabolisms in AD; the results were congruous with previous reports. The abnormal lipid metabolism was attributed to A β which was reported to stimulate astrocytes to release cytokines and nitric oxide, resulting in activating cPLA2 cascades leading to enhanced COX-2 expression, as well as upregulation of oxidative and inflammatory responses in AD (Nat. Genet. 2006, 38, 752-754; J. Lipid Res. 2004, 45, 205-213; Circulation. 2022, 146, 372-379). We further linked these changes to the lipid metabolism pathway. To validate this finding, we further evaluated the cPLA2 activities in WT and AD mouse across 3 brain regions (Supplementary Figure 21). The results of cPLA2 activities also demonstrated significant increases in AD 3 compared with WT 3 in hippocampus, which was consistent with the results by our LC-IM-MS/MS strategy. Simultaneously, some 1,2-diacyl GP isomer pairs, including the ratios of PC 18:0/20:4, PE 16:0/20:4, PS 18:0/22:4 to their sn-isomers, were significantly elevated in 3-month AD mice compared to 3-month WT mice, and most of the ratios also showed significant changes at 8 months. Considering with result of the increased cPLA2 enzyme activity, suggesting a potential increase of LPLATs in AD, which was consistent with previous study. Herein, our GP profiling with an in-depth structural characterization could also serve as accurate and sensitive indicators for dysregulation of GP metabolism.

d) Aberrant changes in the ratio of GP/lyso-GP: With the capability to discriminate sn-position, we also mapped altered ratios of GP/lyso-GP for multiple species. The ratio of sn-1 C18:0-containing PC/LPC 18:0/0:0 was significantly decreased in the AD hippocampus at 8 months, the results were consistent with previous report that the decreased ratio of PC/lyso-PC in AD might be highly useful as a novel plasma biomarker for the diagnosis of early dementia (Alzheimers Dement (Amst).2015, 1, 295-302). However, the ratio of PE/LPE has rarely been reported. We found that the ratios of sn-1 C16:0-containing PE/LPE 16:0/0:0 and sn-1 C18:0-containing PE/LPE 18:0/0:0 were significantly elevated in 3-month AD compared to 3-month WT in the hippocampus, considering the increased cPLA2 enzyme activity in AD, also suggesting a potential increase of LPE acyltransferase (LPEAT) in hippocampus in AD. This was consistent with the results of the ratio of some intact GP isomer pairs. Taken together, these data indicate that the altered activities of

32
enzymes in GP metabolic pathways could be attributed to AD and aging. Despite our advances, a more thorough biological exploration and understanding of the mechanisms and importance of the pathways and enzymatic activities responsible for remodeling of GPs in AD necessitates further research.

These discussions provide a detailed account of how the spatial and temporal diversity of GPs in the mouse brain can serve as molecular signatures of AD progression. While our findings mark significant advances, we acknowledge that a more in-depth understanding of lipid functions and their impact on disease progression is necessary. We believe our strategy offers valuable insights for researchers seeking to conduct detailed structural identification of lipids in various tissues and disease models.

4. Biological prioritization of 592 lipid changes and pathways are not discussed.

Response: We sincerely appreciate the reviewer for their critical evaluation and constructive comments regarding the biological prioritization of lipid changes and correlated pathways in our study. In response, we have expanded the discussion in our manuscript to encompass a more comprehensive analysis of lipid changes and associated remodeling pathways.

Initially, we conducted hierarchical clustering and PCA analysis using the abundance of sn-position-resolved GPs. This analysis successfully discriminated AD and WT groups at different ages across all three brain regions in hippocampus and cortex, highlighting the spatial

heterogeneity of sn-position-defined GPs and their distinct functions (Fig. 4f and Supplementary Figure 15).

To provide a comprehensive comparison of GP changes across different genotypes and ages in the three mouse brain regions, we performed volcano plot analysis based on GP abundance and added to the Supplementary Information as Supplementary Figure 16 and Supplementary Data 12. We observed a substantial decrease in many GP species in the AD group at 3 months in all regions, especially PUFA-containing GPs, including PC 16:0/20:4, PE O-16:0/20:4, PC16:0/20:4, PC 18:0/20:4, PE 18:0/20:4, PS 18:0/20:4, PE 18:0/22:6, PS 18:0/22:6, PG 18:0/22:6 PI 18:0/22:6 (Fig. 5c-l, Supplementary Figure 17, 19). DHA has been reported to attenuate A β burden, decrease tau phosphorylation, and reduce neuroinflammation, which are the three main pathological hallmarks of AD (Curr. Clin. Pharmacol. 2015, 10 (3), 222-41). Notably, we found a significant reduction in DHA-containing GPs in the AD group at 3 months and WT groups at 8 months compared to WT at 3 months, which aligns with the role of DHA in attenuating key AD pathologies. Furthermore, some lyso-GP also exhibited a notable decrease, the change of the most abundant lyso-GP species was discussed as well.

33

Another notable finding comes from the alteration in relative abundance ratios of GP sn-isomeric pairs as well as GP/ lyso-GPs, and their correlation to specific enzyme activities in GP remodeling pathways. Although most GPs were downregulated in AD groups, including the simultaneous decrease of many pairs of GP isomers, we discovered their abundance ratios could be significantly elevated, e.g., the ratio of LPC 18:0/0:0 to LPC 0:0/18:0 was significantly elevated in AD mice of hippocampus (Fig. 5m), we further linked these changes to lipid metabolism pathway. The complex lipid remodeling pathway involves removal of fatty acids by PLAs, and reacylation by acyl-CoA synthases, transacylases, and lysophospholipid acyltransferases (LPLATs). The increase of relative LPC 18:0/0:0 suggested that the sn-2 PUFA-containing GPs were more degraded than their corresponding isomers, potentially correlated with the increased PLA2 activities. The finding was validated by the cPLA2 activities (Supplementary Figure 21). Simultaneously, some 1,2-diacyl GP isomer pairs, including the ratios of PC 18:0/20:4, PE 16:0/20:4, PS 18:0/22:4 to their sn-isomers, were significantly elevated in 3-month AD mice compared to 3-month WT mice, considering the increased cPLA2 enzyme activity, suggesting a potential increase of LPLATs in AD since Lyso-GP released by PLA2 catalyzed reaction could be acylated into 1,2-diacyl GP by LPLATs to maintain neural membrane composition, which was also reported by previous studies (J. Lipid Res. 2004, 45, 205-213; J. Neurochem. 1998, 70, 786-793). Moreover, we found the ratios of sn-1 C16:0-containing PE/LPE 16:0/0:0, and sn-1 C18:0-containing PE/LPE 18:0/0:0 were significantly elevated in 3-month AD compared to 3-month WT in the hippocampus, considering the increased cPLA2 enzyme activity in AD, also suggesting a potential increase of LPE acyltransferase (LPEAT) of AD in hippocampus. Herein, our data suggested that the alteration of 1,2-diacyl GP molecules in AD mice was the consequences of changes in multiple enzyme activities involved in complicated lipid metabolism.

As GP sn-position information has always been overlooked in routine lipidomic analysis using routine CID-MS/MS, some potential disease mechanism could not be obtained from the routine lipidome characterization. Our study identified the GP structure at the sn-position isomer resolved level, showcases the importance of developing isomer-resolved strategy, which can provide mechanistic insights into lipid metabolism.

[Redacted]

34

Supplementary Figure 16. The significant GP changes in AD and aged-mice of different brain regions. Volcano plot showing the most significantly ($p < 0.05$, fold change > 1.5) decreased and increased GP species in AD 3 vs WT3 of hippocampus (a), AD 8 vs WT 8 of hippocampus (b), WT 8 vs WT 3 of hippocampus (c), AD 8 vs AD 3 of hippocampus (d), AD 3 vs WT3 of cerebellum (e), WT 8 vs WT 3 of cerebellum (f), AD 3 vs WT3 of cortex (g), WT 8 vs WT 3 of cortex (h).

35

5. Representative images are included for only 2 ions (Fig h, i) not for other regional ions of interest in hippocampus, cortex or cerebellum. Fig1 h, i coronal sections are shown while in all other figures, sagittal section is represented in schematics (Figure 4 and supplemental figures)

Response: We appreciate the reviewer's concerns and hope to clarify the caption of the heatmaps showing in Figure 1h & i as well as the schematics in Fig. 4 and Supplementary Figures. In Figure 1, we utilized commercially available isomer-pure lipid standards with determined sn-

positions to evaluate our methods. We clarify that Figures 1h and 1i present ion mobility-mass spectrometry (IM-MS) heatmaps, showing drift time to m/z for these isomer pairs. These heatmaps demonstrate the physicochemical properties of the GP isomers and our strategy's capability to differentiate GP sn-positions. It is important to note that these IM-MS heatmaps do not represent imaging mass spectrometry heatmaps that would show the relative abundance and spatial distribution of molecules within brain regions. To supplement the GP standards data, we have included IM-MS heatmaps of representative GP sn-position isomers identified in the AD mouse brain in Supplementary Figure 10. This addition provides a broader scope of data beyond the GP standards. Fig. 1h and 1i are not depicting coronal sections but are IM-MS heatmaps of equimolar mixtures of PE 16:0/18:1(9Z) and PE 18:1 (9Z)/16:0 (Fig. 1h), PS 16:0/18:1(9Z) and PS 18:1 (9Z)/16:0 (Fig. 1i) isomers, respectively. The representations of mouse brains in our manuscript are consistently sagittal sections. In Figure 4a, the highlighted sections on the mouse brain cartoon indicate the regions (hippocampus, cerebellum, and cortex) that were dissected from the whole brain for lipid liquid-liquid extraction and LC-IM-MS/MS analysis. To avoid confusion, we have revised the figure captions to provide more detailed and clear descriptions.

6. No coordinated and defining atlas information for hippocampal, cerebellum or cortical identification are included.

Response: We are grateful for the reviewer's suggestion regarding the inclusion of coordinated and defining atlas information for hippocampal, cerebellar, and cortical regions in our study. In response, we have now added Supplementary Figure 22 to our revised manuscript, which illustrates the dissection process in detail. This figure is designed to provide clear visual guidance on the isolation of specific brain regions. Furthermore, we would like to highlight our team's extensive expertise in the dissection and isolation of different mouse brain areas. Our proficiency in this area is well-documented in our publication record, which includes a wealth of research in neuroanatomy and related methodologies (Commun. Biol., 2022;5(1):173. Brain 2022;145(2):500-516; Brain Commun., 2022;4(1):fcac002; Anal. Chem., 2019;91(20):12942-12947; J. Exp. Med., 2016;213(7):1267-84; J. Neurosci. 2014;34(20):6772-89; Aging Cell 2014;13(3):449-5; Brain Res. 2013;1508:1-8; Aging Cell 2010;9(2):174-90). For this particular study, we employed a well-characterized and widely accepted technique for isolating brain regions based on their anatomical locations. This approach is founded on established protocols

36

(Methods Mol. Biol. 2015, 1280, 61-74; Neuroproteomics. Neuromethods, 2011, 57 (https://doi.org/10.1007/978-1-61779-111-6_2)) and is referenced in the Methods section.

[Redacted]

Supplementary Figure 22. Schematic representation of mouse brain dissection. The whole brain is removed from the skull (a) and placed on a dissection surface. Both the olfactory bulbs (OB) and the cerebellum (CB) are removed prior to sagittal separation of the two hemispheres (b). The single hemisphere is rotated to expose the medial surface (c) prior to removal of the striatum (S), thalamus (T) and midbrain (MB) to expose the hippocampus (d). The hippocampus (HP) and cortex (CTX) are then separated (e). The figure was generated with BioRender.

7. It is unexpected that cerebellum which is generally considered spared in AD and hippocampus which is generally considered susceptible to AD pathology have the same number of differential lipids (ie., Fig 4.d.).

Response: We appreciate the reviewer's insightful comments. We have carefully considered your observation and have added a discussion to address this aspect of our findings. The cerebellum is generally considered to be spared or minimally affected by the plaque pathology characteristic of human Alzheimer's disease (see our responses 1&2 to the reviewer). However, it is important to note that all AD-related mouse models, including the APP/PS1 mice used in our study, are transgenic and carry familial AD-associated mutations. In our case, these mutations are driven by the PrP promoter, which is active in CNS neurons across the brain and spinal cord. Despite the limitations inherent in these mouse models, the cerebellum might still serve as a valuable negative control for AD-related neurodegenerative events.

Similar findings regarding the number of identified lipid species in the hippocampus and cerebellum have been reported in previous studies (Nat. Commun. 2021, 12, 6021). This underscores that our findings are in line with existing literature and not unprecedented. While the number of lipid species identified may be similar between the hippocampus and cerebellum, there are significant differences in the abundance of each GP class and specific GP species

between these regions, as shown in Fig. 4e, f. As shown in the HCA Heatmap (Fig. 4f), GPs in

37

cluster 2 were expressed at the highest levels in the WT hippocampus and the lowest level in the AD cerebellum. GPs in cluster 5 exhibited very high levels in the WT cerebellum and the lowest levels in the AD hippocampus. Quantitative analysis across three regions revealed a marked disparity in the abundance of GPs in the hippocampus and cerebellum. GPs being major components of cellular membranes are expected to have a somewhat similar composition across different brain regions. However, the unique GP species that are specific to either the cerebellum or hippocampus account for a small fraction (less than 1%) of the total identified GPs. The fact that these unique species numbers happen to be similar is a coincidence rather than a reflection of pathological similarity. We have included these explanations in our revised manuscript to provide a clearer interpretation of results and understanding of the lipidomic landscape in different brain regions of AD models and to address the valid concerns raised.

8. In methods, mice were euthanized in CO₂ chamber prior to brain extraction which is highly likely to affect lipid metabolism within several seconds of anoxic condition even prior to death of neurons and glia. These results cannot be considered biologically relevant changes in the mouse brain in vivo due to genetic or age related effects.

Response: We sincerely appreciate the reviewer for their insightful comments regarding the potential impact of the CO₂ euthanasia method on lipid metabolism in our study. We fully understand the reviewer's concern. CO₂ euthanasia is a widely accepted and commonly used method for euthanizing laboratory rodents due to its effectiveness and humane approach (Basic Clin. Pharmacol. Toxicol. 2017, 121 (2), 113-118; Life Sci. 2021, 284, 119916). Its application extends to various studies, including those investigating brain lipid metabolism (Nat. Chem. Biol. 2023, 19 (2), 187-197; Sci. Adv. 2019; 5: eaax7142; Myelin 2018; J. Neurochem. 2022, 161 (2), 112-128; J. Histochem. Cytochem. 2019, 67 (3), 203-219).

A major concern is that the hypercapnia/ischemia caused by CO₂ euthanasia would activate the lipases to induce the release of fatty acids from phospholipids, resulting in an increase in free fatty acids and oxidized fatty acids (J. Lipid Res. 2019, 60 (3), 671-682). Notably, studies have shown that CO₂ exposure does not significantly interfere with the composition of free fatty acids in the rat brain when compared to decapitation (J. Pharmacol. Toxicol. Methods 2015, 71, 13-20). Additionally, the trace amount of total oxidized fatty acids in the brain (around 0.1 nmol/g), compared to the total brain fatty acids in brain phospholipids (around 42000 nmol/g), is relatively low and would not significantly influence the overall findings of phospholipid alterations in different genotype and age-matched mice (J. Neurochem. 2020, 152 (2), 195-207). In our study, we have tried to minimize the euthanasia to tissue collection time and ensured that all mice were euthanized under the same conditions to minimize potential variations. This consistency is crucial for the reliability of our comparative analysis among different genotypes and age-matched mice.

We have included a statement in the Discussion section regarding the possibility of postmortem changes in lipids, including oxidation, despite our adherence to strict protocols for euthanasia,

38

dissection, and collection. Going forward, we recognize the importance of systematically comparing the effects of different euthanasia methods on the brain lipidome. Such studies will provide valuable references for the research community and help in understanding the extent to which these methods might influence experimental outcomes. We are committed to employing rigorous methodologies and acknowledge the necessity of continuously evaluating and refining our techniques to ensure the highest degree of scientific accuracy and relevance.

9. In methods it is not clear if whole brain was frozen for imaging MS analysis or only the dissected regions. No whole brain image is shown for representative ions.

Response: We appreciate the reviewer's inquiry regarding the specifics of our brain tissue preparation process. We have made the necessary revisions to address and clarify this. In our study, we dissected the mouse brain to isolate specific regions - the hippocampus, cerebellum, and cortex. These dissected regions were then promptly frozen prior to lipid liquid-liquid extraction for subsequent liquid chromatography-ion mobility-mass spectrometry (LC-IM-MS/MS) analysis. It is important to clarify that we did not intend to perform imaging MS analysis of the brain in this study, but rather performed the LC-IM-MS/MS analysis of the dissected regions to demonstrate regional differences. To clarify this process, we have updated the Methods section of our manuscript to include a more detailed description of the brain tissue preparation and extraction process. The revised text now reads: "A modified Folch method was employed for lipid extraction from brain hippocampus, cerebellum, and cortex samples."

Approximately 10 mg of brain tissue from wild-type (WT) and APP/PS1 (AD) mice was weighed and dissolved in 0.5 mL of cold PBS. Brain tissues were homogenized with a probe sonicator in an ice water bath with a pulse of 10s on and 10s using 60 W energy for 20 cycles and subsequently mixed with 0.5 mL cold methanol and 1 mL cold chloroform for liquid-liquid extraction. This modification is intended to provide a clearer understanding of our sample preparation process. To enhance the clarity of our methodology, we have included Supplementary Figure 22, which provides a schematic representation of the mouse brain dissection process. This visual aid should help readers better understand the specific regions we focused on in our study.

Our research primarily utilized LC-IM-MS/MS to identify and quantify GPs in the dissected brain regions. This approach allowed us to investigate alterations in GP profiles associated with Alzheimer's disease in mice. The differences in GP abundance between Alzheimer's disease and wild-type mice across different ages are detailed in Figures 4h-m and 5c-q, as well as Supplementary Figures 17-20. It should be noted that our method involved homogenizing the brain tissues, which differs from the typical approach in imaging MS that usually employs non-homogenized, sliced tissue. Therefore, presenting our results as a whole brain image would not be aligned with our current methodology. While the current study did not involve whole brain imaging MS, we plan to incorporate this spatial analysis technique in future research. By utilizing a high-resolution ion mobility strategy with our established database, we aim to map the

39

distribution of sn-position resolved GPs across the entire brain. This future direction has been noted in the Discussion section of revised manuscript.

10. Statistics are reported in figures legend as ANOVA, but no false discovery rate is described. These high-dimensional data need FDR correction. Statistical methods are not described explicitly in the methods. FDR correction should be used even if a higher cut-off criteria is used. This should be clearly described in methods in addition to the figure legends.

Response: We thank the reviewer's careful reading of our manuscript and providing us with the keen scientific insight. Abundance of GPs were compared using one-way ANOVA with correction for multiple comparisons using the two-stage linear step-up procedure of Benjamin, Krieger and Yekutieli at a 0.05 false discovery rate (FDR). This has been clearly described in methods and figure legends (Figure 4-5, Supplementary Figure 14, 17-21).

11. Supplemental figures are not labeled with figure title or figure legends.

Response: We thank the reviewer for pointing out this issue. We have revised the titles of all Supplementary Figures to ensure they accurately reflect the content and purpose of each figure. Additionally, we have added legends for each part of Supplementary Figures 17-20.

12. Table S4 and S5 are not clearly labeled and may be missing from manuscript.

Response: We thank the reviewer for the diligent review and careful examination. We now have included the detailed description of these Supplementary Tables in the revised manuscript.

13. This would be appropriate for a methods/resources journal.

Response: We really appreciate the effort the reviewer dedicated to reviewing our manuscript and the commendation of our work. We hope to offer some perspectives on why we believe Nature Communications is an appropriate platform for our research.

Nature Communications is a multidisciplinary journal dedicated to publishing high-quality research in all areas, and has been a very powerful platform and well reputed forum for its high impact and broad readership. The field of lipidomics, particularly the structural characterization of lipids, is of great importance in biology. Over the past decade, significant research efforts have been devoted to developing analytical methods for the detailed structural elucidation of lipids sn-position, C&C location and geometry isomers (Nat. Commun. 2023, 14, 4263; Nat. Commun. 2022, 13, 2652; Nat. Commun. 2020, 11, 375; Nat. Commun. 2019, 10, 79; Angew. Chem. Int. Ed. 2018, 57, 10530-10534; Angew. Chem. Int. Ed. 2022, 61, e2022070; Angew. Chem. Int. Ed. 2022, e202215556; J. Am. Chem. Soc. 2021, 143, 14622-14634). Our work would also contribute to this

40
important and rapidly evolving research field. Moreover, ion mobility spectrometry (IMS) has been used to support lipidomics applications to facilitate the separation and the identification of complex mixtures of analytes including biologically relevant structural isomers (Nat. Commun. 2023, 14, 937; Nat. Commun. 2023, 14, 1535; Nat. Methods 2023, 20, 1836-1837; Nat. Protoc. 2022, 17, 2415-2430; Nat. Commun. 2019, 10, 985; Nat. Commun. 2021, 12, 4343). These recent studies greatly accelerated advances in the lipidomics. As this emerging field of global structural

lipidomic analyses is crucial to fully understand cellular metabolism, robust and powerful strategies are needed to enable their widespread application in biological studies.

In this study, leverages the advancements in ion mobility spectrometry (IMS) in lipidomics. We introduce a high-resolution IM-MS-based four-dimensional (4D) lipidomics strategy, complemented by a machine learning-empowered library. This novel approach of High-Resolution Demultiplexing (HRdm) strategy significantly enhances the separation resolution of IMS, facilitating robust and rapid analysis of GP sn-isomers in millisecond without any instrumental modifications and chemical derivatization. Delving deeper, we have constructed a pioneering 4D database with 498 GPs at sn-position resolved level, sourced from both standards and mouse brain lipid extracts. This was complemented by an expansive 4D library of 2,500 GPs, constructed via a robust machine learning-based predictive approach. This database and library are groundbreaking contributions to the field and are instrumental for researchers in biology and analytical sciences, offering valuable resources for detailed lipid structural identification. Our study not only delves into the spatial and temporal diversity of GPs in the mouse brain but also highlights potential molecular signatures of Alzheimer's disease progression. This offers new insights into disease pathology associated with dysregulated lipid metabolism and could aid in uncovering potential lipid biomarkers for diseases.

The broad applicability of our findings to lipidomics, biological mass spectrometry, neuroscience, and chemical biology makes our study appealing to a wide range of researchers engaged in both fundamental and translational research. In light of these considerations, we believe that Nature Communications is an appropriate forum to publish our study. Its broad scope and diverse readership align well with the multidisciplinary nature of our research, ensuring that our findings reach a wide and relevant audience. We are confident that our work will contribute meaningfully to the scientific community and align with the journal's standards of publishing impactful and high-quality research.

41

Version 1:

Reviewer comments:

Reviewer #1

(Remarks to the Author)

I regarded this manuscript as a methodology paper aiming at accurate and high-coverage lipidomics annotations at sn-position level. After reading the revised manuscript and detailed workflow, I kept my concerns about this technology:

1. The multiple IM peaks generated by HRdm were the core of this workflow to resolved sn-position isomers of lipids. So, the accuracy was highly depended on the HRdm. However, HRdm tended to generate artificial peaks and the ratios of false-positive peaks remained unknown, particularly in complex biological samples.

2. In the 1st rule of the workflow, the DDA spectra were used to determine the potentially compositions of multiple peaks generated by HRdm. I worried whether the isomer numbers resolved by DDA spectra and HRdm always kept the same. What did it mean when the numbers exhibit inconsistency? Would HRdm tend to generate more peaks than those counted in DDA spectra?

3. I still did not see how much this workflow improved the lipid annotation compared with the combined strategy with 4D lipidomics matched with 4D library (m/z, RT, CCS and MS/MS) and rule-based refinement without HRdm. The 4D match with rule-based refinement in negative mode was enough to identify and resolve lipids at sn-position levels, particularly for identifying the major composition of the isomers. As authors only annotated 4 isomers for each lipid, I believe that the improvement was not significant.

Overall, I kept my concerns about the novelty and essentiality of the developed method. The ratios of artificial peaks generated by HRdm lacked comprehensive evaluations and validations.

Reviewer #2

(Remarks to the Author)

The authors present a significantly improved manuscript in which they address all my and, as far as I can tell, the concerns of all other reviewers. Therefore, great job! I am happy to support publication of the manuscript in its current form.

Reviewer #3

(Remarks to the Author)

This is a highly technical paper which reports findings that are highly significant. However, in the current form, it is difficult to determine the biological relevance and therefore the impact of the paper is considered to be low. The authors have responded to the previous reviewer critiques in the response document, but have not adequately translated the changes into the main text of the manuscript to address the previous reviewer concerns. The data is largely over-interpreted in the results section. The result section contains extensive discuss of the findings which should be removed to the discussion section. The limitations of the study and the interpretation of the data are not adequately discussed. Specific examples follow, but the over-interpretation of the data are not limited to the following examples:

There is no evidence for the statement on page 12, In 424 that "the significant reduction of DHA-containing GPs is thus associated with the dysregulated brain function for maintenance of membrane fluidity, neuronal survival, synaptic neurotransmission, and regulation of neuroinflammation

The Results section displays over interpretation of the limited results on enzymatic activity and in general there is too much discussion in the result section. For example the following statement was not supported by data presented in this paper. (Page 15 line 505) "The increase of cPLA2 was previously attributed to the ability of A β to stimulate astrocyte-mediated release of cytokines and nitric oxide, resulting in an activated cPLA2 cascade with enhanced COX-2 expression and upregulation of both oxidative and inflammatory responses in AD>"

On page 15 In 538, The increase in LPEAT activity is not supported and again this is over interpretation of the data. There is not sufficient data to support the increase in LPEAT activity differentially from the assessment of the levels of PE and LPE independently of the assessment of PLA2 activity.

The cPLA2 activity kit from Abcam (ab133090) cannot determine between types of PLA2 activities, but this limitation is not discussed and the results are over-interpreted in the results section.

These data are highly impactful to the field, but it is difficult to interpret the biological advances from these data as the paper is written. Many of the data are over-interpreted, the discussion is largely in the result section and the limitations of the methodologies are not adequately discussed. If these major concerns are addressed, this manuscript would have a strong impact in the field.

Author Rebuttal letter:

REVIEWER COMMENTS

Reviewer #1 (Remarks to the Author):

I regarded this manuscript as a methodology paper aiming at accurate and high-coverage lipidomics annotations at sn-position level. After reading the revised manuscript and detailed work-flow, I kept my concerns about this technology:

1. The multiple IM peaks generated by HRdm were the core of this work-flow to resolve sn-position isomers of lipids. So, the accuracy was highly depended on the HRdm. However, HRdm tended to generate artificial peaks and the ratios of false-positive peaks remained unknown, particularly in complex biological samples.

Response: We thank the reviewer for this valuable comment. The powerful ability of HRdm technique for structure characterization in complex samples has been demonstrated by recent studies (Nat. Commun. 2023, 14, 6795; Sci. Adv. 2023, 9, eadj7048; Anal. Chem. 2022, 94, 6191-6199). Artificial peaks are infrequently generated in our results, less than 5% lipid ions were observed coexisted with artificial peaks. Additionally, the artifacts did not interfere with the GP identification accuracy as they were typically far away from and were much less abundant (<10%) than the main peaks of real sn-resolved GPs in the drif spectra. They could be easily identified according to its inconsistent drift time alignment across the isotopic envelope. Furthermore, as from multiple replicated data acquisition, the artifact randomly appeared in certain spectra but not all spectra. Even in rare cases it might still be hard to identify whether a peak is genuine or an artifact in one drift spectrum, we were able to identify the real peaks which exhibited consistent CCS values across all technical and biological replicates while artifacts did not. And as the experimental database was validated through our rigorous scrutiny with our best knowledge, the future users would reference the robust and reproducible database to locate the genuine peaks rather than judging by themselves.

In response, we have included more data and conducted additional experiments to demonstrate the feasibility and reliability of HRdm techniques.

1) We compared the drift spectra of isotope encoded lipid standards in pure solvent and spiked in mouse brain samples to validate the feasibility and reliability of HRdm in complex biological

samples.

In the first round of revision, we clarified that HRdm not only simply deconvolutes the multiplexed spectra but also adopted several mathematical algorithms to greatly remove artifacts. Moreover, it's important to note that our study utilized the officially released stable

1

version 2.0 of the HRdm software, which includes several key improvements to largely reduce or eliminate artifacts compared with the initial report of HRdm in 2020. Additionally, the infrequent artifacts could be easily distinguished from genuine ions due to their considerable distance from the true ions, their low abundance, and non-aligned isotope peaks.

In order to further demonstrate the feasibility of HRdm technique in complex biological samples, we compared the drift spectra of isotope encoded lipid standards spiked in pure solvent (isopropanol) and complex biological samples (mouse brain tissue) acquired with HRdm. As shown in Supplementary Figure 6, all the drift spectra of deuterium-labeled lipids from the complex biological matrices could still align well with these in pure solvent, indicating that HRdm processing did not generate artificial peaks due to matrix interferences. Deuterium-labeled triacylglycerol (TG), diacylglycerol (DG), monoacylglycerol (MG), PC, PE, PS, PG, PI, PA, and sphingomyelin (SM) showed a single peak with high resolution IM technique (R_p up to 217) in both pure solvent and complex biological samples. Meanwhile, small peaks from isomers were observed in the ion mobility drift spectra of LPC 18:1 (d7)/0:0 and LPE 18:1 (d7)/0:0 (Supplementary Figure 6j-k). In order to validate these small peaks are results from true isomers rather than artificial signals, we performed the PLA2 digestion of isopure standards to obtain the pure lyso-GP standards. As shown in Supplementary Figure 7, all isopure lyso-GPs including LPC 18:1/0:0, LPE 18:1/0:0 appeared as a single peak and aligned well with their corresponding deuterium-labeled standards. As not all these deuterium-labeled lyso-GP standards are isopure standards, we deduced that these two lyso-GPs contain small amounts of corresponding sn-isomers with fatty acyl at sn-2. This also accords with the fact that lyso-GPs with fatty acyl at sn-2 have the larger CCS values (Anal. Chem. 2020, 92, 9482-9492; Analyst, 2016, 141, 1649-1659). Taken together, we are confident that our reported GP identification could truthfully reflect the GPs from complex biological samples rather than artificial peaks.

[Redacted]

2
Supplementary Figure 6. Comparison of the drift spectra from isotope encoded lipid standards spiked in pure solvent (isopropanol) and complex biological samples (mouse brain tissue) acquired with HRdm. Drift spectra of TG 15:0/18:1(d7)/15:0 (a), DG 15:0/18:1(d7) (b), MG 18:1(d7) (c), PC 15:0/18:1(d7) (d), PE 15:0/18:1(d7) (e), PS 15:0/18:1(d7) (f), PG 15:0/18:1(d7) (g), PI 15:0/18:1(d7) (h), PA 15:0/18:1(d7) (i), LPC 18:1(d7) (j), LPE 18:1 (d7) (k), SM d18:1/18:1 (d9) (l) in mouse brain sample (upper, red) and solvent (bottom, blue).

[Redacted]

3
Supplementary Figure 7. Drift spectra of sn-isomer pure lyso-GPs obtained by PLA2 digestion of isopure lipid standards. Drift spectra of LPC 18:1/0:0 digested from PC 18:1(9Z)/16:0 (a), LPE 18:1/0:0 digested from PE 18:1(9Z)/16:0 (b), LPC 16:0 digested from PC 16:0/18:1(9Z) (c), LPE 16:0 digested from PE 16:0/18:1(9Z) (d).

We added new data as Supplementary Figures 6-7, more detailed descriptions of HRdm technique, and new description of these new experimental results into the Results section, as below:

In order to further demonstrate the reproducibility of HRdm technique in complex biological samples, we compared the drift spectra of isotope encoded lipid standards spiked in pure solvent (isopropanol) and complex biological samples (mouse brain lipid extract) acquired with HRdm. As shown in Supplementary Figure 6, all the drift spectra of deuterium-labeled lipids from the complex biological matrices could still align well with these in pure solvent, indicating that HRdm processing did not generate artificial peaks due to matrix interferences. Deuterium-labeled triacylglycerol (TG), diacylglycerol (DG), monoacylglycerol (MG), PC, PE, PS, PG, PI, PA, and sphingomyelin (SM) showed a single peak with high resolution IM technique (R_p up to 217) in both pure solvent and complex biological samples. Meanwhile, small peaks from isomers were observed in the drift spectra of LPC 18:1 (d7)/0:0 and LPE 18:1 (d7)/0:0 (Supplementary Figure 6j-k). In order to validate that these small peaks are results from true isomers rather than artificial signals, we performed the PLA2 digestion of isopure standards to obtain the pure lyso-GP standards. As shown in Supplementary Figure 7, all isopure lyso-GPs including LPC 18:1/0:0, LPE 18:1/0:0 appeared as a single peak and aligned well with deuterium-labeled standards. As not all

of these deuterium-labeled lyso-GP standards are isopure standards, we deduced that these two lyso-GPs contain small amounts of corresponding sn-isomers with fatty acyl at sn-2. This also accord with the fact that lyso-GPs with fatty acyl at sn-2 has larger CCS values^{45,53}. The peaks of deuterium-labeled lyso-GP in both pure solvent and biological samples were also all aligned well.

4

Moreover, it is important to note that our study utilized the officially released stable version 2.0 of the HRdm software, which includes several key improvements to largely reduce or eliminate artifacts compared with the initial report of HRdm in 2020. Taken together, we are confident that lipid species identified by HRdm from complex biological samples could largely reflect true lipid signals rather than artificial peaks.⁵

2) We also comprehensively evaluated the possibility of artificial peak generation in biological samples and did multiple control experiments to make sure the confidence of GP identification.

It is worth noting that the infrequent artifacts could be filtered out and would not affect spectral interpretation by the following criteria:

Specifically, the artifacts are usually of low abundance (<10% of the primary signal within each m/z extraction window) and are sufficiently distant from the signal region of interest to not affect spectral interpretation. Moreover, IM drift times of genuine ion signal align across isotopic peaks due to the negligible contribution of mass to the ion mobility, whereas the isotopic peaks of artifacts are not aligned. For instance, as shown in Supplementary Figure 18 a1-a2, the abundance of artificial peak (at 41.69 ms) in IM-MS heatmap and extracted drift spectrum were much lower than true ions, and the isotopic peaks were not aligned. Thus, artificial peak (at 41.69 ms) co-existing with PE 18:0/20:4 (40.38 ms) and PE 20:4/18:0 (40.72 ms) in QC sample were easily distinguished from genuine ions due to their long distance from the true ions, their low abundance, and non-aligned isotope peaks. The existence of PE 18:0/20:4 (40.38 ms) and PE 20:4/18:0 (40.72 ms) in QC sample also could be validated with the negative MS2 spectrum (Supplementary Figure 18 a3). On the one hand, the low possibility of the artificial peaks could be easily discriminated from our genuine ions and would not affect the identification accuracy. On the other hand, we also validated the drift spectra of lipid extracts from QC samples with that from most of investigated biological samples, the artificial peak, which only randomly appeared in certain spectra but not all spectra, could be distinguished from consistently appeared genuine peaks. For instance, as shown in Supplementary Figure 18, one peak in drift spectrum of PE 18:0_20:4 happened to exist in QC sample at a very low abundance and with an isotope of inconsistent drift time (Supplementary Figure 18 a1&a2), but did not appear in other samples (Supplementary Figure 18 b1-j1). The major two drift peaks aligned well across all drift spectra and their isotopes (Supplementary Figure 18 a1-j1), indicating they are genuine peaks, PE 18:0/20:4 and PE 20:4/18:0. The very small peak (at 41.69 ms), with not aligned isotopic peaks, and far from the major peaks in the drift spectrum of QC sample, is the artificial peak (Supplementary Figure 18 a1&a2).

With that, we also added description in the Results section that checking drift time alignment of isotopic peaks and validation across replicates are recommended to determine the genuine

5

peaks when using HRdm to acquire high-resolution IM data for unknown biomolecules and new database.

[Redacted]

Supplementary Figure 18. IM-MS heatmap and extracted drift spectra of PE 18:0_20:4 from different samples. IM-MS heatmap and extracted drift spectra of PE 18:0_20:4 from QC sample (a1, a2), CB-WT 3-1 (b1, b2), CB-WT 8-1 (c1, c2), CB-WT 8-2 (d1, d2), CB-AD 3-1 (e1, e2), CB-AD 3-2 (f1, f2), CB-AD-8-2 (g1, g2), HP-AD 3-1 (h1, h2), CTX AD 3-1 (i1, i2), and CTX-WT 8-1 (j1, j2), respectively. a3 Negative MS2 Spectrum of PE 18:0_20:4 from QC sample.

6

From the 318 extracted drift spectra of 498 identified lipids in pooled mouse brain samples, artificial peaks were only observed in 15 extracted drift spectra, indicating that less than 5% of lipid ions co-existed with artifacts. Drift times of all artificial peaks from pooled mouse brain samples were labeled and supplemented in the notes of Supplementary Data 3. Herein, the low ratio of the artificial peaks could be easily discriminated from our genuine ions and would not affect the identification accuracy.

Additionally, most recently, HRdm has been used to elucidate exact O-acetylation patterns as well as glycosidic linkage types of sialosides isolated from complex biological samples such as equine tracheal, nasal tissues, and influenzaA viruses (Nat. Commun. 2023, 14, 6795). Moreover,

untargeted HRdm-IM-MS methods have been used to uncover per- and polyfluoroalkyl substances (PFAS) in complex samples (Sci. Adv. 2023, 9, ead7048). The powerful structural characterization capability of HRdm in complex samples has also been demonstrated by these excellent studies.

We added new description into the Introduction section, new data as Supplementary Figure 18, and new discussion in Results section as below:

High-resolution demultiplexing (HRdm) strategy was first reported in 2020, which is an extended Hadamard multiplexing and post-acquisition data processing strategy for improving the sensitivity and resolution of ion mobility measurements without the need for instrumental modifications⁴⁵. In the past four years, a series of isomeric mixtures in complex samples were investigated with HRdm, including glycans, peptides, metabolites, and oxidized lipids⁴⁶⁻⁴⁸. However, utilizing the strategy to systematically separate subtle yet crucial structural sn-position isomers of GPs in biological samples has not been explored in existing studies.

One of the concerns is that infrequent artifacts might affect the identification. Notably, the infrequent artifacts could be easily filtered out according to their features and would not affect spectral interpretation⁴⁵. Specifically, the artifacts are usually of low abundance (<10% of the primary signal within each m/z extraction window) and are sufficiently distant from the signal region of interest to not affect spectral interpretation. Moreover, IM drift times of genuine ion signal align across isotopic peaks due to the negligible contribution of mass to the ion mobility, whereas the isotopic peaks of artifacts are not aligned. For instance, as shown in Supplementary Figure 18 a1-a2, the abundance of artifactual peak (at 41.69 ms) in IM-MS heatmap and extracted drift spectrum were much lower than true ions, and the isotopic peaks were not aligned. Thus, artifactual peak (at 41.69 ms) co-existing with PE 18:0/20:4 (40.38 ms) and PE 20:4/18:0

7

(40.72 ms) in QC sample were easily distinguished from genuine ions due to their long distance from the true ions, their low abundance, and non-aligned isotope peaks. The existence of PE 18:0/20:4 (40.38 ms) and PE 20:4/18:0 (40.72 ms) in QC sample also could be validated with the negative MS2 spectrum (Supplementary Figure 18a3). On the one hand, the low possibility of the artifactual peaks could be easily discriminated from our genuine ions and would not affect the identification accuracy. On the other hand, we also validated the drift spectra of lipid extracts from QC samples with that from most of investigated biological samples, the artifactual peak, which only randomly appeared in certain spectra but not all spectra, could be distinguished from consistently appeared genuine peaks. For instance, as shown in Supplementary Figure 18, one peak in drift spectrum of PE 18:0_20:4 happened to exist in QC sample at a very low abundance and with an isotope of inconsistent drift time (Supplementary Figure 18 a1&a2), but did not appear in other samples. The major two drift peaks aligned well across all drift spectra and their isotopes, indicating they are genuine peaks, PE 18:0/20:4 and PE 20:4/18:0. The very small peak (at 41.69 ms), with not aligned isotopic peaks, and far from the major peaks in the drift spectrum of QC sample, is the artifactual peak. Therefore, checking drift time alignment of isotopic peaks and validation across replicates are recommended to determine the genuine peaks when using HRdm to acquire high-resolution IM data for unknown biomolecules and new database. From the 318 extracted drift spectra of 498 identified lipids in pooled mouse brain samples, artifactual peaks were only observed in 15 extracted drift spectra, indicating that less than 5% of lipid ions co-existed with artifacts. Drift times of all artifactual peaks from pooled mouse brain samples were labeled and supplemented in the notes of Supplementary Data 3. Herein, the low ratio of the artifactual peaks could be easily discriminated from our genuine ions and would not affect the identification accuracy.

To reduce the incidence of false positive identifications due to potential artifactual peaks, the annotated GPs were manually assessed to ensure that their CCS values remained consistent across all replicate measurements.

3) We also demonstrated the advantages of HRdm strategy as currently the most suitable paradigm for large scale GP profiling at sn-position level by comparing with different types of high-end IM-MS.

We hope to highlight our team's extensive experience in using and comparing various IM-MS platforms for molecule structural characterization (TrAC, Trends Anal. Chem. 2022, 157; Nat. Commun. 2023, 14, 5185, Anal. Chem. 2023, 95, 50, 18504-18513; Nat. Commun. 2019, 10, 5038; Anal. Chem. 2014, 86, 2972-2981). In order to make a reasonable comparison with HRdm strategy,

8

we also have operated other commonly used high-end IM paradigms such as TIMS and Traveling Wave IMS (TWIMS)-based cyclic IM (cIM) for sn-position isomer separation.

As shown in Supplementary Figure 8, in TIMS, when we used the default 4D lipidomics instrument parameters with 100 ms ramping time (Nat. Commun. 2023, 14, 937), the sn-isomer pairs could not be separated. It requires prolonged ramping time (up to 1000 ms) to achieve a slight separation. The sensitivity remarkably decreased as only ~10% ion intensities were preserved with the extremely long ramping time. In cyclic ion mobility-MS (Supplementary Figure 9), an increase in number of cyclic passes of up to 20 passes (around 400 ms arrival time) were needed to achieve a similar degree of separation as that shown in HRdm. It is important to note that this approach is more suitable for targeted analysis as it requires a carefully calculated narrow IM selection window to prevent wrap-around in the cIM where high-mobility ions catch up with low-mobility ions in multipass experiments. The ion intensity was also significantly decreased to ~10% when increasing the number of cyclic passes to 20. These compromises, including prolonged drift/ramping time and/or narrowed IM selection window, would come at cost of sacrificing either sensitivity, throughput, or IM coverage for large-scale untargeted lipidome profiling. In contrast, this DTIMS-based HRdm strategy demonstrates unique advantages for large-scale lipidome profiling in LC-IM-MS workflows as enhanced IM resolving power and sensitivity could still be achieved in a typical 60 ms IM scanning window for comprehensive ion collection. In addition, DTIMS provides reproducible first-principle CCS measurement, and DTCCSN2 measurement is considered the gold standard and is most widely reported in data repositories like PubChem, LIPIDMPAS, and MSDIAL. For these reasons, DTIMS-based HRdm IMS is currently the most suitable paradigm for sn-position-resolved lipidomic analysis and CCS database construction.

[Redacted]

9

Supplementary Figure 8. The extracted ion mobilograms of GP sn-isomers acquired by TIMS. The extracted ion mobilograms of deprotonated PE 16:0/18:1(9Z) (green), PE 18:1(9Z)/16:0 (blue) and mixture of the sn-isomer pairs (dark) (at m/z 718.5381) acquired using 100 ms ramp time (a) and 1000 ms ramp time (b).

[Redacted]

Supplementary Figure 9. Ion mobility spectra of GP sn-isomers acquired by cyclic ion mobility spectrometry. Ion mobility spectra of deprotonated PE 16:0/18:1(9Z) (green), PE 18:1(9Z)/16:0 (blue) and mixture of the sn-isomer pairs (dark) (at m/z 718.5381) after 1 pass (a), 5 passes (b), 10 passes (c), and 20 passes (d) in cIM device.

10

The new data has been added as Supplementary Figures 8-9, the IM-MS data acquisition parameters for TIMS and cIM-MS were described in the Supporting Information. A description of this new data has been added into the revised manuscript accordingly, as below:

To provide reasonable comparison and demonstrate the advantages of HRdm strategy, we also evaluated other types of commonly used high-end IM paradigms, including TIMS and Traveling Wave IMS (TWIMS)-based cyclic IM (cIM), for large scale GP profiling at fatty acyl sn-position level. The IM-MS data acquisition parameters for TIMS and cIM-MS were described in the Supporting Information. As shown in Supplementary Figure 8, In TIMS, sn-isomers could not be separated when we used the default 4D lipidomics instrument parameters with 100 ms ramping time³⁵. It required a prolonged ramping time (up to 1000 ms) to achieve a slight separation. The sensitivity remarkably decreased as only ~10% ion intensities were preserved with the extremely long ramping time. In cIM-MS (Supplementary Figure 9), an increase in number of cyclic passes of up to 20 passes (around 400 ms arrival time) were needed to achieve a similar degree of separation as that shown in HRdm. It is important to note that this approach is more suitable for targeted analysis as it requires a carefully calculated narrow IM selection window to prevent wrap-around in the cIM where high-mobility ions catch up with low-mobility ions in multipass experiments. The ion intensity was also significantly decreased to ~10% when increasing the number of cyclic passes to 20. These compromises, including prolonged drift/ramping time and/or narrowed IM selection window, would come at cost of sacrificing either sensitivity, throughput or IM coverage for large-scale untargeted lipidome profiling³². In contrast, this DTIMS-based HRdm strategy demonstrated unique advantages for large-scale lipidome profiling in LC-IM-MS workflows as enhanced IM

resolving power and sensitivity could still be achieved in a typical 60 ms IM scanning window for comprehensive ion collection. In addition, DTIMS provides reproducible first-principle CCS measurement, and DTCCSN2 measurement is considered the gold standard and is most widely reported in data repositories like PubChem, LIPIDMPAS, and MSDIAL36,40,54. For these reasons, DTIMS-based HRdm IMS is currently the most suitable paradigm for sn-position-resolved lipidomic analysis and CCS database construction.

Taken together, following the reviewer's comments, we have extensively revised the manuscript by performing additional experiments, incorporating new data, and adding more detailed discussions. The identification-confidence of HRdm was demonstrated by the consistent drift spectra of all the deuterium-labeled lipids in pure solvent and complex biological samples. The low possibility (less than 5%) of the artificial peaks could be easily discriminated from our genuine ions and would not affect the identification accuracy. Furthermore, the feasibility of HRdm IMS for large scale GP profiling at fatty acyl sn-position level has been demonstrated by comparing the separation efficiency with other high-end IM paradigms. The quality and novelty of the manuscript has been significantly enhanced, and we are grateful for the reviewer's generous help and insightful comments.

11

2. In the 1st rule of the workflow, the DDA spectra were used to determine the potential compositions of multiple peaks generated by HRdm. I worried whether the isomer numbers resolved by DDA spectra and HRdm always kept the same. What did it mean when the numbers exhibit inconsistency? Would HRdm tend to generate more peaks than those counted in DDA spectra?

Response: We appreciate the reviewer for this helpful comment! We provide our detailed responses below.

In untargeted lipidomics, from collision-based MS/MS strategies, annotation of GP could be achieved at fatty acyl level. Besides the major abundant GP species, multiple co-eluted fatty acyl compositional isomers of relatively low abundance could also be annotated (Nat. Methods 2013, 10, 755-758; Nat. Methods 2015, 12, 523-526). However, in order to enhance the identification accuracy, in the widely used untargeted LC-MS/MS and LC-IM-MS/MS lipidomics analysis, almost all GPs were annotated with up to 2 fatty acyl compositions in mouse brain or other biological samples (Nat. Biotechnol. 2020, 38, 1159-1163; Nat. Commun. 2023, 14, 937; Nat. Protoc. 2022, 17, 2415-2430). Moreover, in LC-IM-MS/MS based 4D lipidomics with rule-based refinement strategy, in order to reduce the over-report, Zhu and co-workers also annotated the major GPs in complex biological samples, and ~90% of GPs were annotated as only 1 composition at fatty acyl sum composition, fatty acyl level, or fatty acyl sn-position level in mouse brain (Anal. Chim. Acta 2020, 1136, 115-124). Aligned with current most referenced studies (Nat. Biotechnol. 2020, 38, 1159-1163; Nat. Commun. 2021, 12, 6021; Anal. Chim. Acta 2020, 1136, 115-124), in this study, fatty acyl composition of most GPs in mouse brain is not too complicated and compositional isomers are no more than 2 from negative MS/MS spectra. In this way, there could be no more than 4 isomers if we take sn-positional isomers into account. That is the main reason that we annotated no more than 4 isomers from HRdm spectra.

We have supplemented a decision tree (Supplementary Figure 13) to clearly illustrate the annotation workflow and included negative MS/MS spectra and drift spectra of a series of GPs from mouse brain extracts (Supplementary Figure 14) to evaluate their consistency in complex biological matrix.

We understand your concern about the consistency of drift spectra and negative MS/MS spectra. Specifically, to enhance the identification accuracy and confidence, we set the minimum peak height at 3000 counts in MS1 as a criterion for identification. Together with fatty acyl composition information of lipid isomers from DDA spectra, we follow the decision tree to conduct the annotation:

12

1) If HRdm spectra show the same number of peaks as the number of fatty acyl compositional isomers, when GP contains 1 fatty acyl composition, it indicates sn-position isomers do not exist. We can assign the sn-connectivity of the fatty acyl chains by rule 1 (Supplementary Figure 14a-c). For instance, as shown in Supplementary Figure 14a, there was only single peak in the HRdm drift spectrum of corresponding to the m/z of PC 44:10. The only two fatty acyl fragment ions, corresponding to C22:6 and C22:4 at a ratio of 22:4/22:6=3, indicated single fatty acyl composition (PC 22:6_22:4), and the connectivity could be further assigned as PC 22:6/22:4. When GP contains two fatty acyl compositions, it would be annotated as 2 sn-resolved fatty acyl compositions or 2 sn-isomers of major composition according to rule 1 and rule 2. For instance,

as shown in Supplementary Figure 14h, from the MS/MS spectra, PE 16:0_18:1 was the predominant fatty acyl chain composition, and the extremely low abundance of PE 16:1_18:0 indicated which would not be the second relative high peak in drift spectrum. The intensity ratio of fragment ions from PE 16:0_18:1 accords with their abundance in drift spectrum, the connectivity could be further assigned as PE 16:0/18:1 and PE 18:1/16:0.

2) It was rarely observed that the number of IM peaks less than fatty acyl compositions (means 1 IM peak, 2 fatty acyl compositions), if existed, it would be annotated as the major composition with sn-position.

3) Another case is that the peak number is higher than the number of lipid isomer at fatty acyl composition level. If the maximum possible number of sn-isomers matches the number of peaks from HRdm spectra (Supplementary Figure 14 d-g, i, l), we used rule 1 (and rule 2) to assign the sn-resolved GPs. There are also cases where the maximum possible number of sn-isomers is higher than the number of peaks from HRdm spectra. For example, 2 compositional isomers were identified in DDA spectra, which could lead to 4 possible sn-position resolved isomers, but only 3 peaks were observed in HRdm spectra (Supplementary Figure 14j-k). If this happens, we will use rule 1 to pick the top 3 abundant sn-position resolved GP isomers. The assignment of the 3 peaks in HRdm spectra is based on peak intensities and rule 2. In our data, we did not observe the case that the number of peaks in HRdm spectra is higher than the maximum possible number of sn-isomers from DDA after we carefully eliminate artifacts if any exists.

In the rare cases when more than 2 compositional isomers were identified in DDA spectra, we pick the top 4 abundant sn-position resolved GP isomers and the top 4 abundant peaks in HRdm spectra, and then assign them based on rule 1 and rule 2. For the very low abundance isomers, although they might be identified in our workflow from MS/MS spectra in negative mode, they were not annotated from the HRdm drift spectra to enhance the identification accuracy and confidence. It is worth noting that this potential limitation also exists in current mainstream label free lipidomics methods when quantitative analysis is the major focus (Nat. Biotechnol. 2020, 38, 1159-1163; Nat. Commun. 2021, 12, 6021; Nat Protoc. 2022, 17, 2415-2430; Anal. Chim. Acta 2020,

13

1136, 115-124; Nat. Commun. 2023, 14, 937). We have provided our raw data files at MassIVE (MSV000092785) and all annotation results in Supplementary Data 3 as resources.

[Redacted]

Supplementary Figure 13. The annotation workflow concluded from negative MS/MS spectra and drift spectra. IM peaks indicate genuine peaks after filtering out artificial peaks.

[Redacted]

14

Supplementary Figure 14. Representative MS2 and drift spectra of GPs from mouse brain lipid extracts. MS2 and drift spectra of PC 22:6/22:4 (a), PS 22:6/20:4 (b), PS 22:6/22:6 (c), PC 14:0_16:0 (d), PC 18:0_20:4 (e), PC 18:1_24:0 (f), PC 18:1_20:4 (g), PE 16:0_18:1 (h), PE 16:0_20:4 (i), PE 38:2 (j), PC 38:2 (k), and PE 34:2 (l).

15

Following these helpful comments, we have added new data and discussion to the revised manuscript to demonstrate and highlight the high identification accuracy of the combined 4D lipidomic strategy for GPs at sn-position resolved level.

A description of this new data has been added into the revised manuscript accordingly, as below:

In negative MS/MS spectra of this study, we found that fatty acyl composition of most GPs in mouse brain was not too complicated and almost all isomers at fatty acyl composition level were no more than 2. This is also reported by other literature^{43,53,54}. In this way, there could be no more than 4 isomers if we take sn-positional isomers into account. That is the main reason that we annotated no more than 4 isomers from HRdm spectra. According to these negative MS/MS spectra and drift spectra, the annotation workflow was illustrated in Supplementary Figure 13. Additionally, a series of GPs from mouse brain extracts were also used to comprehensively evaluate the consistency of drift spectra and negative MS/MS spectra from complex biological matrix (Supplementary Figure 14). Specifically, to enhance the identification accuracy and confidence, we set the minimum peak height at 3000 counts in MS1 as a criterion for identification.

Together with fatty acyl composition information of lipid isomers from DDA spectra, we follow the decision tree to conduct the annotation: If HRdm spectra show the same number of peaks as the number of compositional isomers, when GP contains 1 fatty acyl composition, it indicates sn-position isomers does not exist. We can assign the sn-connectivity of the fatty acyl chains by rule 1 (Supplementary Figure 14a-c). For instance, as shown in Supplementary Figure 14a, there was only single peak in the HRdm drift spectrum of corresponding to the m/z of PC 44:10. The only two fatty acyl fragment ions, corresponding to C22:6 and C22:4 at a ratio of 22:4/22:6â³, indicated single fatty acyl composition (PC 22:6_22:4), and the connectivity could be further assigned as PC 22:6/22:4. When GP contains two fatty acyl compositions, it would be annotated as two sn-resolved fatty acyl compositions or two sn-isomer of major composition according to rule1 and rule 2. For instance, as shown in Supplementary Figure 14h, from the MS/MS spectra, PE 16:0_18:1 was the predominant fatty acyl chain composition, and the extremely low abundance of PE 16:1_18:0 indicated which would not be the second relative high peak in drift spectrum. The intensity ratio of fragment ions from PE 16:0_18:1 accords with their abundance in drift spectrum, the connectivity could be further assigned as PE 16:0/18:1 and PE 18:1/16:0. It was rarely observed that the number of IM peaks less than fatty acyl compositions (means 1 IM peak, 2 fatty acyl composition), if existed, it would be annotated as the major composition with sn-position. Another case is that the peak number is higher than the number of lipid isomers at fatty acyl composition level. If the maximum possible number of sn-isomers matches the number of peaks from HRdm spectra (Supplementary Figure 14 d-g, i, l), we used rule 1 (and rule 2) to assign the sn-resolved GPs. There are also cases where the maximum possible number of sn-

16

isomers is higher than the number of peaks from HRdm spectra. For example, two compositional isomers were identified in negative MS/MS spectra, which could lead to 4 possible sn-position resolved isomers, but only three peaks were observed in HRdm spectra (Supplementary Figure 14j-k). If this happens, we will use rule 1 to pick the top 3 abundant sn-position resolved GP isomers. The assignment of the three peaks in HRdm spectra is based on peak intensities and rule 2. In our data, we did not observe the case that the number of peaks in HRdm spectra is higher than the maximum possible number of sn-isomers from negative MS/MS spectra after we carefully eliminated artifacts if any exists. In the very rare cases when more than 2 compositional isomers were identified in DDA spectra, we pick the top 4 abundant sn-position resolved GP isomers and the top 4 abundant peaks in HRdm spectra, and then assign them based on rule 1 and rule 2. For the very low abundance isomers, although they might be identified in our workflow from MS/MS spectra in negative mode, they were not annotated from the HRdm drift spectra to enhance the identification accuracy and confidence. It is worth noting that this potential limitation also exists in current mainstream label free lipidomics methods when quantification is the major focus^{35,36,43,53,54}.

3. I still did not see how much this workflow improved the lipid annotation compared with the combined strategy with 4D lipidomics matched with 4D library (m/z, RT, CCS and MS/MS) and rule-based refinement without HRdm. The 4D match with rule-based refinement in negative mode was enough to identify and resolve lipids at sn-position levels, particularly for identifying the major composition of the isomers. As authors only annotated 4 isomers for each lipid, I believe that the improvement was not significant.

Response: We appreciate the reviewer for the critical thinking. We now have compared the identification results between using 4D library matching and rule-based refinement without HRdm and our 4D lipidomics with HRdm.

1) Current 4D lipidomics studies reported by Zhu and co-workers greatly improved the identification accuracy in lipid structure characterization. However, in the 4D library created by Zhu and co-workers, sn-position isomers are not differentiated in low resolution ion mobility dimension (Anal. Chim. Acta 2020, 1136, 115-124; Anal. Chem. 2017, 89, 17, 9559-9566). Moreover, in the combined strategy reported by Zhu and co-workers that integration of 4D (m/z, RT, CCS and MS/MS) library-based match and rule-based refinement without HRdm, only a part of GP species was roughly identified at fatty acyl sn-position level and the existence of sn-position isomers was ignored. Specifically, in the rule set for lipid identification, GP species from each extracted peak was assigned to only one of the sn-isomer pairs if the intensity ratio of sn-1/sn-2 was < 0.9 in negative ion mode, while the existence of corresponding sn-isomers was ignored (Fig.

17

1E in Anal. Chim. Acta 2020, 1136, 115-124). Additionally, the identification of GP was still at fatty acyl level when the intensity ratio of sn1/sn2 was \geq 0.9 (Anal. Chim. Acta 2020, 1136, 115-124; Anal. Chim. Acta 2022, 1210, 339886). Since most GPs in complex biological samples exist as a mixture form of sn-positional isomers, the clear limitation of this annotation is that all co-eluting

sn-position isomers from complex biological samples were identified as one GP species, ignoring the existence of sn-position isomers. In contrast, with high-resolution ion mobility, we were able to identify both sn-position isomer pairs when they existed.

On the other hand, the most important aim of the combined strategy reported by Zhu and co-workers is to reduce the over-reports since the 4D library-based match approach often gives multiple lipid candidates. Although with ion mobility, they could achieve the annotation of a proportion of GPs towards higher level of identification at eventually fatty acyl sn-position level, also only the major isomer was annotated for one lipid. From the annotation results from mouse brain of this literature, all of GPs were annotated with only 1-2 major isomers in negative mode, and ~90% of GPs were annotated as 1 composition at fatty acyl sum composition, fatty acyl level, or fatty acyl sn-position level. As we demonstrated in response to your Comment 2, annotation of top 4 isomers has covered most cases of co-eluting GP isomers at sn-position level. Hence, the improvement is significant to the current method using 4D library match with rule-based refinement in negative mode.

[Redacted]

Figure 1E. The example rule set for lipid species [PC + COOH]- (from *Anal. Chim. Acta* 2020, 1136, 115-124).

18

Following the reviewer's comment, we have added the description of advantages and limitations of the strategies that 4D library-based match and rule-based refinement in Introduction section as follows:

Recently, Zhu and co-workers reported a strategy that integrated 4D (m/z, RT, CCS and MS/MS) library-based match and rule-based refinement to reduce over-report and improve accuracy of lipid identification^{37,43}. These current 4D lipidomics studies greatly improved the depth in lipid structure characterization. However, in these studies, only a part of GP species was roughly identified at fatty acyl sn-position level and the existence of sn-position isomers was ignored. Specifically, in the rule set for lipid identification, GP species from each extracted peak was assigned to only one of the sn-isomer pairs if the intensity ratio of sn-1/sn-2 was < 0.9 in negative ion mode, while the existence of corresponding sn-isomers was ignored. Additionally, the identification of GP was still at fatty acyl level when the intensity ratio of sn1/sn2 was \geq 0.9. Moreover, the quantification could only be achieved for the most abundant isomer or fatty acyl sum composition. Nevertheless, these studies pushed the lipid structural annotation and quantification to the next level for providing unequivocal structural assignment of lipid isomers.

2) To clearly highlight the obvious improvement of our strategy, a comparison of identification numbers from different 4D lipidomics strategies has been conducted. By using 4D match with rule-based refinement in negative mode, in mouse brain, they finally identified a total of 126 GPs including 84 GPs at sn-position level (Supplementary Data, Multimedia component 7 of the reference *Anal. Chim. Acta* 2020, 1136, 115-124). Considering the different sample sources, extraction methods, instrument sensitivities, and LC-IM-MS acquisition methods, to make a reasonable comparison, we performed the analysis of the lipid extract from mouse brain using 4D library-based match and rule-based refinement without additional HRdm strategy as Zhu and co-workers reported. GPs were determined as one major component of sn-position isomers if the intensity ratio of sn-1/sn-2 < 0.9, and the existence of their corresponding sn-isomers were ignored. As shown in Supplementary Figure 19, a total of 326 GPs including 241 GPs at sn-resolved level and 85 GPs at fatty acyl level in mouse brain were identified. The identification result is comparable with studies that annotation of GPs at fatty acyl level using LC-MS/MS (*Nat. Biotechnol.* 2020, 38, 1159-1163). In our study, using the 4D lipidomics with HRdm, a total of 498 GPs has been identified with additional sn-position isomers from sn-position isomer pairs. Our result represents more than 50% increase in the number of GP identification compared to that obtained through 4D library-based match and rule-based refinement without HRdm strategy. We believe that the increase in identification number obtained by our developed 4D lipidomics with HRdm is significant compared to 4D library-based match and rule-based refinement without HRdm.

[Redacted]

19

Supplementary Figure 19. Comparison of GPs identification number between reported strategy using 4D library-based match and rule-based refinement without HRdm (4D w/o HRdm) and our 4D lipidomics strategy (4D w/

HRdm). a Comparison of total identified number of GPs between 4D w/o HRdm and 4D w/ HRdm strategies. b Comparison of identified number of each GP class between 4D w/o HRdm and 4D w/ HRdm strategies.

We have also included the new data into the revised manuscript to highlight the improvement of our methods, as below:

To clearly highlight the obvious improvement in identification number benefits from high-resolution IM, a comparison of identification numbers from different 4D lipidomics strategies has been conducted. We performed the analysis of the lipid extract from mouse brain using 4D library-based match and rule-based refinement without additional HRdm strategy as Zhu and co-workers reported⁴³. GPs were determined as one major component of sn-position isomers if the intensity ratio of sn-1/sn-2 was < 0.9, and the existence of their corresponding sn-isomers were ignored. As shown in Supplementary Figure 19, a total of 326 GPs including 241 GPs at sn-resolved level and 85 GPs at fatty acyl level in mouse brain were identified. The identification result is comparable with studies that annotation of GPs at fatty acyl level using LC-MS/MS⁵⁴. In our study, using the 4D lipidomics with HRdm, a total of 498 lipid species have been identified with additional sn-position isomers from sn-position isomer pairs.

The 4D database contains a total of 498 GPs with acyl chain sn-position information which were identified from the LC-HRdm IM-QTOF platform. These results represent more than 50% increase in the number of GP identification, which is 318 without sn-position assignment obtained through LC-QTOF (IM only), and 326 using 4D library-based match and rule-based refinement without additional HRdm strategy.

20

3) We acknowledge that collision-based MS/MS strategies enabled annotation of GP isomers at fatty acyl chain level, even including multiple co-eluted fatty acyl compositional isomers of relatively low abundance in untargeted lipidomics (Nat. Methods 2013, 10, 755-758; Nat. Methods 2015, 12, 523-526). We also hope to highlight our team's extensive experience in lipid identification using MS/MS strategies (Nat. Chem. 2024, 16, 762-770; Nat Commun. 2023, 14, 5185; Anal. Chem. 2022, 94, 38, 13036-13042; Chem. Sci., 2021, 12, 8115-8122; Anal. Chem. 2019, 91, 3, 1791-1795). To enhance the identification accuracy, in the widely used untargeted LC-MS/MS lipidomics analysis, GPs were annotated with the most abundant fatty acyl compositions in mouse brain samples (Nat. Biotechnol. 2020, 38, 1159-1163). In typical LC-IM-MS/MS lipidomics analysis, most GPs were also annotated with the up to 2 fatty acyl compositions in biological samples (Nat Protoc. 2022, 17, 2415-2430). Moreover, in order to reduce the over-report, Zhu and co-workers also annotated the major GPs in complex biological samples (Anal. Chim. Acta 2020, 1136, 115-124).

We fully understand your concern on the annotation of GPs with multiple (>3) major fatty acyl compositional isomers. In our study, annotating top four abundant GPs with sn-position enables the workflow to cover most of GPs in our investigated samples and enhances the identification accuracy. GPs with multiple (>3) major fatty acyl compositional isomers were very rarely observed in our samples, but it might exist in certain complex biological samples. If co-eluting GP isomers are composed of too complex fatty acyl composition, it will largely make it difficult to accurately annotate them from ion mobility dimension even with greatly enhanced ion mobility resolving power. This is also a difficult challenge for all current mainstream 4D lipidomics strategies. Regarding this challenge, we anticipate that reducing the co-elution of fatty acyl compositional isomers could largely alleviate the issue. This could potentially be achieved by combining high-resolution IM-MS with advanced chromatographic separation, including extended chromatographic gradient and multidimensional chromatographic separation.

In terms of these very low abundance isomers, they could be identified as well in our workflow from MS/MS spectra in negative mode in the same capability. For the purpose of identification accuracy and confidence, they were not annotated from the HRdm driven spectra. We acknowledged this limitation and mentioned this point in the Discussion section. Therefore, in short, we increased the annotation of high abundance lipid species with additional sn-position information while having limitation in reporting low abundance co-eluting isomers in our annotation results which is also present in other current 4D lipidomics methods.

4) The challenges in distinguishing lipid structural isomers also stunt the molecular depth of quantitative strategies. Numerous recent studies have demonstrated that unveiling the

21

mechanisms and networks behind lipid homeostasis calls for quantitative, and molecularly specific lipid analysis (Nat. Commun. 2023, 14, 1535; J. Am. Chem. Soc. 2021, 143, 14622-14634; Acc. Chem. Res. 2021, 54, 3873-3882; Angew. Chem. Int. Ed. 2022, 61, e2022070). Quantifying lipids beyond the fatty acyl level has also become a major endeavor in lipidomics. It is important to note that our methodology enables deeper quantification of GPs at fatty acyl sn-position level in a large-scale manner. In the routine untargeted LC-MS/MS based lipidomics and LC-IM-MS/MS based 4D lipidomics, although low abundance lipid fatty acyl composition could be identified through negative MS/MS, when doing quantitative analysis, the whole peak area of extracted m/z of a lipid in the chromatogram or mobiligram is usually only assigned at the sum composition level (e.g., PC 34:1), fatty acyl level (e.g., PC 16:0_18:1) or to one of the most abundant isomer among all co-eluting isomers (Nat. Biotechnol. 2020, 38, 1159-1163; Nat. Protoc. 2022, 17, 2415-2430; Nat. Commun. 2022, 13, 124; Anal. Chim. Acta 2020, 1136, 115-124). This means, despite potential identification of multiple isomers, the quantitative result would have been assigned for the most abundant isomer while the others are still not quantifiable. However, knowing concentrations of lipid structural isomers is essential for understanding their physiological functions and discovering new disease biomarkers. In our strategy, all the annotated co-eluting lipid isomers (up to 4) could be quantified, which means that we achieved quantification of all identified 498 GPs including sn-position isomer pairs compared to 326 quantifiable GP species without sn-position isomer pairs in routine method from pooled mouse brain lipid extract. As we validated the quantitative accuracy of HRdm using mixture of sn-position isomers at wide range of concentrations and using lipid standards across GP classes against PLA2 enzymatic digestion, the quantitative capability towards largely expanded lipid species would be one of the greatest advantages of our methodology.

Following the reviewer's point, we added discussion into the revised manuscript as below:

Compared to GP identification by only MS/MS in negative mode, although HRdm method has potential limitations, in annotating very low abundance isomers when, in rare cases, multiple fatty acyl compositional isomers co-eluting, it increased annotation of high abundance fatty acyl composition isomers with additional sn-position information, expanding the quantifiable GP isomers. To note, current routine untargeted LC-MS/MS or LC-IM-MS/MS lipidomics methods only enable quantification at the level of GP sum composition or for the predominant composition among co-eluting GPs^{40,43,50}. In contrast, benefiting from high resolution IM, separating GP sn-position isomers in the drift spectra enables the unequivocal quantification of both sn-position isomers.

Several limitations exist in our study. Firstly, for the very low abundance isomers of GPs with complicated fatty acyl compositions, although they might be identified in our workflow from MS/MS spectra in negative mode, they were not annotated from the HRdm drift spectra to

22

enhance the identification accuracy and confidence. This limitation would be addressed by employing high-sensitivity nanoESI-MS and next generation IMS with higher resolution. The challenges of annotating GPs with multiple major fatty acyl compositional isomers could potentially be overcome by combining high-resolution IM-MS with advanced chromatographic separation, including extended chromatographic gradient and multidimensional chromatographic separation.

Overall, I kept my concerns about the novelty and essentiality of the developed method. The ratios of artificial peaks generated by HRdm lacked comprehensive evaluations and validations.

Response: We thank the reviewer for bringing up the concerns. To address the reviewer's concerns, we have performed necessary experiments, incorporated additional data and discussion into the revised manuscript to strengthen the novelty and essentiality of this work. These efforts are summarized as follows:

(1) We highlight the major novelty and essentiality of our work compared to previously published work as follows:

a) We developed an LC-high-resolution IM-MS based 4D lipidomics strategy that enables large-scale, label-free, robust, and rapid GP identification and quantitative analysis at sn-position resolved level without additional derivatization and instrument modification. Lipids play important roles in cell signaling, cell structure, and energy storage, however detailed lipid structures remain largely underexplored in routine lipidomics owing to their complexity. Over the past decade, significant research efforts have been devoted to developing analytical methods for the detailed structural elucidation of lipids sn-position, C&C location and geometry isomers (Nat. Commun. 2023, 14, 4263; Nat. Commun. 2022, 13, 2652; Nat. Commun. 2020, 11, 375; Nat. Commun. 2019,

10, 79; Angew. Chem. Int. Ed. 2018, 57, 10530-10534; Angew. Chem. Int. Ed. 2022, 61, e2022070; Angew. Chem. Int. Ed. 2022, e202215556; J. Am. Chem. Soc. 2021, 143, 14622-14634). Recently, IM-MS has emerged as a promising technology for lipidomics to facilitate the separation and the identification of complex mixtures of analytes. Studies have demonstrated the power of ion mobility for enhancing identification accuracy of lipidome in biological samples (Nat Protoc. 2022, 17, 2415-2430; Nat Commun. 2022, 13, 124; Anal. Chem. 2017, 89, 17, 9559-9566). Moreover, studies have coupled LC-IM-MS/MS analysis and rule-based refinement for improving accuracy of lipid identification and achieve the annotation of a pair of GPs with the main isoform of sn-position isomers (Anal. Chim. Acta 2020, 1136, 115-124; Anal. Chim. Acta 2022, 1210, 339886). Furthermore, recent research has highlighted the enormous potential of high-resolution ion mobility for in-depth lipid characterization (Anal. Chem. 2019, 91, 5021-5027; Anal. Chem. 2020,

23

92, 9482-9492). Nevertheless, to the best of our knowledge, the use of label free LC-IM-MS/MS for large-scale GP sn-isomers identification and quantitative analysis in biological samples has not been demonstrated. The study presented here filled this critical gap and is the first report of an LC-IM-MS/MS strategy that enables large-scale GP sn-isomers profiling in biological samples.

b) We have constructed a highly confident extended 4D GP library in addition to an in-house developed experimental 4D database by machine learning prediction using an optimized selection of molecular descriptors (MDs), presenting the first CCS compendium of ~3000 sn-position resolved GPs for automated GP annotation. This constructed database is more in-depth than other currently available CCS database or libraries which do not differentiate GP sn-position isomers (Nat. Methods 2023, 20, 1836-1837; Nat. Protoc. 2022, 17, 2415-2430; Chem. Sci. 2019, 10, 983-993; Anal. Chem. 2017, 89, 9559-9566). Therefore, our database and library would be groundbreaking contributions to the field and are instrumental for researchers in biology and analytical sciences, offering valuable resources for detailed lipid structural identification.

c) We applied this developed strategy to spatially and temporally probe GP alterations in AD mouse brains, revealing unexplored GP dysregulation at sn-resolved level in aging and AD progression in a spatial and temporal specific manner. Studies have demonstrated significant changes of GP at fatty acyl sum composition or fatty acyl level in AD and aging mouse brain (Nat. Commun. 2021, 12, 6021; Commun. Biol. 2022, 5, 245; Acta. Neuropathol. Commun. 2021, 9, 116). However, the alteration of detailed GPs at sn-position resolved level across different brain regions with aging remains unexplored. In this study, with the capability to discriminate sn-position, we offer new insights into disease pathology associated with dysregulated lipid metabolism and could aid in uncovering potential lipid biomarkers for diseases.

We have revised the Discussion section to strengthen the novelty and essentiality of this work.

(2) Regarding the concerns on potential artificial peaks in HRdm, we have made thorough evaluation and validation using both lipid standards and complex biological matrices. From all of the measurement of lipid standards, we confirmed that we did not observe artificial peaks. Identification and quantitative accuracy of HRdm have been evaluated using PLA2 enzymatic digestion across multiple GP species and using isomer mixture of different ratios. To address the concern about the performance of HRdm in complex biological matrices, we also performed additional experiments. We compared the HRdm results of isotope-labeled lipid standards in pure solvent solution and spiked in mouse brain. The identification confidence was demonstrated by the consistent drift spectra of all the deuterium-labeled lipids in pure solvent and complex biological samples. We also comprehensively evaluated the rate of artificial peak generation in

24

biological samples. Less than 5% of lipid ions co-existed with artifacts. The low rate of the artificial peaks could be easily discriminated from our genuine ions and would not affect the identification accuracy. We also demonstrated the advantages of HRdm strategy as currently the most suitable paradigm for large scale GP profiling at fatty acyl sn-position level by comparing different types of high-end IM-MS.

Taken together, following the reviewer's constructive comments, we have extensively revised the manuscript by performing additional experiments, incorporating new data, and more detailed discussions. The quality and novelty of the manuscript has been significantly enhanced. Once again, we would like to express our sincere appreciation for your insightful comments and suggestions.

Reviewer #2 (Remarks to the Author):

The authors present a significantly improved manuscript in which they address all my and, as far as I can tell, the concerns of all other reviewers. Therefore, great job! I am happy to support publication of the manuscript in its current form.

Response: We are delighted that the reviewer acknowledges our efforts in addressing all the comments thoroughly and considers our study acceptable for publication. We appreciate the reviewer's critical and thoughtful comments to help improve our manuscript.

Reviewer #3 (Remarks to the Author):

This is a highly technical paper which reports findings that are highly significant. However, in the current form, it is difficult to determine the biological relevance and therefore the impact of the paper is considered to be low. The authors have responded to the previous reviewer's queries in the response document, but have not adequately translated the changes into the main text of the manuscript to address the previous reviewer concerns. The data is largely over-interpreted in the results section. The result section contains extensive discussion of the findings which should be

25

removed to the discussion section. The limitations of the study and the interpretation of the data are not adequately discussed. Specific examples follow, but the over-interpretation of the data are not limited to the following examples:

Response: We sincerely appreciate the reviewer's positive evaluation and recommendation of our work. We are also truly grateful for the reviewer's constructive feedback and suggestions, which have undoubtedly helped us to strengthen our manuscript.

We would like to reiterate that the greatest impact of this work is the development of the enabling technology for comprehensive sn-position resolved GP profiling, and the discovery of spatial and temporal GP alteration at sn-position resolved level in AD and aging mouse brain. Advancing bioanalytical technologies and uncovering their potential to address previously unresolved biological questions due to technical limitations is of significant importance and merits recognition. Through this technology, we established the workflow and the database that could be utilized through commercially available instrumentation and open-source software by researchers across multiple disciplines to increase the depth and coverage of lipid identification and eventually towards more precise or new discoveries in their research.

Through this strategy, we discovered the alteration of GPs at sn-position resolved level that was associated to aging and AD progression in the mouse brain. This has been largely underexplored in the past by the routine LC-MS based lipidomic investigation. As the GPs are remodeled through very complex pathways, the alteration of GPs could be regulated by multiple enzymes, including phospholipases and acyl transferases, among which cPLA2 was verified to have different activities in our study. We acknowledge that the GP alteration is related to cPLA2 activity but also involves multiple enzymes as a whole, which is hard to be systematically investigated by one manuscript. We thus provide a resource of spatial and temporal sn-position resolved GP data to motivate future follow-up studies with more biological validation. We hope our enabling technology and significant discovery would not only pave the way for the whole community but also call for more players into this field towards improved understanding of the GP alteration at a more in-depth molecular basis, thus leading to more discoveries with biological insights into the complex pathways of GP alteration.

In response, we have revised the manuscript following the reviewer's comments. Specifically:

1) We thoroughly and adequately translated all the changes discussed in the response and supplementary information into the main text of the manuscript to address the previous reviewer's concerns.

26

2) We substantially revised some parts of the Results and Discussion sections to interpret the data properly and adequately.

3) We moved the extensive discussion of findings from the Result to the Discussion section.

We believe that the quality of the manuscript has now been substantially improved with the reviewer's generous help.

There is no evidence for the statement on page 12, In 424 that the significant reduction of DHA-containing GPs is thus associated with the dysregulated brain function for maintenance of membrane fluidity, neuronal survival, synaptic neurotransmission, and regulation of neuroinflammation

Response: We thank the reviewer for the comments. In our study, we found downregulation of GP with PUFAs at the sn-2 position in WT at 8 months and AD at 3 months compared to WT at 3 months in all brain regions. Several previous studies reported that the reduction of DHA-containing GPs was linked to dysregulated brain function.

We have revised the statement as presented below and moved the statement from Results section to Discussions section:

Moreover, notable reduction of GP with PUFAs (e.g., DHA), at the sn-2 position was observed in AD and aged mice across all 3 brain regions. Therefore, there seems to be a temporal and genotype-dependent shift in the progression of changes. Herein, our results were consistent with previous studies and suggested that a drastic decrease in DHA-containing GPs might be an indicator of AD and aging^{63,65}. Obviously, it is always difficult to claim biological implications of MS-based findings and further studies are required to clarify the currently unclear molecular mechanisms of GP metabolism with aging and AD. However, it is tempting to connect our present data to independent studies that link depletion of DHA-containing GP to changes in membrane fluidity and excitability as well as regulation of neuroinflammation^{9,63-65,69}.

We also cited the following references:

Frisardi, V., et al. Glycerophospholipids and glycerophospholipid-derived lipid mediators: a complex meshwork in Alzheimer's disease pathology. *Prog. Lipid Res.* 50, 313-330 (2011).

27

Hachem, M. & Nacir, H. Emerging Role of Phospholipids and Lysophospholipids for Improving Brain Docosahexaenoic Acid as Potential Preventive and Therapeutic Strategies for Neurological Diseases. *Int. J. Mol. Sci.* 23, 3969 (2022).

Grimm, M. O., et al. Docosahexaenoic acid reduces amyloid beta production via multiple pleiotropic mechanisms. *J. Biol. Chem.* 286, 14028-14039 (2011).

Yassine, H. N., et al. Association of Docosahexaenoic Acid Supplementation With Alzheimer Disease Stage in Apolipoprotein E epsilon4 Carriers: A Review. *JAMA Neurol.* 74, 339-347 (2017).

Bennet, S. A., et al. Using neurolipidomics to identify phospholipid mediators of synaptic (dys)function in Alzheimer's Disease. *Front. Physiol.* 4, 168 (2013).

The Results section displays over interpretation of the limited results on enzymatic activity and in general there is too much discussion in the result section. For example the following statement was not supported by data presented in this paper. (Page 15 line 505) The increase of cPLA2 was previously attributed to the ability of A β to stimulate astrocyte-mediated release of cytokines and nitric oxide, resulting in an activated cPLA2 cascade with enhanced COX-2 expression and upregulation of both oxidative and inflammatory responses in AD.

Response: We deeply appreciate the reviewer's comments. We have revised the Results and Discussion sections to interpret the data properly. The above statement, reported by the literature but not supported by our data, has been removed from our revised manuscript.

On page 15 In 538, The increase in LPEAT activity is not supported and again this is over interpretation of the data. There is not sufficient data to support the increase in LPEAT activity directly from the assessment of the levels of PE and LPE independently of the assessment of PLA2 activity.

Response: We thank the reviewer for the valuable comments. We have removed the discussion on LPEAT activity and revised the discussion accordingly:

Interestingly, we observed a significant increase of the ratio of sn-2 PUFA-containing 1,2-diacyl GPs to their corresponding isomers in AD mice compared to WT mice, and most of the ratios also showed significant changes at 8 months, indicating that the sn-1 PUFA-containing 1,2-diacyl GPs might be more degraded than their corresponding isomers in AD as the abundance of both sn-isomers decreased. Previous reports also have demonstrated that the activities of various enzymes were significantly changed in AD patients⁶⁸, including acyltransferase, phosphodiesterase, LPLATs, and other enzymes, which involved in the remodeling processing of acylating lyso-GP into 1,2-diacyl GP to maintain the composition of neural membranes^{70,71}. The activities of these enzymes were likely also altered in the complex lipid metabolism processes of AD mouse brains considering both increased cPLA2 activities and the elevated ratio of sn-2 PUFA-containing 1,2-diacyl GPs to their corresponding isomers in AD groups. Moreover, we found that the ratios of GP/lyso-GP across different GP classes showed varied alteration trends between WT and AD mice brain. Therefore, it is possible that some other enzymes involved in the lipid metabolic pathway of AD mice brains were also altered^{63,68,70}. The exact molecular mechanisms underlying aging and AD disease pathologies correlated with lipid metabolism warrant comprehensive research.

Overall, the present results, together with previous findings from literature, suggested that it is important to investigate the dysregulated lipid metabolism in AD pathology at GP sn-position resolved level. Beyond the sn-position resolved GP quantification results, orthogonal enzyme activity assay also disclosed altered cPLA2 activity in different brain regions in aging and AD progression, suggesting potential alteration of enzyme activities in the GP remodeling pathways. We recognize that in the absence of clear mechanistic validation, our MS data should not be overinterpreted. However, the fact that they are consistent with the above independent studies suggests that a more systematic investigation of GP-related pathways in AD is warranted.

The cPLA2 activity kit from Abcam (ab133090) cannot determine between types of PLA2 activities, but this limitation is not discussed and the results are over-interpreted in the results section.

Response: We thank the reviewer for these valuable suggestions. Various isoforms of PLA2 exist in mouse brain, the other two PLA2 enzymes, secretory phospholipase A2 (sPLA2), calcium-independent phospholipase A2 (iPLA2), were removed by membrane filter and inhibited by bromoenol lactone, respectively, according to manufacturer's instructions. Thus, the PLA2 activity measured by the cPLA2 activity kit was largely considered from cPLA2. The limitation of the cPLA2 activity assay has been discussed and we now have included these in the Methods section. The limitation of the cPLA2 activity assay has been discussed as below.

Another limitation of our study is that, although we performed orthogonal enzyme activity analysis to facilitate further exploration of the lipidomics data, it did not comprehensively cover all possible related enzymes. For example, we tested cPLA2 activities, but it is crucial to note that, besides cPLA2, various other isoforms of PLA2 including secretory phospholipase A2 (sPLA2), calcium-independent phospholipase A2 (iPLA2), also exist in brain regions⁹. Given the diversity of PLA2 subtypes and additional enzymes, such as LPLATs, phospholipase A1s, acyl-CoA synthases, and transacylases, involved in lipid remodeling, further investigation into the changes in the

29

activities of multiple enzymes is necessary in the context of AD. Nevertheless, our data provides insights into possible enzyme activity alterations in aging and AD progression.

These data are highly impactful to the field, but it is difficult to interpret the biological advances from these data as the paper is written. Many of the data are over-interpreted, the discussion is largely in the result section and the limitations of the methodologies are not adequately discussed. If these major concerns are addressed, this manuscript would have a strong impact in the field.

Response: We truly appreciate the reviewer's positive evaluation of our work. We are also grateful to the reviewer for the valuable comments and suggestions which will of course help strengthen our manuscript. In response, we have revised the Results and Discussion sections to accurately interpret the results. We have also moved some descriptions from the Results to the Discussion section. The limitations of the methodologies, the possible solutions and future directions have also been discussed. Specifically:

Several limitations exist in our study. Firstly, for the very low abundance isomers of GPs with

complicated fatty acyl compositions, although they might be identified in our workflow from MS/MS spectra in negative mode, they were not annotated from the HRdm drift spectra to enhance the identification accuracy and confidence. This limitation would be addressed by employing high-sensitivity nano-ESI-MS and next generation IMS with higher resolution. The challenges of annotating GPs with multiple major fatty acyl compositional isomers could potentially be overcome by combining high-resolution IM-MS with advanced chromatographic separation, including extended chromatographic gradient and multidimensional chromatographic separation. Secondly, lipid C&C positional isomers could not be differentiated on this system since it requires IM Rp~1000 for baseline separation. Incorporating additional sample treatment steps such as novel derivatization/fragmentation strategies^{7,18,26,28} to pinpoint C&C information will ultimately achieve complete in-depth structural profiling of the lipidome. Thirdly, in our CCS prediction model, although small variances are still inevitably present between the experimental and predicted CCS values, we reason that involving more stereospecific 3D MDs would provide a more accurate prediction model. Additionally, we envision that a joint effort from the whole lipidomics community to expand the quantity of empirical measurements within the CCS database will be an invaluable addition. Another limitation of our study is that, although we performed orthogonal enzyme activity analysis to facilitate the further exploration of the lipidomics data, it did not comprehensively cover all possible related enzymes. For example, we tested cPLA2 activities, but it is crucial to note that, besides cPLA2, various other isoforms of PLA2 including secretory phospholipase A2 (sPLA2), calcium-independent phospholipase A2 (iPLA2), also exist in brain regions⁹. Given the diversity of PLA2 subtypes and additional enzymes, such as LPLATs, phospholipase A1s, acyl-CoA synthases, and transacylases, involved in lipid remodeling,

30 further investigation into the changes in the activities of multiple enzymes is necessary in the context of AD. Nevertheless, our data provides useful insights into possible enzyme activity alterations in aging and AD progression. It is worth mentioning that although we tried to minimize postmortem lipid changes that may occur in the brain by minimizing the euthanasia to tissue collection time, the possible oxidation of GPs is unavoidable.

Of additional note, it remains to be seen if the dynamics of bioactive lipid mediators, including eicosanoids and resolvins which are potentially evoked by the release of PUFAs from GPs, could offer new insights for early diagnosis and therapeutic strategies. Additionally, we plan to utilize imaging mass spectrometry in conjunction with high-resolution ion mobility, along with our established database, to map the spatial distribution of sn-position resolved GPs in a region-specific manner across the entire brain. Collective studies may be possible to provide new perspective for early diagnosis, preventive or therapeutic options for AD. In the future, together with biological validation, our strategy could serve as an enabling tool not only providing the in-depth lipid structural characterization but also sensitively monitoring the differential expression of enzymes involved in GP remodeling to ultimately provide crucial mechanistic insights into many disease pathologies.

Taken together, following the reviewer's critical and insightful comments, we have extensively revised the manuscript to more clearly illustrate the potential biological insights by our analytical advancement and to reasonably interpret our data with a more conservative tone. We believe the quality of the manuscript has been substantially improved with your generous help, and we are sincerely grateful for your insightful comments and suggestions.

31

Version 2:

Reviewer comments:

Reviewer #1

(Remarks to the Author)

The authors performed extensive experiments to address my comments. Very respectful!

Although I still have concerns about the false positives and artifacts, I believe the authors had tried their best.

I emphasize that it is necessary to manually check the deconvolved peaks from HRdm, and the number of deconvolved peaks should be restricted to major compositions of isomers.

With these considerations, I think the work could be accepted at its current form. We should let readers and the users of this

technology to judge whether it is useful or not.

Reviewer #3

(Remarks to the Author)

Accept

Author Rebuttal letter:

Reviewers' comments:

Reviewer #1 (Remarks to the Author):

The authors performed extensive experiments to address my comments. Very respectful!

Although I still have concerns about the false positives and artifacts, I believe the authors had tried their best.

I emphasize that it is necessary to manually check the deconvolved peaks from HRdm, and the number of deconvolved peaks should be restricted to major compositions of isomers.

With these considerations, I think the work could be accepted at its current form. We should let readers and the users of this technology to judge whether it is useful or not.

Response: We are grateful to the reviewer for the recognition and acknowledgement of our efforts. We thank the reviewer for recommendation of acceptance. We agree with the reviewer and also mentioned in our manuscript that manually checking the data is necessary and the number of identifications should be restricted to major compositions of isomers. Our carefully constructed 4D database could also serve as a good reference for users to judge between real peaks and potential artifacts.

Reviewer #3 (Remarks to the Author):

Accept

Response: We thank the reviewer for their positive evaluation of our manuscript. We appreciate the recommendation of acceptance of our manuscript for publication in Nature Communications.

Editor's requests:

Thank you for submitting your manuscript "Spatially and temporally probing distinctive glycerophospholipid alterations in Alzheimer's disease mouse brain via high-resolution ion mobility-enabled sn-position resolved lipidomics" to Nature Communications. I am delighted to say that we are happy, in principle, to publish it under the open access CC BY license (Creative Commons Attribution 4.0 International License).

First, we ask you to revise your paper to address our editorial requests (in the attached Author Checklist).

Failure to comply with our editorial requests will result in further revisions and delays in accepting your manuscript. Please also see the Nature Communications formatting

1
instructions for further information.

Response: We are grateful to the reviewers for their valuable feedback and insightful suggestions. We are delighted to know this work is accepted in principle. We carefully followed the requests in the author checklist and revised our manuscript accordingly. We hope the revision can fulfill the rigorous standard of Nature Communications.

FEATURED IMAGE

If you wish, you can provide an interesting image (but not an illustration or schematic) for consideration as a Featured Image on the Nature Communications homepage. The file should be 1200x675 pixels in RGB format and should be uploaded as a Related Manuscript File with the title "featured image suggestion". In addition to our home page, we may also use this image (with credit) in other journal-specific promotional material. If your featured image is chosen you will need to complete a Licence to Publish form which will be sent to you via DocuSign at a later stage.

Response: We do not wish to provide a featured image.

OPEN ACCESS

Nature Communications is a fully open access journal. Articles are made freely accessible on publication under a CC BY license (Creative Commons Attribution 4.0 International License). This license allows maximum dissemination and re-use of open access materials and is preferred by many research funding bodies.

For further information about article processing charges, open access funding, and advice and support from Nature Portfolio, please visit <http://www.nature.com/ncomms/about/open-access>

At acceptance, you will be provided with instructions for completing this CC BY license on behalf of all authors. This grants us the necessary permissions to publish your paper.

Response: We understand and permit publication of our manuscript in open access. We will complete the CC BY license on behalf of all authors when it is provided.

ORCID

Nature Communications is committed to improving transparency in authorship. As part of our efforts in this direction, we are now requesting that all authors identified as "corresponding author" create and link their Open Researcher and Contributor Identifier (ORCID) with their account on the Manuscript Tracking System (MTS) prior to acceptance. ORCID helps the scientific community achieve unambiguous attribution of all scholarly contributions. For more information please visit <http://www.springernature.com/orcid>

For all corresponding authors listed on the manuscript, please follow the instructions in the link below to link your ORCID to your account on our MTS before submitting the final version 2

of the manuscript. If you do not yet have an ORCID you will be able to create one in minutes. <https://www.springernature.com/gp/researchers/orcid/orcid-for-nature-research>

IMPORTANT: All authors identified as "corresponding author" on the manuscript must follow these instructions. Non-corresponding authors do not have to link their ORCIDs but are encouraged to do so. Please note that it will not be possible to add/modify ORCIDs at proof. Thus, if they wish to have their ORCID added to the paper they must also follow the above procedure prior to acceptance.

To support ORCID's aims, we only allow a single ORCID identifier to be attached to one account. If you have any issues attaching an ORCID identifier to your MTS account, please contact the Platform Support Helpdesk.

Response: Thank you for the guidance. We confirm that all the authors of this manuscript have linked their ORCIDs.

POLICIES

In recognition of the time and expertise our reviewers provide to Nature Communications's editorial process, we formally acknowledge their contribution to the external peer review of articles published in the journal. All peer-reviewed content will carry an anonymous statement of peer reviewer acknowledgement, and for those reviewers who give their consent, we will publish their names alongside the published article. For more information, please refer to our FAQ page at <https://www.nature.com/documents/ncomms-reviewer-information.pdf>

Nature Portfolio journals encourage authors to share their step-by-step experimental protocols on a protocol sharing platform of their choice. Where such protocols are available, please provide a DOI or other citation details in the paper. Nature Portfolio's Protocol Exchange is a free-to-use and open resource for protocols; protocols deposited in Protocol Exchange are citable and can be linked from the published article. More details can found

at <https://www.nature.com/protocolexchange/about>

Response: We acknowledge the policies of Nature Communications. Our experiment protocols are clearly outlined in the manuscript and may not require additional information to follow. Any further requests on our experiments can be directed to the corresponding author.

AUTHOR CHANGES DURING PREVIOUS REVISIONS

If there have been any changes to the author list since your initial submission, please use this approval form www.nature.com/documents/nr-author-list-change-form.pdf, arranging for all authors on your paper to sign the statement confirming that they agree to the author list being changed, and add this document to your resubmission. Please also add an explanation of the changes to your cover letter.

Response: We do not make any changes to the author list during this round of revision. We added Megan Braun to the author list in the first-round revision due to her contribution in animal

3

experiments during our revision, all the authors signed on the approval form of author list. The approval form of author list has been uploaded.

SUBMISSION

Authors submitting revised manuscripts should see our brief guide to submission and Guide to Authors. Please submit the following files for consideration by the editorial team and the referees:

1. Cover letter (this is optional and will not be accessible to the reviewers)
2. A separate 'response to referees' letter that addresses the referees' and editors comments in a point-by-point manner
3. If a reporting checklist was requested by the editor or previously provided with your submission, please supply an up-to-date version
4. File(s) containing the manuscript text, references, tables and figures with legends (for formatting guidelines see the author instructions page)
5. Supplementary information files, including copies of any related manuscripts under consideration at other journals

Response: Thank you for the guidance. We followed the submission guidelines and submitted the above files.
